# CryoBench: Diverse and challenging datasets for the heterogeneity problem in cryo-EM

**Minkyu Jeon**[1], **Rishwanth Raghu**[1], **Miro Astore**[2,3], **Geoffrey Woollard**[2,3,4], **Ryan Feathers**[1], **Alkin Kaz**[1], **Sonya M. Hanson**[2,3], **Pilar Cossio**[2,3], and **Ellen D. Zhong**[1]

[1]Department of Computer Science, Princeton University, Princeton, NJ, USA
[2]Center for Computational Biology,[3]Center for Computational Mathematics, Flatiron Institute, New York, NY, USA
[4]Department of Computer Science, University of British Columbia, Vancouver, BC, Canada

## Abstract

Cryo-electron microscopy (cryo-EM) is a powerful technique for determining high-resolution 3D biomolecular structures from imaging data. Its unique ability to capture structural variability has spurred the development of *heterogeneous reconstruction algorithms* that can infer distributions of 3D structures from noisy, unlabeled imaging data. Despite the growing number of advanced methods, progress in the field is hindered by the lack of standardized benchmarks with ground truth information and reliable validation metrics. Here, we introduce CryoBench, a suite of datasets, metrics, and benchmarks for heterogeneous reconstruction in cryo-EM. CryoBench includes five datasets representing different sources of heterogeneity and degrees of difficulty. These include *conformational heterogeneity* generated from designed motions of antibody complexes or sampled from a molecular dynamics simulation, as well as *compositional heterogeneity* from mixtures of ribosome assembly states or 100 common complexes present in cells. We then analyze state-of-the-art heterogeneous reconstruction tools, including neural and non-neural methods, assess their sensitivity to noise, and propose new metrics for quantitative evaluation. We hope that CryoBench will be a foundational resource for accelerating algorithmic development and evaluation in the cryo-EM and machine learning communities. Project page: https://cryobench.cs.princeton.edu.

## 1 Introduction

Single particle cryo-electron microscopy (cryo-EM) is a widely used imaging technique for visualizing biomolecular complexes at near-atomic resolution. A major challenge in cryo-EM structure determination is the task of reconstructing 3D density maps from an experimentally-derived dataset of 2D images [1]. These images characteristically exhibit extremely low signal-to-noise ratios (SNR), with unknown poses (3D orientations and 2D translations) of individual images, and there is often heterogeneity in both conformational and compositional aspects of the target protein complex (Fig. 1a). Despite these challenges, cryo-EM holds significant promise due to its ability to capture structural heterogeneity of intrinsic biological interest, which is typically inaccessible to structure prediction tools such as AlphaFold [2, 3]. Consequently, numerous heterogeneous reconstruction methods have been proposed in recent years to address these challenges [4, 5, 6, 7, 8, 9, 10].

Several state-of-the-art heterogeneous reconstruction methods leverage deep learning to capture the structural heterogeneity through a continuous latent variable, allowing structural changes to be represented as trajectories on a manifold [11, 12]. However, current research on heterogeneous reconstruction methods has two main limitations: 1) the absence of common benchmarks comparable to MNIST [13] or ImageNet [14] in computer vision that drove tremendous progress for the field and 2) the lack of metrics suitable for evaluation and comparison of methods.

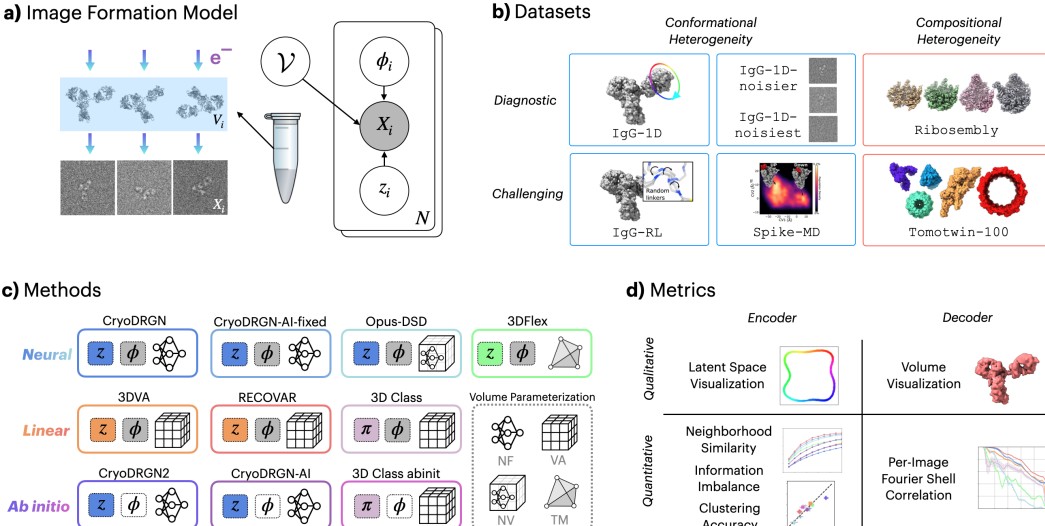

Figure 1: **Overview of CryoBench. a) Image formation model.** In cryo-EM, each image $X_i$ captures a molecule $V_i$ projected at an unknown pose $\phi_i$. A latent variable $z_i$ models the conformational space $\mathcal{V}$ that describes the heterogeneity among the set of molecules $\{V_i\}$. **b) Datasets.** CryoBench includes 5 synthetic datasets of varying difficulty, characterized by heterogeneity arising from conformational (i.e. shape) or compositional (i.e. identity) changes. **c) Methods.** Methods can be grouped into using either a continuous latent variable $z$ or discrete latent variable $\pi$ for modeling heterogeneity. Hidden variables assumed to be known are shown in gray. Volumes are represented as a neural field (NF), voxel array (VA), neural volume (NV), or tetrahedral mesh (TM). Generative models are colored blue for nonlinear neural methods; orange for linear generative models, pink for mixture models; and green for density-preserving motion models. **d) Metrics.** Summary of metrics used to assess both latent inference and volume reconstruction quality.

Previous methods development has relied on various types of datasets for benchmarking and validation. These include: a) real datasets known to have sensible conformal trajectories, allowing experts to perform qualitative benchmarking by visually assessing reconstructed volumes, b) datasets of synthetic blob-like volumes used to demonstrate that models' conformational latent spaces can represent simple 1D and 2D motions as parameterized motions, and c) pseudo-real motions, where real molecular structures are used, but simple motions such as rotations are applied to the subregions of the structure to generate a continuum of conformations. The lack of realistic and common benchmarks poses challenges in training models whose performance can generalize well across different datasets or in comparing existing heterogeneous reconstruction methods without relying on expert intuition. Moreover, because each method has been applied to different datasets designed to showcase different types of heterogeneity, practitioners cannot compare and determine the most promising method for their application. Finally, since these methods are mostly tested on real datasets with no ground truth, it's hard to assess whether a given method produces accurate results on a new dataset, or how much to trust its scientific conclusions.

In this work, we design new challenging datasets and evaluation metrics for the heterogeneous cryo-EM reconstruction task. Our datasets contain a range of types of heterogeneity and are synthetically generated in order to have ground truth poses, conformational states, and imaging parameters for quantitative evaluation. Our datasets range from having simple heterogeneity for diagnostic use to more challenging forms of heterogeneity for motivating new methods in cryo-EM (Fig. 1b). Using these datasets, we conduct extensive experiments on existing state-of-the-art methods (Fig. 1c). We introduce metrics for both qualitative and quantitative comparison of methods (Fig. 1d) and suggest new directions for methods development. CryoBench, including tools for dataset generation and model evaluation, is available at `https://cryobench.cs.princeton.edu/`.

## 2 Background and Related Work

**Heterogeneous reconstruction.** Over the past decade, advances in cryo-EM technology have led to major increases in data quality, enabling atomic resolution structure determination of static protein structures [15]. The ability to image heterogeneous systems both presents a unique opportunity for

structural biology, and poses a major computational challenge for 3D reconstruction. To address this opportunity, there has recently been an explosion of methods for heterogeneous cryo-EM reconstruction [4, 6, 7, 16, 8, 9, 17, 18, 19, 10, 20, 21, 22, 23, 24, 25]. Methods take a variety of approaches, with differences in volume representation (real, Fourier; explicit voxel grids, meshes, or implicit neural representations; Gaussian kernels in coordinate spaces; other basis functions), inference approach (statistical inference approaches, end-to-end gradient-based methods, minimax optimization), use of physically informed priors, and assumed inputs to the task [11, 12]. However, these methods typically use different datasets to illustrate their algorithm's efficacy, making it difficult to judge which methods outperform the others, and for which cases.

**Past Benchmarks.** While standard benchmarks in this field have yet to be established, there are some consistent datasets that have been used. For example, an experimental dataset that has been used for compositional heterogeneity is the different assembly states of the large subunit of a bacterial ribosome (EMPIAR-10076) [26], and a commonly used real dataset for conformational heterogeneity is the pre-catalytic spliceosome (EMPIAR-10180) [27]. However, it is also common to generate simulated cryo-EM datasets for which a ground truth is known [28, 6, 29, 30], but so far these are generated *ad hoc* for a given study, and not taken from a standard benchmark dataset for which the performance of other methods is known. There have been attempts at benchmark studies for heterogeneity where many methods are applied to a single dataset [31, 32], but this approach does not address performance across different types of heterogeneity such as conformational motions or compositional changes. Ideally, a common set of diverse benchmark datasets can be used by the community to compare different methods. Here, we focus on synthetic but challenging datasets where ground truth information is available for all latent variables, allowing for rigorous, quantitative benchmarking.

**Metrics.** In addition to benchmark datasets, metrics with which to assess heterogeneity methods themselves are also lacking. Metrics for assessing homogeneous 3D reconstruction methods, where only a single volume is achieved from a cryo-EM image stack, are more mature and often incorporate the Fourier shell correlation (FSC) to judge resolution [33, 34], and a gold standard approach for computing FSC [35] has been widely adopted by the community (though this has its own degree of controversy [36]). However, when it comes to evaluating heterogeneous reconstructions, metrics become much less straightforward, and the standard FSC-based approach is flawed as it provides a global assessment of resolution and is typically performed on independent half-sets and thus is only a self-consistency measure. Recent work has introduced new metrics either to improve priors for variational autoencoders or to disentangle latent embeddings based on ground truth [37, 38, 39]. However, analysis metrics that can compare different methods remain challenging. Here, we choose to use metrics that depend on knowing ground truth from synthetic datasets, which either use (a) a distributional volume reconstruction quality metric based on the FSC or (b) applying local neighborhood comparisons and clustering accuracy on latent representations [40, 41].

## 3 CryoBench Design

We generate CryoBench datasets by designing an ensemble of atomic models to serve as ground truth structures, followed by simulating the cryo-EM forward process to generate synthetic images.

### 3.1 Image Formation Model

Cryo-EM density volumes are first generated from each atomic model by simulating the electron scattering potential of each atom. From the volumes, we generate cryo-EM images $I_i$ following the standard image formation model in the Fourier domain:

$$I_i = C_i P_\phi V + \eta_i \tag{1}$$

where $C_i$ is the Contrast Transfer Function (CTF), $P_\phi$ is a slice operator corresponding to an tomographic projection of the volume $V$ according to the pose $\phi = (R, \mathbf{t}) \in \mathrm{SO}(3) \times \mathbb{R}^2$, and $\eta_i$ is additive white Gaussian noise. Images are sampled on a $D \times D$ grid with $D = 128$ by default unless otherwise specified. Additional details for each dataset are below with full dataset generation details given in Appendix A.

### 3.2 Conformational Heterogeneity

`IgG-1D`. We use an atomic model of the human immunoglobulin G (IgG) antibody complex (PDB: 1HZH). Conformational heterogeneity is produced by rotating a dihedral angle connecting one of

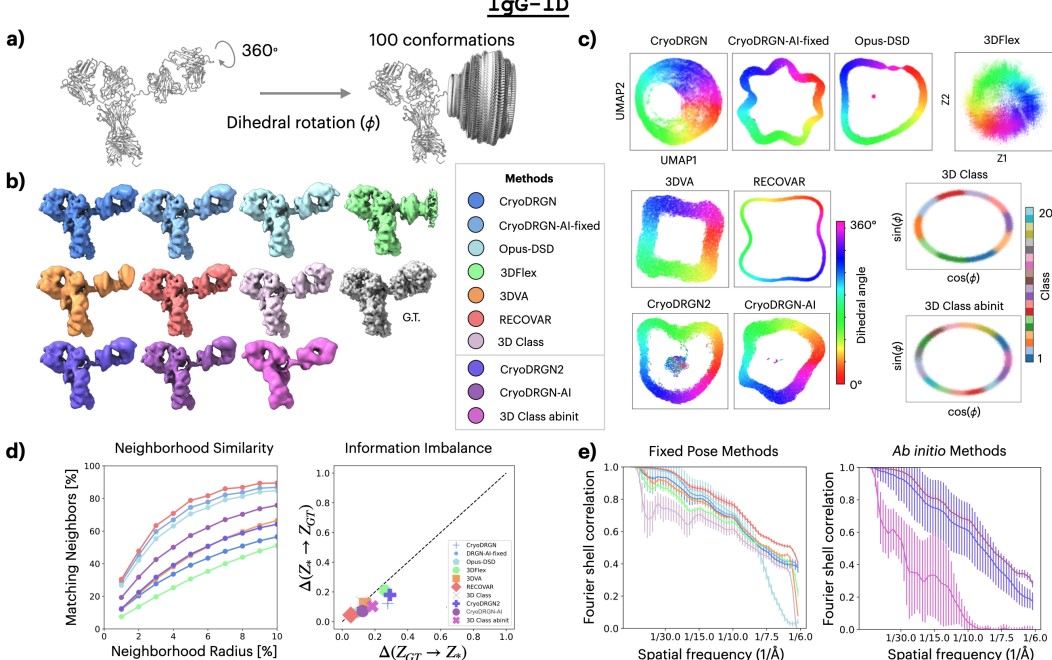

Figure 2: **IgG-1D results. a)** Dataset design. Conformational heterogeneity of an IgG antibody complex produced from a simple, 1D continuous circular motion. **b)** Representative reconstructed and ground truth (G.T.) volumes. **c)** Latent embeddings visualized by UMAP and colored by the G.T. dihedral angle parameterizing the circular motion. Discrete class assignments are plotted by G.T. dihedral angles. **d)** Latent embedding analysis by neighborhood similarity and information imbalance. **e)** Per-Image FSC curves. Each curve shows the average FSC curve across all conformations with error bars indicating the standard deviation. Colors in **b)**, **d)**, and **e)** correspond to methods shown in the legend. Additional results shown in Figure S15.

the fragment antibody (Fab) domains (Fig. 2a), simulating a simple one-dimensional continuous circular motion. This process yields 100 atomic models approximating the continuous dihedral rotation (at 3.6-degree intervals). We simulate 1,000 projection images for each conformation, apply CTF and add noise at a signal-to-noise (SNR) ratio of 0.01 to produce a dataset of 100k images.

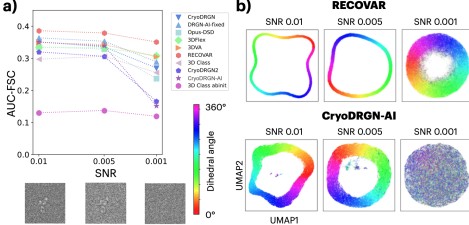

Figure 3: **IgG-1D with noise. a)** Per-Image FSC for each method at different noise levels. Markers correspond to the legend in Figure 2. **b)** Example cryo-EM images for different noise levels and latent embeddings visualized by UMAP for CryoDRGN-AI. Additional results shown in Figure S13, S18, and S19.

`IgG-1D-noisier` and `-noisiest`. As one of the most salient characteristics of cryo-EM images is the high degree of noise, we also create versions of the `IgG-1D` dataset at SNR 0.005 for `IgG-1D-noisier` and 0.001 for `IgG-1D-noisiest` to test the robustness of each method to the amount of noise.

`IgG-RL`. Many protein complexes, including IgG, possess relatively rigid domains connected by a disordered peptide linker (e.g. exemplifying "beads on a string"). To model this more realistic and complex motion and provide a challenging case of heterogeneity, we generate random conformations for the linker connecting the Fab to the rest of the IgG complex (Fig. 4b). We identified a sequence of 5 residues as the linker and generated random realizations of its structure by sampling the backbone dihedral angles according to the Ramachadran distributions of disordered peptides [42], using rejection sampling to eliminate structures with steric clashes. We generate 100 such atomic models and simulate 1k projections per conformation, apply CTF and add noise at an SNR of 0.01 to produce a dataset of 100k images.

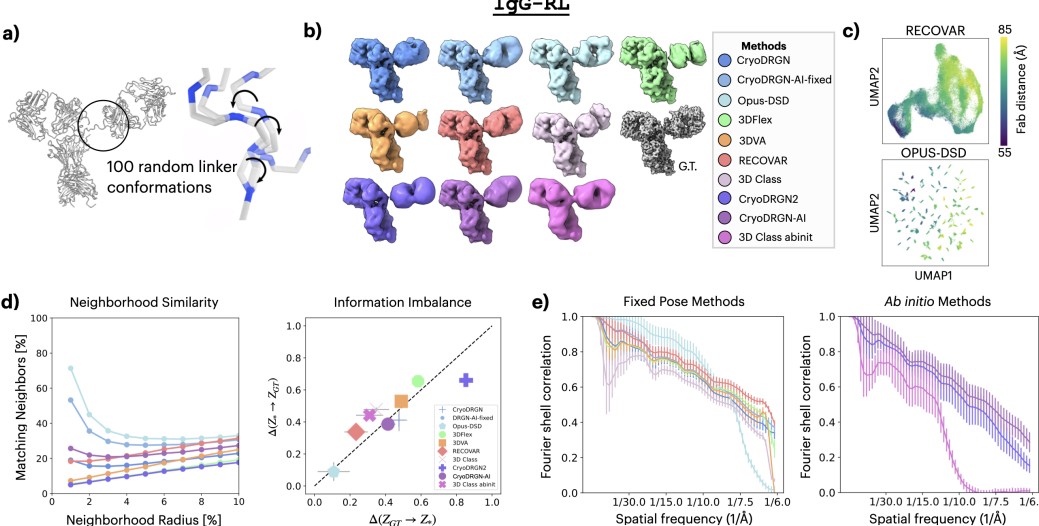

Figure 4: **IgG-RL results. a)** Dataset design. Conformational heterogeneity is produced by sampling 100 configurations of a peptide linker, randomly orienting the FAb domain in the IgG antibody complex. **b)** Representative reconstructed and ground truth (G.T.) volumes. **c)** The UMAP plots of RECOVAR and OPUS-DSD latent spaces colored by the distance between the FAb and the Fc domain in the G.T. volumes. **d)** Latent embedding analysis by neighborhood similarity and information imbalance. **e)** Per-Image FSC curves. Each curve shows the average FSC curve across all conformations with error bars indicating the standard deviation. Colors in (d), (e), and (f) correspond to methods shown in the legend. Additional results shown in Figure S14 and S16.

`Spike-MD.` For a challenging dataset containing detailed motions, we use a long-timescale molecular dynamics simulation as a source of ground truth models for dataset construction. Specifically, we use a simulation of the activating SARS-CoV-2 spike protein from Wieczór et al. [43]. Images for Spike-MD are generated at a higher resolution ($D = 256$) to assess the ability of methods to capture high-resolution motions. We generate a dataset of 100k projection images at an SNR of 0.1.

### 3.3 Compositional Heterogeneity

`Ribosembly.` We create a dataset modeling ribosome assembly as a simple example of compositional heterogeneity. In particular, these structures contain a common core that is successively grown through the addition of proteins and ribosomal RNA. We use the bacterial ribosome assembly states from Qin et al. [44] consisting of 16 different atomic models, which we color in groups according to structural similarity in Figure 6a. We generate a non-uniform number of images for each ground truth structure following the distribution in the original publication, apply the CTF, and add noise to an SNR of 0.01 to produce a dataset of 335,240 images.

`Tomotwin-100.` While cryo-EM is typically performed on purified samples or simple mixtures, in principle, the technique can be used to image complex mixtures e.g. from cellular lysates or *in situ* samples from cryo-ET. To create a challenging dataset of modeling compositional heterogeneity of this scale, we use atomic models from the curated set of cellular complexes in Rice et al. [45]. Here we use 100 out of the original set of 120 after excluding the 15 smallest and 5 largest complexes. Figure 7a shows 10 out of 100 ground truth volumes colored by size.

## 4 Evaluation Framework

As part of CryoBench, we consider methods for heterogeneous reconstruction with fixed (ground truth) poses, and the more challenging task of *ab initio* reconstruction where no input poses are provided. We evaluate seven fixed pose state-of-the-art methods and three *ab initio* variants. The fixed pose methods include 3D Classification (3D Class) in cryoSPARC [46], 3DVA [8], 3DFlex [9], CryoDRGN [7], CryoDRGN-AI-fixed [47], Opus-DSD [18], and RECOVAR [19]. For *ab initio* methods, we use multiclass *ab initio* reconstruction in cryoSPARC (3D Class abinit) [46], CryoDRGN2 [48], and CryoDRGN-AI [47]. An overview of these methods and training details can be found in Appendix B.

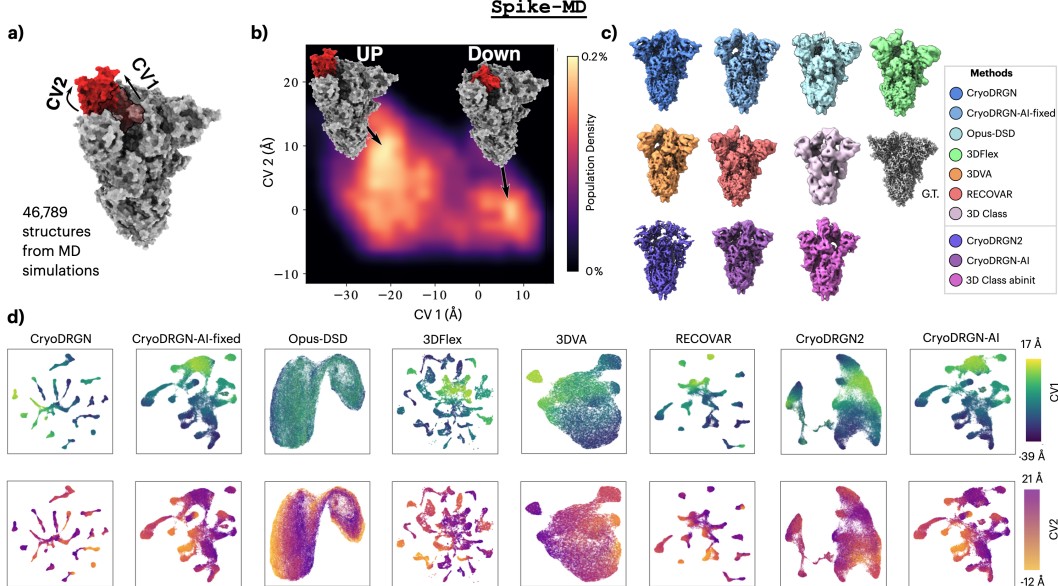

Figure 5: **Spike-MD results. a)** Dataset design. 46,789 structures were sampled from a MD simulation of the SARS-COV-2 spike protein, including opening of the receptor binding domain (RBD, shown in red). The motion in the MD simulation can be described with two collective variables (CV), corresponding to opening and twisting of the RBD. **b)** The population density of molecular states projected onto these CVs. **c)** Representative reconstructed and ground truth (G.T.) volumes. **d)** Latent embeddings visualized by UMAP and colored by the first and second CV. Additional results shown in Figure S9, S17, and S24.

## 4.1 Analysis and Metrics

We present analyses and metrics for comprehensive comparison of existing state-of-the-art methods, addressing both qualitative and quantitative aspects. For qualitative evaluation, we visualize each method's distribution of latent embeddings, which collectively define a low-dimensional manifold capturing conformational and compositional variability in the particle images. We additionally sample representative density volumes for visual inspection.

For quantitative evaluation, we propose three metrics for comparison of embeddings: 1) Neighborhood Similarity, 2) Information imbalance, and 3) Adjusted Rand Index (ARI). To assess reconstructed volumes, we use "Per-Image Fourier Shell Correlation (FSC)" as a metric that jointly evaluates image conformation estimation and reconstruction quality [6]. Moreover, we compute the pose error for *ab initio* methods in Appendix C.7.

### 4.1.1 Embedding Comparisons

**Neighborhood Similarity.** To quantify the similarity between local neighborhoods of the learned embedding spaces and a ground truth heterogeneity embedding, we first use a generalization of neighborhood similarity [40]. We quantify the percentage of matching neighbors (pMN) with respect to the ground truth that are found within a neighborhood radius $k$,

$$\mathrm{pMN}(k) = \frac{100}{k\,N} \sum_i S(\mathrm{NN}_i^k, \mathrm{gt}_i^k) \,, \tag{2}$$

where $\mathrm{NN}_i^k$ and $\mathrm{gt}_i^k$ are the $k$-neighbors of point $i$ in the embedding and ground truth, respectively; $S$ is the number of neighbor matches between the two lists and $N$ is the total number of data points. Each data point is an image projected onto the embedding space. The details of ground truth embeddings are described in Appendix B.8. The local neighbors are defined using Euclidean distance on embeddings. We plot pMN as a function of the neighborhood radius in terms of the percentage of $k$-points relative to the entire the dataset. pMN values close to 100% indicate that ground truth and embedding neighborhoods are very similar at the given radius k, pinpointing that the local proximity between the points is preserved relative to the ground truth. See Appendix B.9 for more details.

**Information imbalance.** Information imbalance is a statistical test that compares how much one feature space, A, is contained in another, B, within a $k$-sized local neighbourhood, using the embedding

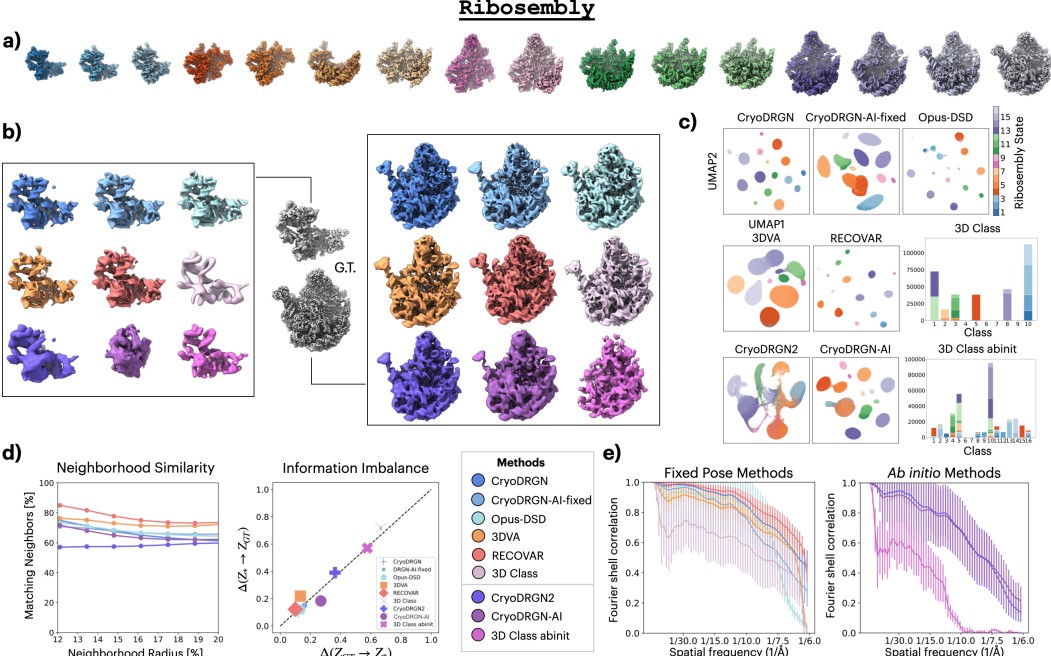

Figure 6: **Ribosembly results. a)** Dataset design. 16 ground truth (G.T.) ribosome assembly states. **b)** Representative reconstructed and G.T. volumes. **c)** Latent embeddings visualized by UMAP and colored by the 16 G.T. states. **d)** Latent embedding analysis by neighborhood similarity and information imbalance. **e)** Per-Image FSC curves. Each curve shows the average FSC curve across all conformations with error bars indicating the standard deviation. Colors in (b), (d), and (e) correspond to methods shown in the legend. Additional results shown in Figure S20.

distances $d_A$ and $d_B$ [41]. In the context of cryo-EM, this is a way of quantifying disentanglement of latent variables [38]. Information imbalance ranges between 0 and 1 and is defined as

$$\Delta(d_A \to d_B; k) = \Delta_{AB} = \frac{2}{N^2 k} \sum_{\substack{i,j \\ \text{s.t.} r_{ij}^A < k}} r_{ij}^B \,, \tag{3}$$

where $r_{ij}^A$ is the rank from point $i$ to $j$ under distance $d_A$, and $r_{ij}^B$ is the rank from point $i$ to $j$ under distance $d_B$. This comparison is not symmetric in general, i.e. $\Delta_{AB} \neq \Delta_{BA}$. Ref. [41] characterized four regimes of $\Delta_{AB}$ plotted against $\Delta_{BA}$: A and B have equivalent information with $0 \approx \Delta_{AB} \approx \Delta_{BA}$; A and B have orthogonal information with $1 \approx \Delta_{AB} \approx \Delta_{BA}$; B is contained in A with $0 \approx \Delta_{AB}$ and $\Delta_{BA} \approx 1$, and thus B could be better predicted from A using some classifier, than predicting A from B; A and B contain both identical and independent information with $0 \ll \Delta_{AB} \approx \Delta_{BA} \ll 1$ along the diagonal. Here, we compared the ground truth heterogeneity coordinate for each dataset (see Appendix B.10 for details) to the inferred latent embeddings from the various cryo-EM heterogeneity methods. We use Euclidean distance for $d$. If the inferred heterogeneity latent embedding and the ground truth heterogeneity latent embedding have an information imbalance of (0,0) then they are locally equivalent. The information imbalance plot can also be used to assess whether the latent embeddings capture other non-structural sources of image heterogeneity such as pose and CTF (See Appendix C.5 for more details).

**Clustering Accuracy.** To assess each method's ability to separate compositional states, we first use $k$-means clustering on the latent embeddings with $k$ equal to the number of ground truth states ($k = 16$ for Ribosembly and $k = 100$ for Tomotwin-100). We then compare these cluster assignments to the true structural labels for each particle using the Adjusted Rand Index (ARI) and Adjust Mutual Information (AMI) [49] [50]. We note that this metric is sensitive to the performance of the clustering, and might differ with alternative algorithms to $k$-means.

### 4.1.2 Volume Metrics

We use Fourier Shell Correlation (FSC), a standard volume comparison metric in cryo-EM, to evaluate reconstruction quality against ground truth structures. Typically, FSC compares two independent

| Method | IgG-1D | | IgG-RL | | Ribosembly | | Tomotwin-100 | | Spike-MD | |
|---|---|---|---|---|---|---|---|---|---|---|
| | Mean (std) | Median | Mean (std) | Med | Mean (std) | Med | Mean (std) | Med | Mean (std) | Med |
| CryoDRGN | 0.351 (0.028) | 0.356 | 0.331 (0.016) | 0.333 | 0.412 (0.023) | 0.415 | **0.316 (0.046)** | **0.321** | 0.340 (0.009) | 0.340 |
| CryoDRGN-AI-fixed | 0.364 (0.002) | 0.364 | 0.348 (0.012) | 0.350 | 0.372 (0.032) | 0.375 | 0.202 (0.044) | 0.207 | 0.301 (0.012) | 0.303 |
| Opus-DSD | 0.335 (0.026) | 0.339 | 0.343 (0.016) | 0.346 | 0.362 (0.083) | 0.382 | 0.237 (0.049) | 0.251 | 0.229 (0.027) | 0.242 |
| 3DFlex | 0.335 (0.003) | 0.335 | 0.337 (0.007) | 0.337 | - | - | - | - | 0.304 (0.011) | 0.306 |
| 3DVA | 0.349 (0.004) | 0.350 | 0.333 (0.014) | 0.335 | 0.375 (0.038) | 0.375 | 0.088 (0.04) | 0.077 | 0.324 (0.010) | 0.323 |
| RECOVAR | **0.386 (0.005)** | **0.388** | **0.363 (0.011)** | **0.363** | **0.429 (0.018)** | **0.432** | 0.258 (0.109) | 0.254 | **0.362 (0.011)** | **0.365** |
| 3D Class | 0.297 (0.019) | 0.291 | 0.309 (0.01) | 0.307 | 0.289 (0.081) | 0.288 | 0.046 (0.026) | 0.037 | 0.307 (0.023) | 0.308 |
| CryoDRGN2 | 0.32 (0.062) | 0.342 | 0.301 (0.03) | 0.306 | 0.341 (0.059) | 0.356 | 0.076 (0.016) | 0.072 | 0.245 (0.042) | 0.260 |
| CryoDRGN-AI | **0.351 (0.01)** | **0.352** | **0.329 (0.028)** | **0.333** | **0.341 (0.083)** | **0.367** | 0.072 (0.015) | 0.072 | **0.279 (0.017)** | **0.281** |
| 3D Class abinit | 0.13 (0.046) | 0.119 | 0.184 (0.022) | 0.188 | 0.144 (0.036) | 0.138 | 0.032 (0.012) | 0.031 | 0.206 (0.009) | 0.208 |

Table 1: **Area under the Per-Image FSC Curve.** FSC curves are computed after masking out background noise. See Appendix C.1.3 and Figure S6 for unmasked performance. We also provide results for IgG-1D with a mask only around the FAb in Table S3, Table S4, and Figure S7.

reconstructions from separate dataset halves, with resolution determined at an FSC cutoff of 0.143 [35]. Here, we compare reconstructed volumes against ground truth volumes and compute the Area Under the FSC Curve ($FSC_{AUC}$) as a summary statistic. We show that compared to an FSC cutoff, the AUC is more sensitive to structural differences amongst the ground truth volumes (Appendix C.1.2, Fig S5).

For heterogeneity analysis, we use the *Per-Image FSC* [6] as a distributional metric that jointly assesses conformation estimation and reconstruction quality. Here, we use a set of $N$ images from the dataset (either sampled at random or uniformly for each conformation). For each image, we reconstruct an associated volume at its inferred latent coordinate. We then compute the $FSC_{AUC}$ between the reconstructed volume and the ground truth volume for the image. We also investigated two alternative methods for sampling representative volumes to compute volume metrics: *Sample FSC* and *Per-Conformation FSC*. See Appendix C.2 and Figure S8-S11 for more details.

## 5 Results

We present the benchmarking results for each dataset in Figures 2-7, and we report summary statistics of volume metrics in Table 1.

**IgG-1D.** All methods generally demonstrate interpretable latent spaces (Fig. 2c) and well-preserved local neighborhoods relative to the ground truth quantified by a high neighborhood similarity and close to the equivalent region, (0,0), in the information imbalance plane (Fig. 2d). RECOVAR exhibits a latent space structure most consistent with the ground truth manifold, as shown by the embedding metrics in Figure 2d, while also achieving the highest reconstruction quality as shown in Table 1. In contrast, the rotating Fab domain is not well captured by 3DFlex (Fig. 2b), indicating limitations in effectively representing intricate conformal motions. With increasing noise levels, we find that the volume metrics decrease as expected (Fig. 3a). *Ab initio* methods fail to reconstruct at the lowest SNR (Fig. 3b).

**IgG-RL.** The overall performance of IgG-RL in Table 1 is lower than that of IgG-1D, attributed to the more challenging dataset design. In particular, Fab orientation inference is challenging, limiting the resolution of the moving Fab (Fig. 4c). Nonetheless, some methods, such as Opus-DSD or CryoDRGN-AI-fixed, are able to conserve high neighborhood similarity and equivalent information with the ground truth for small radii (Fig. 4d). RECOVAR demonstrates superior performance in the FSC metric. Among *ab initio* methods, CryoDRGN-AI generally outperforms the others. Despite the random orientations of the moving Fab, *ab initio* methods are able to align on the fixed region.

**Spike-MD.** The high number of unique structures (46k) makes this dataset comparatively challenging with some methods modeling the continuous motion better than others by visual inspection (Fig. 5c). UMAP visualizations of the latent space reveal that methods have learned a variety of topologies, with neither of the ground truth coordinates correlating well with the different UMAP embeddings (Fig. 5d). As shown in Figure S24, the neighborhood similarity metric also shows a poor local neighborhood overlap (<35%) at small-to-intermediate radii suggesting that there is low neighborhood correlation with the ground truth, indicating that the flexibility is indeed a challenging inference task. The *Per-Image FSC* plot for Spike-MD is provided in Figure S12. Future work could explore more custom analyses and metrics for MD data, and we hope to enable future methods development connecting MD simulations with cryo-EM.

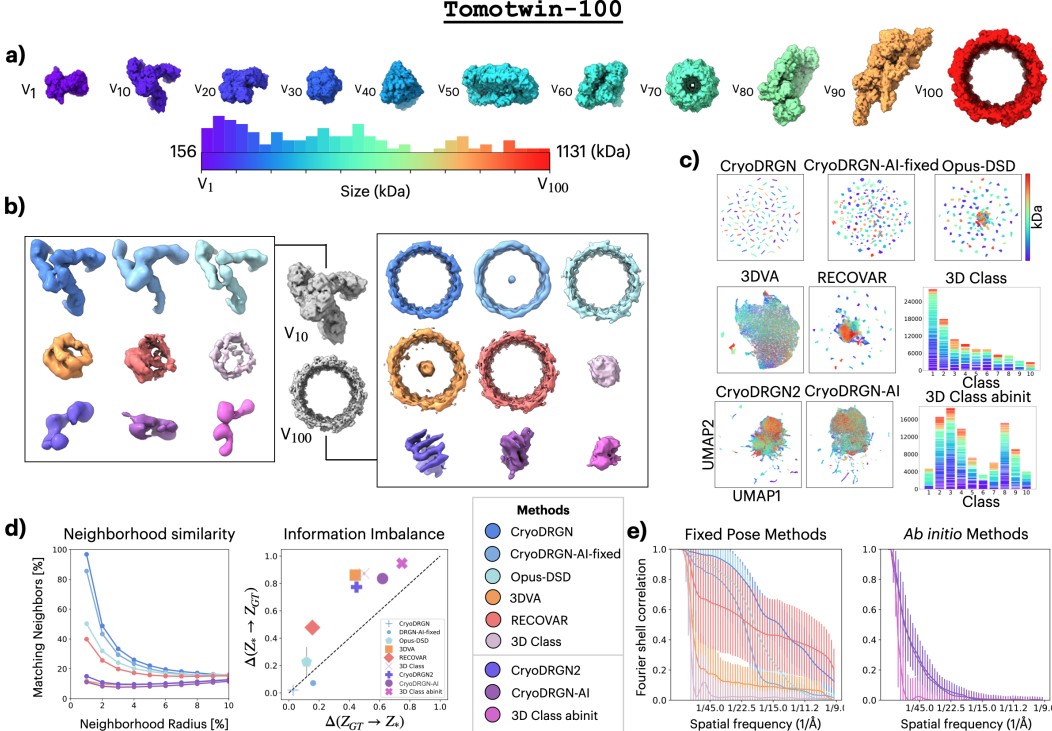

Figure 7: **Tomotwin-100 results. a)** Dataset design. Density maps for 10 out of 100 ground truth volumes and a histogram of molecular weights. **b)** Representative reconstructed and ground truth (G.T.) volumes. **c)** Latent embeddings visualized by UMAP and colored by the 100 G.T. molecular weight. **d)** Latent embedding analysis by neighborhood similarity and information imbalance. **e)** Per-Image FSC curves. Each curve shows the average FSC curve across all conformations with error bars indicating the standard deviation. Colors in (b), (d), and (e) correspond to methods shown in the legend. Additional results shown in Figure S21.

**Ribosembly.** Methods are largely capable of reconstructing the ground truth states (Fig. 6b,e). However, latent embeddings do not perfectly cluster each ground truth assembly state (Fig. 6c), quantified in Table 2. The neighborhood similarity ranges are intermediate, from 50-80%, for a neighborhood clustering size k corresponding to the true number of images belonging to each state (Fig. 6d). Here, the neighborhood similarity uses a rank-size embedding for ground truth heterogeneity, while information imbalance uses the voxel intensities, where similar assembly states would have a close distance. RECOVAR demonstrates better clustering, with CryoDRGN-AI also performing well amongst *ab initio* methods. In contrast, both 3D Class and 3D Class abinit exhibit a mixture of ground truth states within each class, leading to lower volume accuracy as indicated in Table 1.

Table 2: **Clustering Accuracy.** Adjusted Rand Index (ARI) and Adjusted Mutual Information (AMI) between ground truth labels and predicted labels for each particle image. Predicted labels are obtained by $k$-means clustering the particle latent embeddings, with $k$ set to the number of ground truth structures.

| Method | Ribosembly | | Tomotwin-100 | |
|---|---|---|---|---|
| | ARI | AMI | ARI | AMI |
| CryoDRGN | 0.873 | 0.935 | **0.956** | **0.983** |
| CryoDRGN-AI-fixed | 0.624 | 0.771 | 0.791 | 0.906 |
| Opus-DSD | 0.891 | 0.934 | 0.500 | 0.781 |
| 3DVA | 0.666 | 0.823 | 0.058 | 0.335 |
| RECOVAR | **0.968** | **0.976** | 0.315 | 0.649 |
| CryoDRGN2 | 0.529 | 0.618 | **0.116** | **0.374** |
| CryoDRGN-AI | **0.644** | **0.729** | 0.086 | 0.275 |

**Tomotwin-100.** *Ab initio* and 3D Classification-based methods fail to resolve the Tomotwin-100 dataset (Fig. 7b, Table 1). Linear methods (i.e., 3DVA and RECOVAR) also exhibit limited capacity to represent small-sized proteins (200 kDa) (Fig. 7b). Surprisingly, some fixed pose methods are able to cluster structures according to the ground truth (Fig. 7c, d), e.g. cryoDRGN has an almost perfect $k$-means clustering performance (Table 2). However, we note that fixed pose heterogeneous reconstruction is an unrealistic setting for cryo-EM of complex mixtures (where a common structural core is typically needed for upstream pose estimation). Current *ab initio* reconstruction methods

are not capable of recovering the 100 ground truth structures, posing a challenge for future methods development.

## 6 Conclusion and Future Directions

CryoBench presents: 1) five synthetic datasets containing conformational and compositional heterogeneity; 2) comprehensive analyses of ten existing state-of-the-art tools across these datasets; and 3) quantitative metrics and qualitative visualizations to compare these baselines. Our metrics assess representation learning, reconstruction quality, and the end-to-end heterogeneous reconstruction task. Overall, we find that quantification of the reconstruction performance aligns well with qualitative observations.

In addition to simple datasets that provide interpretable results for development and validation, we propose challenging datasets that push the frontiers of heterogeneous cryo-EM reconstruction. We anticipate that these datasets and metrics could inspire the exploration of new tasks for methods development, both in cryo-EM and within related areas of computer vision and computational imaging. These tasks may include generalization beyond training dataset distributions, high-resolution neural rendering, and the incorporation of biophysical priors into models. Our datasets are designed such that methods geared towards these challenges will also address key frontiers in cryo-EM and structural biology.

In future work, we aim to explore several limitations of the current benchmark. For example, we could use more complex or realistic noise statistics in image formation or simulate the joint presence of compositional and conformational heterogeneity. To explore realistic noise statistics in image formation, we tested cisTEM's (Computational Imaging System for Transmission Electron Microscopy) [51] multislice simulator, which considers artifacts from sample thickness, noise from explicitly modeled water molecules, and the effects of sample motion and radiation damage as a function of electron dose. However, we note that this simulator requires several minutes to generate each image. Since the images it produced and the performance of each method were qualitatively similar to those generated using our simpler Gaussian noise model, we did not follow this procedure for the benchmark datasets.

Through the active use of these datasets by the community, we anticipate that custom metrics designed for each type of heterogeneity will be fruitful, such as analyses of free energy landscapes and atomic-level motions in the case of MD data. Lastly, while the presented synthetic datasets provide a challenging setting for methods development, future versions of the benchmark could include non-structural sources of heterogeneity found in real cryo-EM datasets, such as junk particles and non-uniform pose distributions. We anticipate that these developments will facilitate the application of novel methods towards biological discovery on real data.

## 7 Data and Software Availability

CryoBench datasets and tools are available at https://cryobench.cs.princeton.edu/.

### 7.1 Data Availability

CryoBench datasets are deposited to Zenodo with the following DOIs:

- Conf-het: https://doi.org/10.5281/zenodo.11629428
- Comp-het: https://doi.org/10.5281/zenodo.12528292
- MD: https://doi.org/10.5281/zenodo.12528784

We include the downsampled images ($D = 128$) analyzed in this study ($D = 256$ for Spike-MD) in .mrcs, .txt, and .star file formats, along with CTFs and pose data in Python pickle files. We also include the consensus volume (needed for some methods) and the mask used for FSC computation. Full-resolution images ($D = 256, 384$) and ground truth PDB files and volumes will be deposited to BioImage Archive [52]. We provide the datasets under the Creative Commons Attribution 4.0 International license.

### 7.2 Software Availability

Scripts for simulating cryo-EM images and computing metrics are available at [https://github.com/ml-struct-bio/CryoBench](https://github.com/ml-struct-bio/CryoBench).

## Acknowledgements

We thank M. Grzadkowski for software engineering support, D. Shustin for discussions on dataset generation, M. Brubaker and N. Grigorieff for feedback on an early version of this benchmark, and M. Aurèle Gilles and A. Singer for feedback on the manuscript. We acknowledge that the work reported herein was substantially performed using the Princeton Research Computing resources at Princeton University, which is a consortium of groups led by the Princeton Institute for Computational Science and Engineering (PICSciE) and Office of Information Technology's Research Computing. The Zhong lab gratefully acknowledges support from the Princeton Catalysis Initiative, Princeton School of Engineering and Applied Sciences, and Generate Biomedicines. The funders had no role in study design, data collection and analysis, decision to publish or preparation of the manuscript.

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

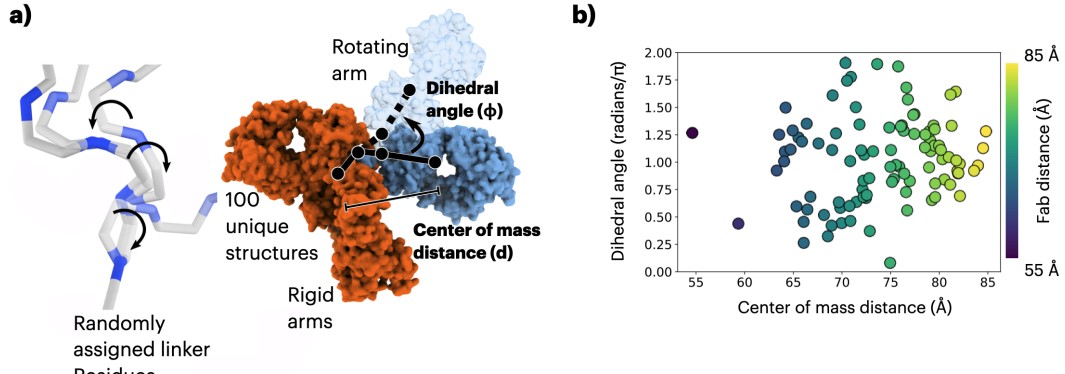

Figure S1: **Additional details for IgG-RL dataset design. a)** The dihedral angles of a linking peptide are chosen to randomly orient one arm of the IgG molecule in 100 unique conformations. **b)** The distribution of the 100 structures across coordinates can be characterized by the angle and distance between the rotating arm and the rigid arms.

# A  Dataset Design

## A.1  Generating IgG-1D

Starting from an atomic model of the human immunoglobulin G (IgG) antibody (PDB: 1HZH), conformational heterogeneity is produced by rotating a dihedral angle connecting one of the fragment antibody (Fab) domains (Fig. 2a), simulating a simple one-dimensional continuous circular motion. Specifically, we rotate the backbone $\psi$ angle of residue 230 in the heavy chain H. We sample 100 atomic models at 3.6 degree intervals to approximate a continuous 360 degree dihedral rotation. For each atomic model, the `molmap` command in ChimeraX [53] was used to generate the corresponding density volume at a resolution of 3 Å with a bounding box of dimension $D = 256$ pixels and a pixel size of 1.5 Å. Poses in Eq. 1 were uniformly sampled from $R \in \mathrm{SO}(3)$ and $t$ was sampled uniformly from $[20, 20]^2$ pixels. For the CTF, the accelerating voltage was set at 300 kV, spherical aberration at 2.7 mm, and amplitude contrast at 0.1. Defocus parameters were sampled randomly without replacement from EMPIAR-11247 [54]. Noise was added at a signal-to-noise (SNR) ratio of 0.01. See Appendix A.6 for a definition of the SNR. We simulate 1,000 images for each conformation to produce a dataset of 100k images. The dataset is then downsampled to $D = 128$ by Fourier cropping. We use the same projection images but increase the amount of added noise to SNR 0.005 and SNR 0.001 for `IgG-1D-noisier` and `IgG-1D-noisiest`, respectively.

## A.2  Generating IgG-RL

For `IgG-RL`, we identified a sequence of 5 residues (D232, K235, T236, H237, T238) from `1HZH` PDB as the linker and generated 100 random realizations of its structure by sampling the backbone dihedral angles according to the Ramachadran distributions of disordered peptides from Towse et al. [42], using rejection sampling to eliminate structures with steric clashes. Details are provided in S1. For each atomic model, the `molmap` command in ChimeraX [53] was used to generate the corresponding density volume at a resolution of 3 Å with a bounding box of dimension $D = 256$ pixels and a pixel size of 1.5 Å. Poses in Eq. 1 were uniformly sampled from $R \in \mathrm{SO}(3)$ and $t$ was sampled uniformly from $[20, 20]^2$ pixels. For the CTF, the accelerating voltage was set at 300 kV, spherical aberration at 2.7 mm, and amplitude contrast at 0.1. Defocus parameters were sampled randomly without replacement from EMPIAR-11247 [54]. Noise was added at a signal-to-noise (SNR) ratio of 0.01. We simulate 1,000 images for each conformation to produce a dataset of 100k images. The dataset is then downsampled to $D = 128$ by Fourier cropping.

## A.3  Generating Spike-MD

We sourced the individual MD structures from the enhanced sampling molecular dynamics simulations performed in [43]. Using the free-energy landscape calculated with these simulations for the wild-type Spike, we sampled molecular structures assuming a Boltzmann distribution with $T = 6000$ K. By

using an artificially high temperature, we were able to increase the number of sampled conformations—particularly in regions with a high free energy barrier connecting the receptor binding domain (RBD) open and closed states. This process resulted in 46,789 unique conformations. For each atomic model, the `molmap` command in ChimeraX [53] was used to generate the corresponding density volume at a resolution of 3 Å with a bounding box of dimension $D = 256$ pixels and a pixel size of 1.5 Å. Poses in Eq. 1 were uniformly sampled from $R \in$ SO(3) and $t$ was sampled uniformly from $[20, 20]^2$ pixels. For the CTF, the accelerating voltage was set at 300 kV, spherical aberration at 2.7 mm, and amplitude contrast at 0.1. Defocus parameters were sampled randomly without replacement from Walls et al. [55]. Noise was added at a signal-to-noise (SNR) ratio of 0.1. We simulated 100,000 images in total with at least 1 image per sampled conformation, resulting in approximately two images for each unique conformation.

## A.4 Generating Ribosembly

For `Ribosembly`, we used the 16 atomic models of the bacterial ribosome assembly states from Qin et al. [44]. We first centered all atomic models using the `move` in ChimeraX. Subsequently, the models were aligned to the last state (PDB: 8C8X) using `matchmaker` in ChimeraX. For each atomic model, the `molmap` command in ChimeraX [53] was used to generate the corresponding density volume at a resolution of 3 Å with a bounding box of dimension $D = 256$ pixels and a pixel size of 1.5 Å. Poses in Eq. 1 were uniformly sampled from $R \in$ SO(3) and $t$ was sampled uniformly from $[20, 20]^2$ pixels. For the CTF, the accelerating voltage was set at 300 kV, spherical aberration at 2.7 mm, and amplitude contrast at 0.1. Defocus parameters were sampled randomly without replacement from EMPIAR-10076 [26]. Noise was added at a signal-to-noise (SNR) ratio of 0.01. We simulate images with a non-uniform distribution following Qin et al. [44] listed below, totaling 335,240 particle images. The dataset is then downsampled to $D = 128$ by Fourier cropping.

PDB: 8C9C, 8C9B, 8C9A, 8C99, 8C98, 8C97, 8C96, 8C95, 8C94, 8C93, 8C92, 8C91, 8C90, 8C8Z, 8C8Y, 8C8X
$N_{particles}$: [9076, 14378, 23547, 44366, 30647, 38500, 3915, 3980, 12740, 11975, 17988, 5001, 35367, 37448, 40540, 5772]

## A.5 Generating Tomotwin-100

We created `Tomotwin-100` from the curated set of cellular complexes in Rice et al. [45]. Here we use 100 out of the original set of 120 after excluding the 15 smallest and 5 largest complexes by molecular weight. We centered all atomic models using the `move` in ChimeraX. Then, the `molmap` command in ChimeraX [53] was used to generate the corresponding volume map. For each atomic model, the `molmap` command in ChimeraX [53] was used to generate the corresponding density volume at a resolution of 3 Å with a bounding box of dimension $D = 384$ pixels and a pixel size of 1.5 Å. Poses in Eq. 1 were uniformly sampled from $R \in$ SO(3) and $t$ was sampled uniformly from $[20, 20]^2$ pixels. For the CTF, the accelerating voltage was set at 300 kV, spherical aberration at 2.7 mm, and amplitude contrast at 0.1. Defocus parameters were sampled randomly without replacement from EMPIAR-11247 [54]. Noise was added at a signal-to-noise (SNR) ratio of 0.01. Figure S2 illustrates all 100 ground truth volumes, with PDB codes listed below.

PDB: 2CG9, 6VGR, 5A20, 1UL1, 5LJO, 5CSA, 7WBT, 7SGM, 7BLR, 6ZQJ, 7NIU, 1U6G, 3ULV, 5JH9, 3D2F, 3CF3, 6LMT, 2RHS, 1BXN, 1N9G, 5HOS, 6CES, 7K5X, 7JSN, 6VN1, 1QVR, 2WW2, 6U8Q, 6KRK, 6Z80, 6LXK, 6WZT, 3MKQ, 6KSP, 2XNX, 7B7U, 6CNJ, 1SS8, 6X5Z, 7KJ2, 6KLH, 6PIF, 2DFS, 6AHU, 6F8L, 2VZ9, 7NHS, 6TGC, 6M04, 4XK8, 7E1Y, 7R04, 6IOD, 6BQ1, 7LSY, 7DD9, 3LUE, 7SFW, 7NYZ, 5O32, 6YT5, 6SCJ, 7EGE, 5VKQ, 6VZ8, 6W6M, 7T3U, 6TAV, 7E8H, 7ETM, 7AMV, 1G3I, 6Z3A, 7EGD, 7Q21, 6XF8, 6EMK, 6TA5, 6TPS, 7QJO, 7KDV, 7EGQ, 6LXV, 6GYM, 7OO1, 5GO4, 7BKC, 6MRC, 6JYO, 7WOO, 7EEP, 7MEI, 6GY6, 6DUZ, 7VTQ, 7EY7, 6Z6O, 4CR2, 6ID1, 6UP6

## A.6 Signal to Noise Ratio (SNR)

We define SNR as the ratio between the variance of the signal and the variance of the noise. We calculate the variance of the signal over all CTF-applied projection images for a given dataset where

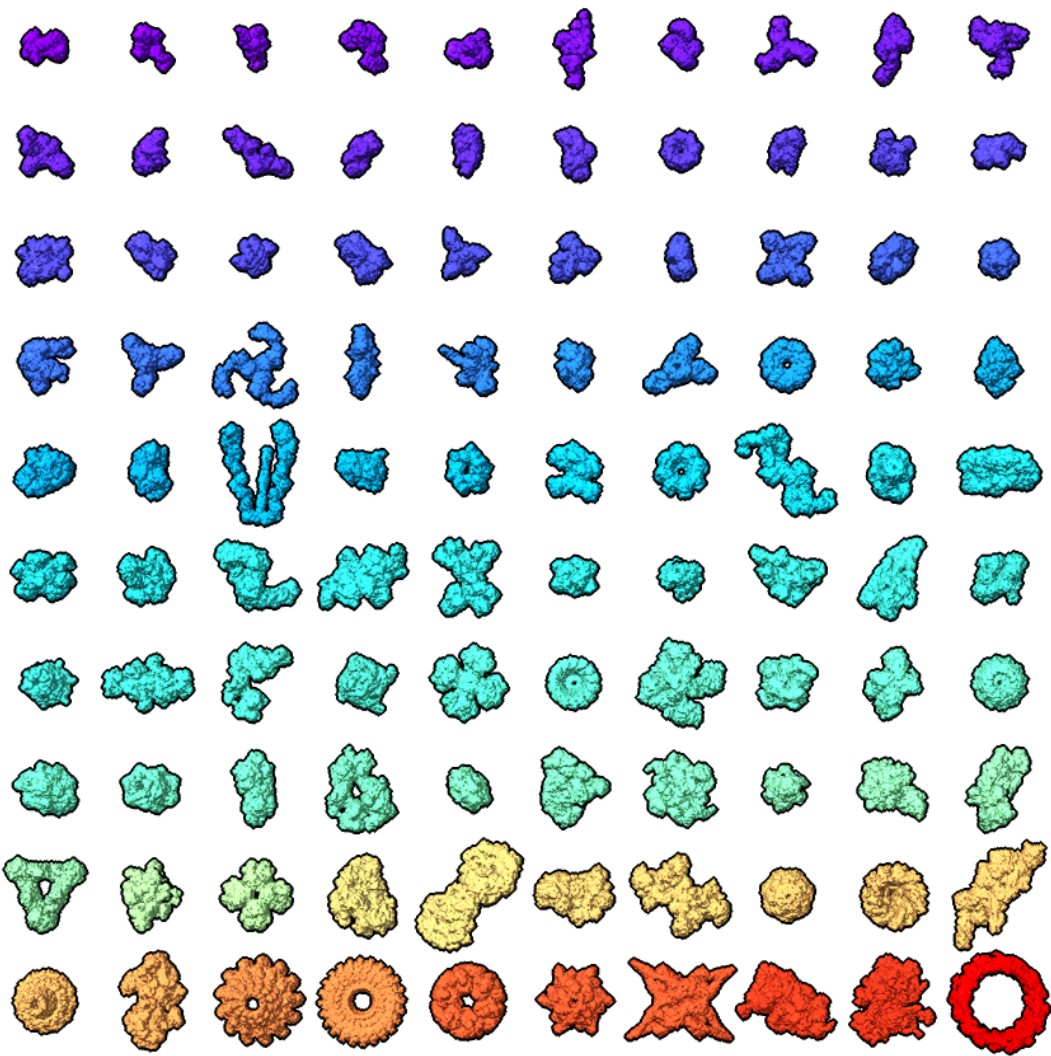

Figure S2: **Tomotwin-100.** All 100 ground truth structures of the `Tomotwin-100` dataset, colored by molecular weight as in Figure 7.

we define the entire $D \times D$ image as the signal. We apply Gaussian white noise to the desired SNR level based on the computed variance. As SNR values can be arbitrarily set based on the definition of the signal, we additionally visualize example cryo-EM images for all datasets in Figure S3.

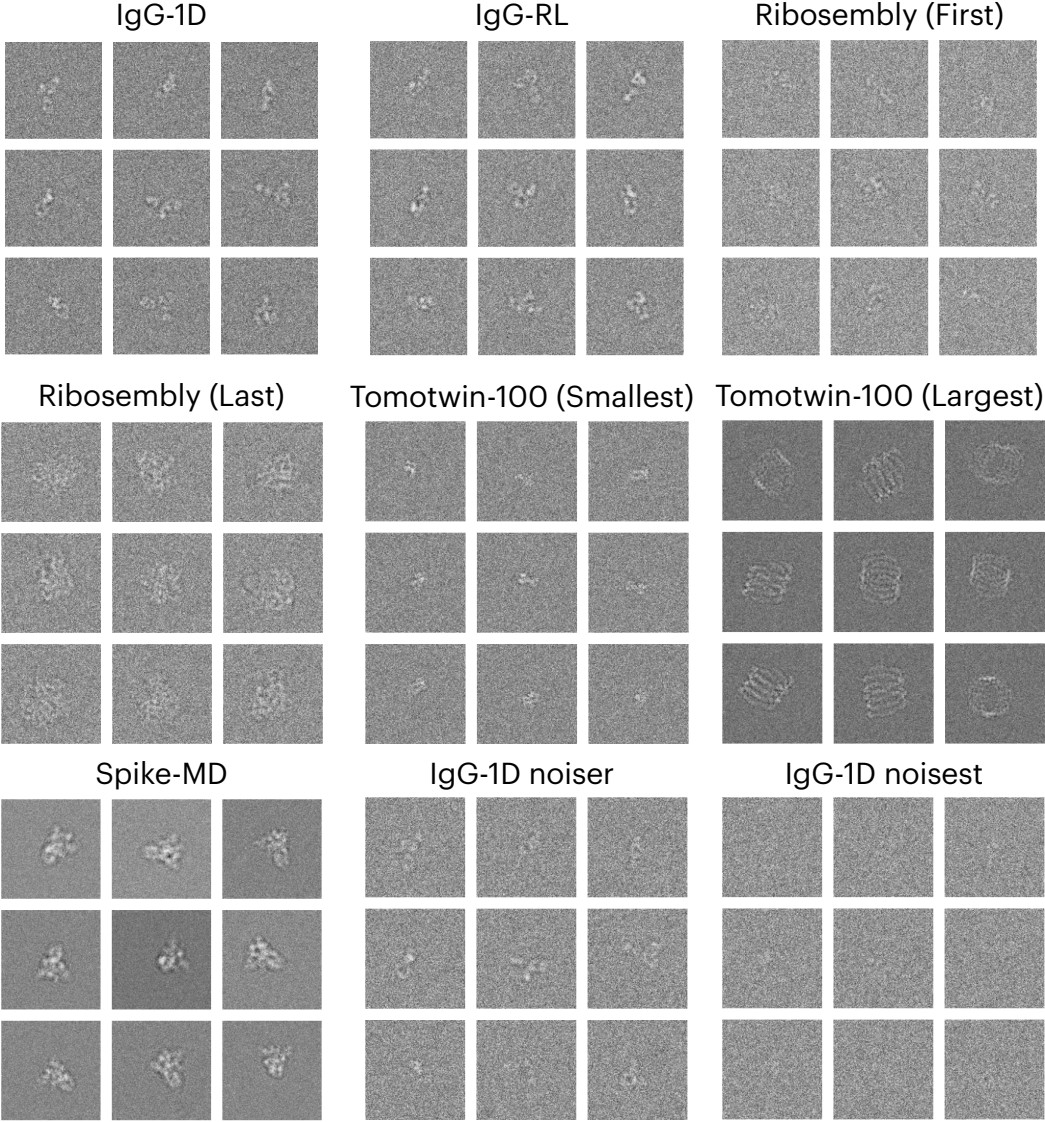

Figure S3: **Example cryo-EM images for all datasets.** Images from the first structure are shown for `IgG-1D` and `IgG-RL`, and images from the first and last structures are shown for `Ribosembly` and `Tomotwin-100`.

# B Experimental Settings

## B.1 Mask Generation

Some reconstruction methods require as input a volume mask defining the region of interest. For mask generation, we first aggregated all ground truth volumes using the `volume add` in ChimeraX. Subsequently, we then applied the `Volume Tools` in CryoSPARC. Specifically, for `IgG-1D`, `IgG-RL`, and `Ribosembly`, the `Dilation radius (pix)` and `Soft padding width (pix)` were set at 8 and 5, respectively. For `Tomotwin-100`, these parameters were adjusted to a `Dilation radius (pix)` of 5 and a `Soft padding width (pix)` of 3. For `Spike-MD`, we take the union of all binarized volumes and use the cryoDRGN `gen_mask` command with a dilation of 25 Å and soft cosine edge of 15 Å. Masks for each dataset are shown in Figure S4b.

## B.2 CryoDRGN, CryoDRGN2

CryoDRGN [7] is a deep generative network-based method where the input images are encoded in the (conformational) latent space and the latent coordinates are decoded into 3D volumes in Fourier domain via an implicit neural representation [6]. In its second version cryoDRGN2 [48], better *ab initio* capabilities were improved with changes to the hierarchical pose search (HPS) algorithm for image pose inference. In our benchmark, we use cryoDRGN for *fixed*, and cryoDRGN2 for *ab initio* purposes.

We trained cryoDRGN and cryoDRGN2 using the official PyTorch implementation[1], version 3.0.0b. We used the default settings with the z-dimension set to 8. For the total number of training epochs, 20 and 30 were used for cryoDRGN and cryoDRGN2, respectively. For Ribosembly, we trained cryoDRGN using 50 epochs. We used one V100 GPU for training.

## B.3 CryoDRGN-AI, CryoDRGN-AI-fixed

CryoDRGN-AI [47] is a deep generative network-based method, inspired by cryoDRGN. CryoDRGN-AI uses both HPS and stochastic gradient descent in pose estimation, while utilizing a differential lookup table instead of an encoder network to encode the pose and conformational latent variable information. We denote the *fixed pose* mode of operation with "CryoDRGN-AI-fixed" and *ab initio* with "CryoDRGN-AI."

We trained CryoDRGN-AI and CryoDRGN-AI-fixed using the official PyTorch implementation[2], version 0.2.2b0. We used the default settings with the z-dimension set to 4 and the total number of training epochs set to 100. We used one A100 GPU for training.

## B.4 Opus-DSD

Opus-DSD [18] is also a deep generative network-based method, built upon cryoDRGN. The network architecture is similar to cryoDRGN except that it uses a 3D Convolutional Neural Network (CNN) and priors for the latent conformational variable.

We trained Opus-DSD using the official PyTorch implementation[3], commit ID `6bc7b86` in GitHub. We used the default settings with the z-dimension set to 12, `valfrac` of 0.25, `downfrac` of 0.75, and `lamb` of 1.0, `bfactor` of 4.0, and `templateres` of 192 as recommended on the official GitHub. For the `Spike-MD` dataset, we use a `downfrac` of 1.00 and `templateres` of 256. The total number of training epochs was set to 20. The volumes reconstructed by Opus-DSD are smaller than the original image dimensions. Consequently, to compute the volume comparison metric (Per-Conformation FSC), we zero-padded the volumes to match the original image dimension. Since the volumes are also misaligned, we aligned them as we did for *ab initio* methods to compute the FSC. We used four A100 GPUs for training.

---

[1] https://github.com/ml-struct-bio/cryodrgn
[2] https://github.com/ml-struct-bio/drgnai
[3] https://github.com/alncat/opusDSD

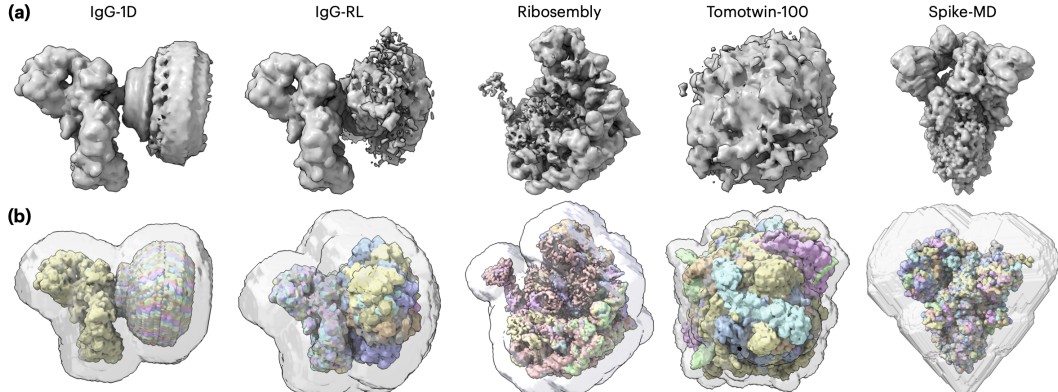

Figure S4: **Consensus volumes and masks.** (a) Consensus volumes for each dataset generated via homogeneous voxel-based backprojection. (b) Mask for each dataset. 10 G.T. volumes are shown within the mask for `Spike-MD`, and all G.T. volumes are shown for other datasets.

## B.5 RECOVAR

RECOVAR [19] is a white-box approach that utilizes principal component analysis (PCA), which is computed through regularized covariance estimation, and an adaptive spatial and fourier domain kernel regression for volume reconstruction.

We trained RECOVAR using the official JAX implementation[4], commit ID `1977aa9` in GitHub. We used the default settings with the z-dimension set to 10 and provided the mask described in Section B.1 as an input. We used one V100 GPU for training.

## B.6 CryoSPARC

We used cryoSPARC[5] version 4.4.0 to train 3DFlex, 3DVA, 3D Classification (fixed, *ab initio*). Some methods in cryoSPARC require a consensus volume, in addition to the masks described in Section B.1. We created the consensus volume for each dataset by backprojecting all images of a given dataset in cryoDRGN [7] with ground truth poses. These volumes are shown in Figure S4a.

**3DFlex.** 3DFlex [9] is a deep learning-based method for heterogeneous reconstruction designed to model *conformational heterogeneity*. 3DFlex models a single canonical 3D volume represented by a tetrahedral mesh and its deformation through training a flow field as a function of conformational latent space coordinates.

To train 3DFlex, the particle stack was normalized such that the mean of each image was 0 and the variance was 1. In the mesh preparation phase (*Flex Mesh Prep* job), we provided the consensus volume and mask as inputs. We adjusted the `Min.rigidity weight` to 1. For training (*3D Flex Training* job), we use all default settings. The latent dimension is 2.

`Spike-MD` required additional modifications to produce reasonable results. A 3DFlex model was trained with consensus poses and volume from *ab initio* reconstruction in cryoSPARC, and the following hyperparameters. The number of latent dimensions was 3, the MLP neural network which models the deformations of the consensus volume had 256 hidden layers, and we trained the model for 32 epochs beyond the standard training time. All other parameters were left to their default values.

**3DVA.** 3DVA [8] is a heterogeneous reconstruction algorithm that is formulated as a probabilistic PCA approach and uses Expectation-Maximization (E-M) to fit a linear subspace model.

We provided the particle images and mask as inputs and set the latent dimension to 3 (default). The `Filter resolution` was set to 5 for `Spike-MD`, 10 for `IgG-1D`, `IgG-RL`, and `Ribosembly`, and 15 for `Tomotwin-100`.

---

[4] https://github.com/ma-gilles/recovar
[5] https://cryosparc.com

| Method | Number of Latent Dimensions |
|---|---|
| CryoDRGN | 8 |
| CryoDRGN-AI-fixed | 4 |
| Opus-DSD | 12 |
| 3DFlex | 2 (3 for MD-Spike) |
| 3DVA | 3 |
| RECOVAR | 10 |
| 3D Class | 10 |
| CryoDRGN2 | 8 |
| CryoDRGN-AI | 4 |
| 3D Class abinit | 20 (16 for Ribosembly, 10 for Tomotwin-100) |

Table S1: Number of latent dimensions modeling heterogeneity for each method

**3D Classification.** 3D Classification is a standard method for analyzing and filtering heterogeneous cryo-EM datasets due to its ease of use and interpretability [4, 56, 57, 46, 51]. This approach models heterogeneity as originating from a discrete mixture model of $K$ independent voxel arrays, where class assignment probabilities are jointly optimized with the molecular volumes via E-M. While use of 3D classification is ubiquitous, the method requires ad hoc, user-driven choices such as the number of classes and initialization for E-M, which leads to complex processing pipelines and often misses conformations, especially when the simple model of heterogeneity is mismatched with the true distribution.

For fixed pose classification in cryoSPARC, we set a `Target resolution` to 3 for `Spike-MD` and 9 for `Tomotwin-100`. We used 20 classes for `Spike-MD` and 10 classes for all other datasets. All other parameters were left at their defaults. For *ab initio* classification, the `Target resolution` was set to 6 for `Spike-MD`. We used 10 classes for `Tomotwin-100`, 16 classes for `Ribosembly`, and 20 classes for `IgG-1D`, `IgG-RL`, and `Spike-MD`. All other parameters were left at their defaults.

The z-dimension, for the purposes of the latent space analysis, was defined as the class posterior, whose length was dataset dependent: 10 (fixed) and 20 (abinit) for `IgG-1D`, `IgG-RL`, and `Ribosembly`, 10 (fixed and abinit) for `Tomotwin-100`, and 20 (fixed and abinit) for `Spike-MD`.

## B.7 Number of Latent Dimensions

An overview of the number of latent dimensions for each method is given in Table S1.

## B.8 Ground Truth Heterogeneity Embeddings

Here we define the ground truth heterogeneity embeddings used to compute Neighborhood Similarity scores and Information Imbalance. The ground truth embedding for each `IgG-1D` structure is a 2D vector of the sine and cosine of the rotation angle. The embedding for each `IgG-RL` conformation is a 3D vector of the centre of mass, and the sine and cosine of the dihedral angle. The `Ribosembly` embeddings are defined in two different ways: *i)* size rank of the atomic models or *ii)* 4096D vector of voxel intensity (real spaced cropped to $156^3$ and downsampled via Fourier cropping to $16^3 = 4096$ voxels). The `Tomotwin-100` embeddings are defined as the size rank of the atomic models. The embeddings for `Spike-MD` are defined as CV1 and CV2 as in Wieczór et al. [43] and Figure 5.

## B.9 Neighborhood Similarity

The percentage of matching neighbors (pMN) (Eq. 2) was calculated using Python with JAX GPU acceleration [58] as a function of the neighborhood radius. All datasets, except for Ribosembly, were divided into five independent sets (Ribosembly was divided into three). The mean pMN and the standard deviation of its mean were computed using these independent sets. The neighborhood radius, expressed as a percentage of the total number of images, was $k = \frac{100\,n}{N_s}$, where $N_s$ the total number of structures in the dataset and $n = 1, \ldots, N_s$. Note that the pMN for $n = 1$ (i.e., $k = \frac{100}{N_s}[\%]$) evaluates how well the embeddings cluster images originating from each structure, effectively measuring structural clustering. In contrast, the pMN for $n > 1$ provides insights into how the connections between ground truth structures relate to the embeddings generated by each method, revealing how images from different structures are interconnected.

## B.10 Information Imbalance

Information imbalance was computed via the implementation in DADApy [59], using a `maxk` (maximum number of neighbors to be considered for the calculation of distances) of the total number of points (335,240 for Ribosembly and 100,000 for the other datasets), and a `subset_size` of 2,000. Error was defined by computing the standard deviation of information imbalances computed with different neighborhood sizes, and here we used $k = 1, 3, 10, 30$ $(0.05, 0.015, 0.5, 1.5\%)$ of neighborhood size ($k = 1, 3, 10$ for Tomotwin-100). Error bars are visible in Tomotwin-100 (Fig. 7d), but smaller than marker size for other datasets.

Small amounts of random noise were applied to average over the 1000-fold duplication of the ground truth heterogeneity embeddings for each image. Additive noise from a uniform distribution, $u \sim U[-\epsilon, \epsilon]$ was added according to Table S2.

The ground truth pose embedding is a 9-dimensional flattened vector of the rotation matrix (translation neglected). The ground truth CTF embedding is a 4-dimensional vector of the two defocus values and the sine and cosine of the angle of astigmatism, normalized by subtracting off the mean and dividing by the standard deviation.

| Dataset | Collective Variable | $\epsilon$ |
|---|---|---|
| IgG-1D | angle in degrees (noise added before sine / cosine transform) | 0.05 |
| IgG-RL | center of mass (Å), angle in degrees (noise added before sine / cosine transform) | 0.1 |
| Ribosembly | voxel intensity | 0.1 |
| Tomotwin-100 | rank size | 0.1 |
| MD | CV1 and CV2: opening and twisting of the RBD; cf. Fig 5 | 0.1 |

Table S2: Random noise with distribution $U[-\epsilon, \epsilon]$ added to the ground truth heterogeneity embeddings.

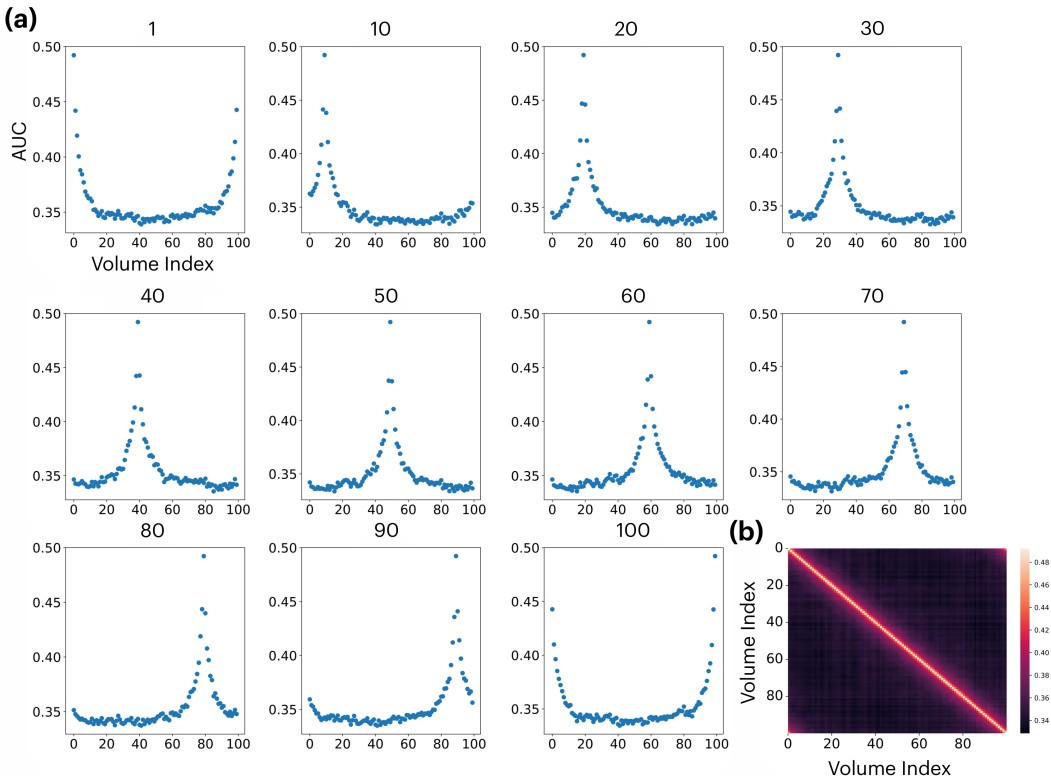

Figure S5: **Metric validation.** (a) AUC-FSC between one G.T and all 100 G.T.s of the `IgG-1D` dataset. Each plot corresponds to the reference G.T volume, indicated by the number above the plot. (b) Heatmap comparing all 100 G.Ts against all 100 G.Ts.

## C  Supplementary Results

### C.1  Volume Metrics

#### C.1.1  Fourier Shell Correlation

We use metrics based on the Fourier Shell Correlation (FSC) curve to quantify how well the distribution of reconstruction volumes compares to the ground truth set of conformations.

The Fourier Shell Correlation (FSC) curve is a standard method for comparing two volumes in cryo-EM. For two volumes $\hat{U}$ and $\hat{V}$, the FSC curve measures the correlation between $\hat{U}$ and $\hat{V}$ as a function of spherically-averaged radial shells in the Fourier domain:

$$FSC(k) = \frac{\sum_{s \in S_k} \hat{U}_s \hat{V}_s^*}{\sqrt{(\sum_{s \in S_k} |\hat{U}_s|^2)(\sum_{s \in S_k} |\hat{V}_s|^2)}} \qquad (4)$$

where $\hat{V}^*$ is the complex conjugate of $\hat{V}$, and $S_k$ represents the set of Fourier voxels in a spherical shell at a distance $k$ from the origin. We note that FSC is typically used in a two-fold cross validation approach, where independent reconstructions of random halves of the dataset are compared. The *resolution* of the final volume is often reported as $1/k_0$ where $k_0 = \mathrm{argmax}_k \, FSC(k) < C$ and $C$ is some fixed threshold ($C = 0.143$ for the gold standard FSC). This "half-map" FSC metric has sufficed in the absence of ground truth, however, as it is only a consistency check, it is not sensitive to bias in the model.

Here, we use FSC to compare reconstructed volumes against ground truth volumes, and report the area under the FSC curve as a summary statistic (max 0.5; higher is better).

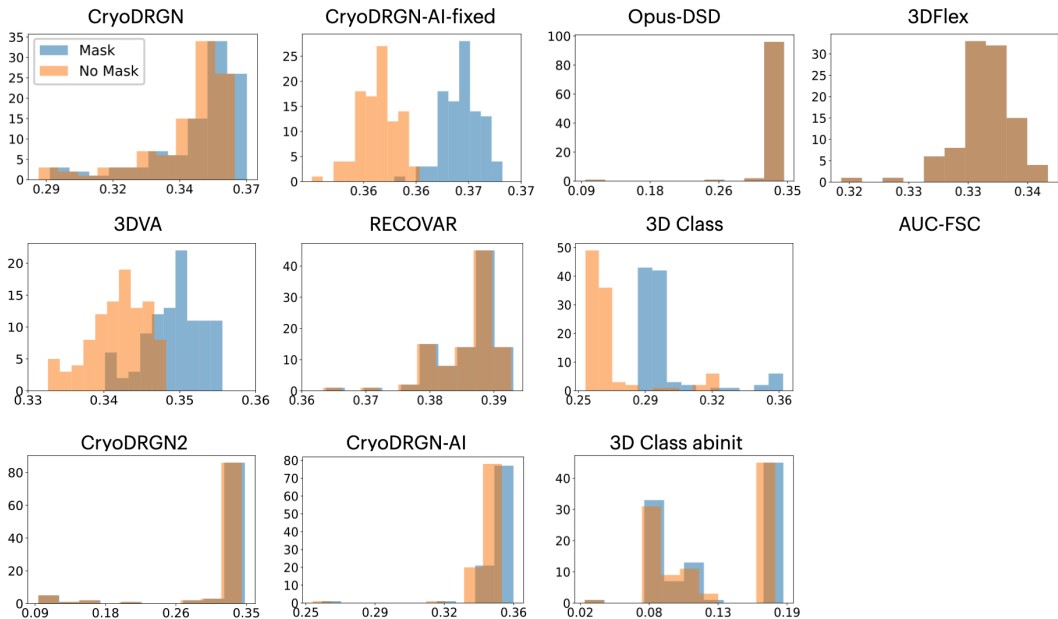

Figure S6: **Unmasked vs. masked FSC calculations for IgG-1D.** Histogram comparing *Per-Conformation FSC* for each method, with (blue) and without (orange) applying a mask around the particle before computing the FSC.

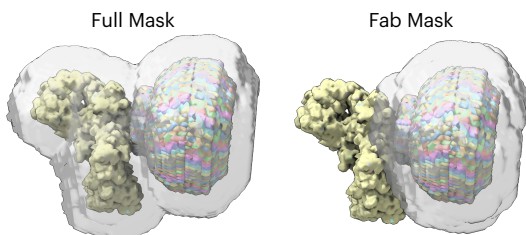

Figure S7: **Full vs Fab Mask.** Different masks with all IgG-1D 100 G.T volumes. The full mask covers all volumes, while the Fab mask covers only the Fab region.

### C.1.2 Sensitivity to Structural Heterogeneity

We first validate that our volume comparison metric, the area under the Fourier Shell Correlative curve (FSC$_{AUC}$), is sensitive to structural differences *between ground truth volumes*. In Figure S5, we compute FSC$_{AUC}$ between different ground truth volumes for the IgG-1D dataset. The FSC-AUC between a given volume and all other volumes reaches its highest point (0.5) at the ground truth index and is unimodal across the ground truth conformational coordinate (note that IgG-1D describes a circular 1D motion and thus the x-axis is cyclic) (Fig. S5a). Figure S5b summarizes the all-to-all comparison.

### C.1.3 Unmasked vs. Masked FSC

Following standard convention in cryo-EM, we apply a mask around the particle to zero out low-isosurface background volume when computing FSC metrics to avoid spurious correlations with solvent noise. Here, we provide an analysis comparing the use of a mask versus no mask in our *Per-Conformation FSC* metric (Fig. S6). The generation of the masks is described in Section B.1. Masking out background noise generally enhances performance when computing volume metrics.

| Method | CryoDRGN | CryoDRGN-AI-fixed | Opus-DSD | 3DFlex | 3DVA | RECOVAR | 3D Class |
|---|---|---|---|---|---|---|---|
| Full (Mean) | 0.366 (0.003) | 0.366 (0.001) | 0.34 (0.006) | 0.336 (0.002) | 0.351 (0.003) | 0.391 (0.001) | 0.297 (0.019) |
| Full (Median) | 0.366 | 0.366 | 0.341 | 0.336 | 0.351 | 0.391 | 0.291 |
| Fab Only (Mean) | 0.349 (0.005) | 0.349 (0.002) | 0.335 (0.009) | 0.292 (0.003) | 0.321 (0.005) | 0.385 (0.002) | 0.238 (0.032) |
| Fab Only (Median) | 0.349 | 0.349 | 0.336 | 0.189 | 0.32 | 0.385 | 0.228 |

Table S3: **(IgG-1D) Full vs Fab mask for fixed pose methods.** *Per-Conformation FSC* for both the full and Fab masks.

| Method | CryoDRGN2 | CryoDRGN-AI | 3D Class abinit |
|---|---|---|---|
| Full (Mean) | 0.344 (0.003) | 0.356 (0.002) | 0.13 (0.046) |
| Full (Median) | 0.344 | 0.356 | 0.119 |
| Fab Only (Mean) | 0.328 (0.003) | 0.339 (0.003) | 0.114 (0.031) |
| Fab Only (Median) | 0.329 | 0.339 | 0.122 |

Table S4: **(IgG-1D) Full vs Fab mask for *ab initio* methods.** *Per-Conformation FSC* for both the full and Fab masks.

### C.1.4  Full vs. Fab Mask

We also investigate the effect of applying a mask only on the moving Fab region when computing FSC-AUC values for IgG-1D. Table S3 and S4 shows the *Per-Conformation FSC* of IgG-1D dataset for both the full and Fab masks. The overall performance decreases, and although the performance ranking between 3DVA and Opus-DSD changes, the others remain the same. This indicates that the performance of 3DVA is largely stems from non-Fab regions. Figure S7 illustrates each mask with all ground truth volumes. The full mask was used unless otherwise mentioned.

### C.2  Volume FSC metrics for heterogeneous reconstruction

To evaluate the quality of reconstructed volumes from a heterogeneous reconstruction, we sample a set of representative structures for each method and compare them against the set of ground truth volumes. We provide three different approaches for sampling volumes from the latent representation: *Per-Conformation FSC*, *Sample FSC*, and *Per-Image FSC*.

**Per-Conformation FSC.**  For each ground truth conformation $V$, we compute a corresponding latent coordinate by averaging all latent embeddings $\bar{z}$ of images generated by $V$. To remain on the data manifold, we identify the closest latent coordinate $z^*$ to $\bar{z}$. Using this coordinate, we reconstruct the associated volume and evaluate its FSC$_{\text{AUC}}$ against the ground truth volume. Figure S8 and S9 provides *Per-Conformation FSC* curves for each method across all datasets and Table S5 shows the FSC$_{\text{AUC}}$ values. We additionally show all 100 FSC curves for the `IgG-1D` dataset for each method in Figure S10.

**Sample FSC.** While *Per-Image FSC* is the main metric we report, it may leverage information from the ground truth clustering when computing representative latents/volumes. As an alternative, we also generate volumes after unsupervised clustering of the latent embeddings to yield representative samples (without any use of G.T. information). However, as there is then no obvious correspondence between the sampled volume and its G.T. conformation, we compute the FSC$_{\text{AUC}}$ for each sampled volume against all G.T. volumes and take the max value. We call this quantification the *Sample FSC* and Table S5 shows the FSC$_{\text{AUC}}$ values.

Specifically, we use $k$-means clustering to cluster the latent embeddings for each method, and use the $k$ centroids as representative for evaluation. We choose $k$=20 for `IgG-1D`, `IgG-RL`, and `Tomotwin-100`, and $k$=16 for `Ribosembly`.We show *Sample FSC* curves for each method across the four datasets in Figure S11.

**Per-Image FSC.** Finally, we use *Per-Image FSC* as a metric that jointly assesses conformation estimation and reconstruction quality. Here, we use a set of $N$ images from the dataset (either sampled at random or uniformly for each conformation). For each image, we reconstruct an associated volume at its inferred latent coordinate. We then compute the FSC AUC for the reconstructed volume to the image's ground truth volume. Thus, unlike *Per-Conformation FSC* and *Sample FSC*, *Per-Image FSC* assesses how well each method has reconstructed the *distribution of volumes*. We use 100 images for each dataset to compute *Per-Image FSC* with one random image per ground truth conformation.

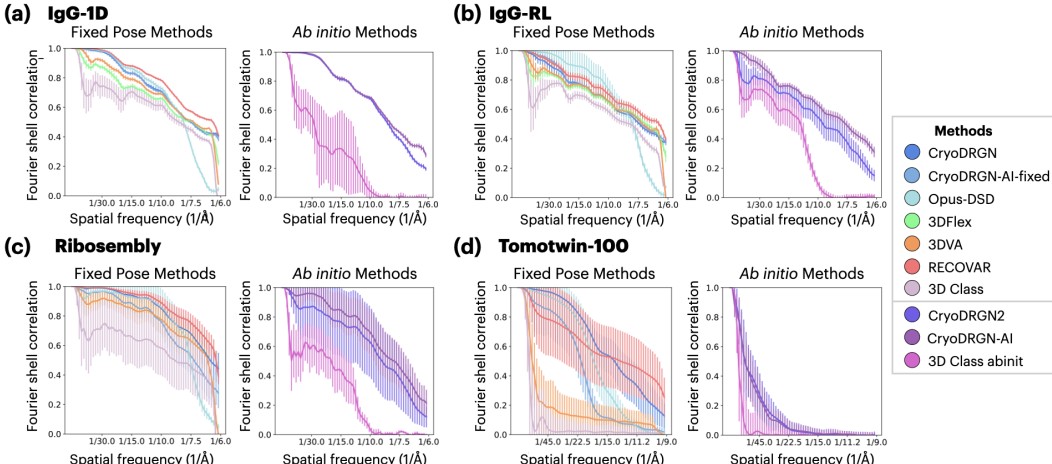

Figure S8: **Per-Conformation FSC.** Each curve shows the average FSC curve across all conformations with error bars indicating the standard deviation.

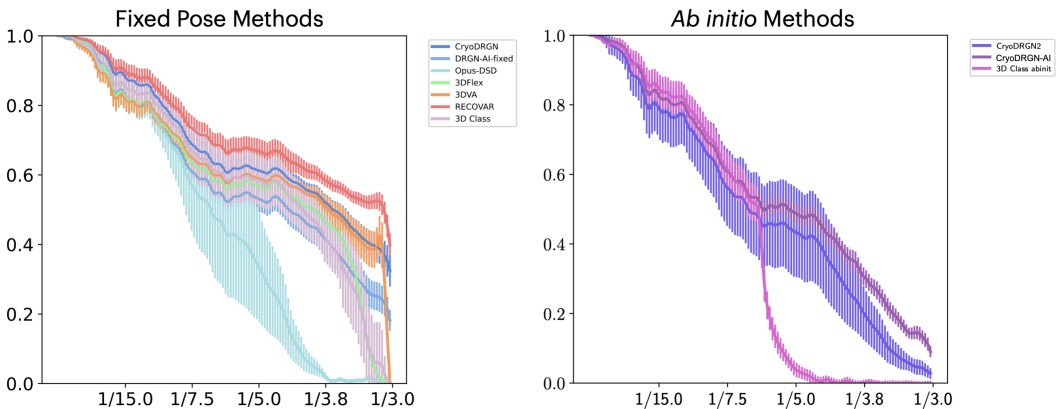

Figure S9: **Per-Conformation FSC for Spike-MD.** Each curve shows the average FSC curve across all conformations with error bars indicating the standard deviation.

We present FSC curves averaged across all ground truth conformations for all methods for `IgG-1D`, `IgG-RL`, `Ribosembly`, and `Tomotwin-100` in Figures 2, 4, 6, 7, respectively. FSC curves for `Spike-MD` are shown in Figure S12.

### C.3 Additional Noise Comparison Results

Figure S13 shows extended results on `IgG-1D-noisier` and `IgG-1D-noisiest` (SNR 0.005, 0.001) as compared to the `IgG-1D` dataset. With increasing noise levels, there is a noticeable reduction in volume metrics, and the capability to differentiate between different conformations decreases.

### C.4 Additional Qualitative Results

For additional qualitative evaluation, we provide visualizations of the reconstructed volumes and latent spaces for each method and dataset in CryoBench. Figure S14 shows the latent spaces colored by dihedral angle (CV1) and center of mass distance (CV2) for IgG-RL dataset. Figures S15-S21 display volumes from $k$-means cluster centers, with points in the UMAP visualization of the latent space corresponding to each center.

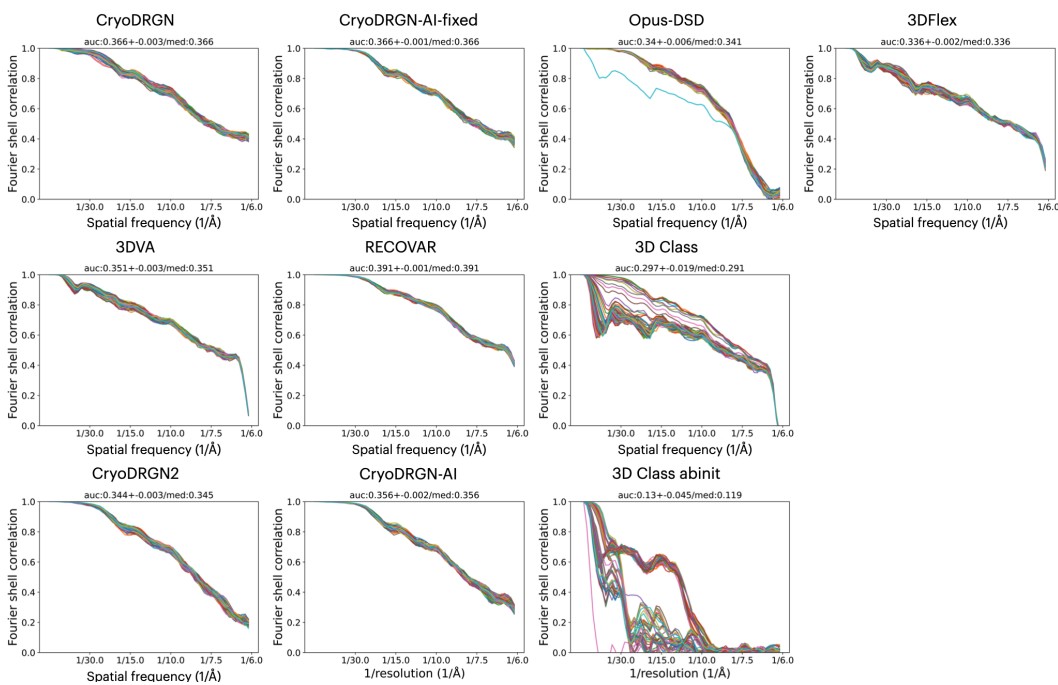

Figure S10: **Per-Conformation FSC curves.** All 100 FSC curves for the `IgG-1D` dataset. Masks were applied when computing FSCs.

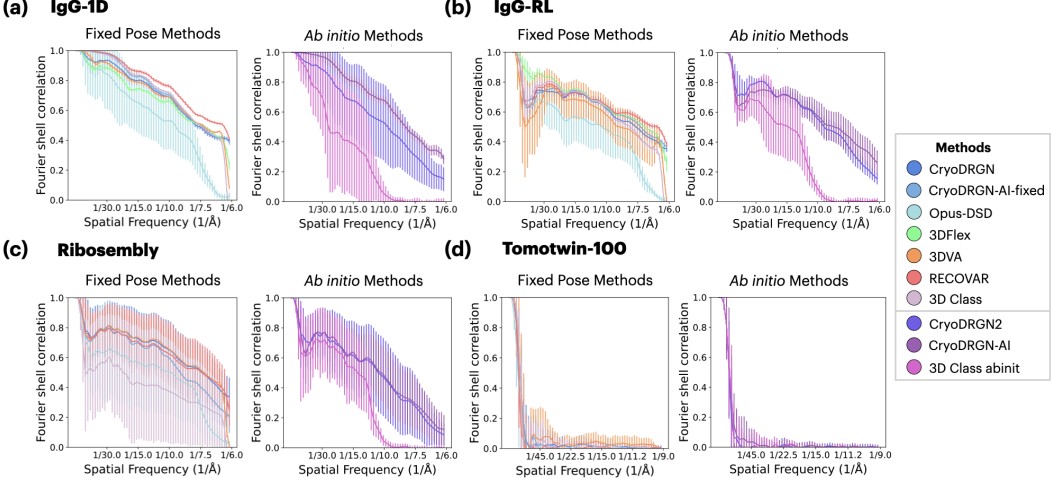

Figure S11: **Sample FSC.** Each curve shows the average FSC curve across all conformations with error bars indicating the standard deviation.

## C.5   Additional Information Imbalance Results

**CTF and Pose:** Information imbalance with respect to the ground truth latent pose (rotation only, not translation) and CTF parameters is generally in the orthogonal region (1,1) for all methods (Figs. S22, S23). However, zooming in, for pose, CryoDRGN and Opus-DSD are off the shared information x=y line, indicating their minor entanglement is more pronounced than other methods. For CTF, the trends are less clear, but Opus-DSD and 3D Class abinit are generally the furthest away from the orthogonal region.

| Method | IgG-1D | | IgG-RL | | Ribosombly | | Tomotwin-100 | |
|---|---|---|---|---|---|---|---|---|
| | Mean (std) | Med | Mean (std) | Med | Mean (std) | Med | Mean (std) | Med |
| CryoDRGN | 0.356 (0.016) | 0.364 | 0.346 (0.011) | 0.345 | 0.419 (0.019) | 0.4 | 0.277 (0.064) | 0.3 |
| CryoDRGN-AI-fixed | 0.367 (0.001) | 0.367 | 0.356 (0.007) | 0.355 | 0.385 (0.029) | 0.4 | 0.183 (0.027) | 0.184 |
| Opus-DSD | 0.292 (0.062) | 0.311 | 0.291 (0.069) | 0.313 | 0.321 (0.128) | 0.378 | 0.177 (0.036) | 0.186 |
| 3DFlex | 0.341 (0.001) | 0.341 | 0.348 (0.005) | 0.348 | - | - | - | - |
| 3DVA | 0.354 (0.002) | 0.354 | 0.319 (0.051) | 0.344 | 0.407 (0.019) | 0.413 | 0.176 (0.051) | 0.168 |
| RECOVAR | **0.391 (0.001)** | **0.391** | **0.371 (0.007)** | **0.369** | **0.433 (0.013)** | **0.433** | **0.331 (0.094)** | **0.385** |
| 3D Class | 0.363 (0.001) | 0.363 | 0.338 (0.006) | 0.335 | 0.303 (0.148) | 0.413 | 0.221 (0.032) | 0.221 |
| CryoDRGN2 | 0.294 (0.089) | 0.342 | 0.312 (0.02) | 0.318 | 0.312 (0.088) | **0.356** | **0.081 (0.013)** | **0.079** |
| CryoDRGN-AI | **0.352 (0.014)** | **0.355** | **0.324 (0.036)** | **0.336** | **0.319 (0.069)** | **0.32** | 0.071 (0.01) | 0.072 |
| 3D Class abinit | 0.161 (0.072) | 0.118 | 0.204 (0.054) | 0.225 | 0.24 (0.014) | 0.244 | 0.068 (0.011) | 0.065 |

Table S5: **AUC of Sample FSC.** The metrics are computed after masking out background noise. Standard deviation of all FSC-AUCs per method is given in parentheses.

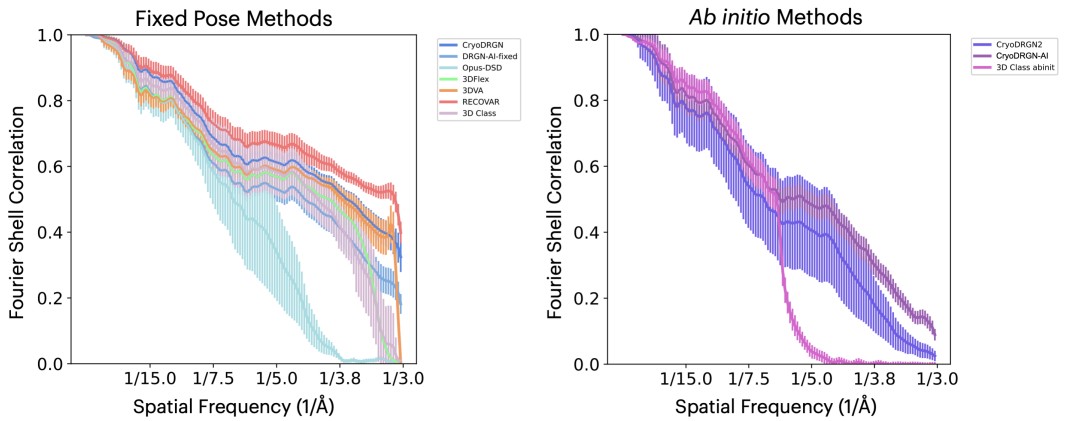

Figure S12: **Per-Image FSC for Spike-MD.** Each curve shows the average FSC curve across 100 sampled images with error bars indicating the standard deviation.

## C.6 Additional `Spike-MD` embedding analysis

Neighborhood Similarity and Information Imbalance plots for `Spike-MD` are shown in Figure S24. We observe a relatively low similarity in neighborhoods between the embeddings and the ground truth molecular dynamics collective variables for small neighbhoorhood radii, consistent with qualitative observations from UMAP visualizations (Fig. 5). Information imbalance of the Spike-MD dataset (Fig. S24-right) shows 3DVA on the shared information line at (0.5,0.5) - a very similar result as in IgG-1D. Opus-DSD and CryoDRGN2 are near (0.9,0.6), the closest to the orthogonal region for the Spike-MD dataset compared with other methods. For Opus-DSD, this is the closest to the orthogonal region compared with its information imbalance on the other datasets. For CryoDRGN2, this is a similar value as the challenging datasets (IgG-RL and Tomotwin-100). The other methods employed in these experiments (CryoDRGN, CryoDRGN-AI-fixed, 3DFlex, RECOVAR, CryoDRGN-AI) are closer to the equivalent zone and cluster together near (0.5,0.2).

## C.7 Pose Error

We compute the pose error for *ab initio* methods (Table S7). CryoDRGN-AI generally exhibits the lowest pose error. The distribution of pose errors are plotted in Figure S25-S31.

| Method | IgG-1D | | IgG-RL | | Ribosembly | | Tomotwin-100 | | Spike-MD | |
|---|---|---|---|---|---|---|---|---|---|---|
| | Mean (std) | Med | Mean (std) | Med | Mean (std) | Med | Mean (std) | Med | Mean (std) | Med |
| CryoDRGN | 0.366 (0.003) | 0.366 | 0.349 (0.008) | 0.348 | 0.415 (0.019) | 0.415 | **0.321 (0.034)** | 0.322 | 0.340 (0.009) | 0.340 |
| CryoDRGN-AI-fixed | 0.366 (0.001) | 0.366 | 0.355 (0.007) | 0.354 | 0.372 (0.032) | 0.374 | 0.206 (0.041) | 0.212 | 0.301 (0.012) | 0.304 |
| Opus-DSD | 0.34 (0.006) | 0.341 | 0.346 (0.029) | 0.349 | 0.372 (0.046) | 0.382 | 0.256 (0.038) | 0.266 | 0.228 (0.030) | 0.242 |
| 3DFlex | 0.336 (0.002) | 0.336 | 0.339 (0.007) | 0.339 | - | - | - | - | 0.304 (0.010) | 0.306 |
| 3DVA | 0.351 (0.003) | 0.351 | 0.341 (0.006) | 0.341 | 0.375 (0.038) | 0.372 | 0.095 (0.04) | 0.083 | 0.324 (0.010) | 0.325 |
| RECOVAR | **0.391 (0.001)** | **0.391** | **0.372 (0.008)** | **0.371** | **0.43 (0.016)** | **0.432** | 0.306 (0.093) | **0.33** | **0.363 (0.011)** | **0.366** |
| 3D Class | 0.297 (0.019) | 0.291 | 0.309 (0.01) | 0.307 | 0.289 (0.081) | 0.288 | 0.046 (0.026) | 0.037 | 0.307 (0.023) | 0.308 |
| CryoDRGN2 | 0.344 (0.003) | 0.345 | 0.294 (0.022) | 0.296 | 0.314 (0.09) | 0.348 | **0.076 (0.016)** | 0.074 | 0.253 (0.037) | 0.265 |
| CryoDRGN-AI | **0.356 (0.002)** | **0.356** | **0.341 (0.01)** | 0.342 | **0.351 (0.052)** | **0.369** | 0.073 (0.014) | 0.073 | **0.281 (0.009)** | **0.282** |
| 3D Class abinit | 0.13 (0.046) | 0.119 | 0.184 (0.022) | 0.188 | 0.144 (0.036) | 0.138 | 0.032 (0.012) | 0.032 | 0.206 (0.010) | 0.207 |

Table S6: **AUC of Per-Conformation-FSC.** The metrics are computed after masking out background noise. Standard deviation of all FSC-AUCs per method is given in parentheses.

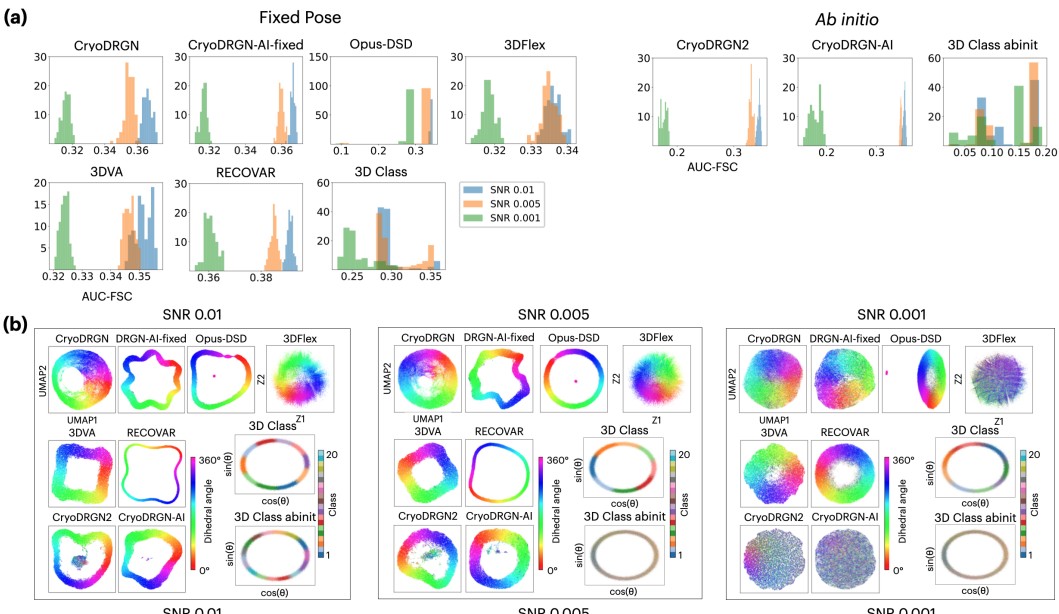

Figure S13: **IgG-1D with noise. (a)** Histogram of *Per-Conformation FSC* for each method at SNR levels of 0.01, 0.005, 0.001. **(b)** UMAP visualizations colored by G.T. dihedral conformations of each method.

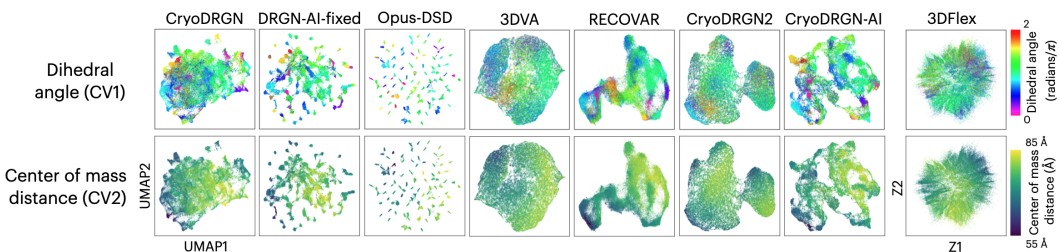

Figure S14: **IgG-RL latent spaces colored by collective variables (CVs) describing the IgG-RL conformations.**

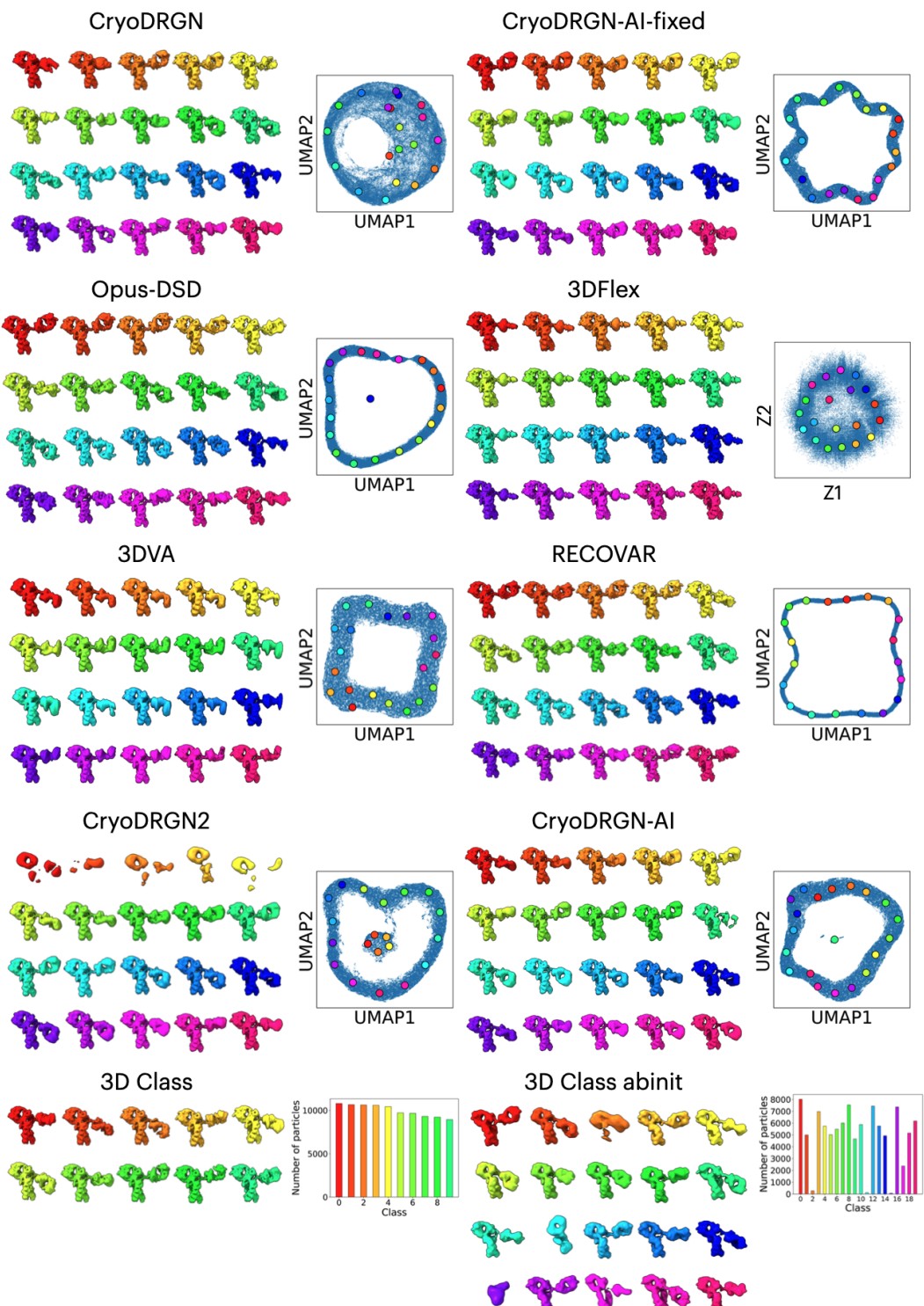

Figure S15: **Qualitative Results (IgG-1D).** Representative volumes and UMAP visualization of the latent embeddings for each method. Volumes correspond to $k$-means cluster centers with $k$=20 and are colored according to the associated point. Class volumes and particle counts are shown for 3D Classification.

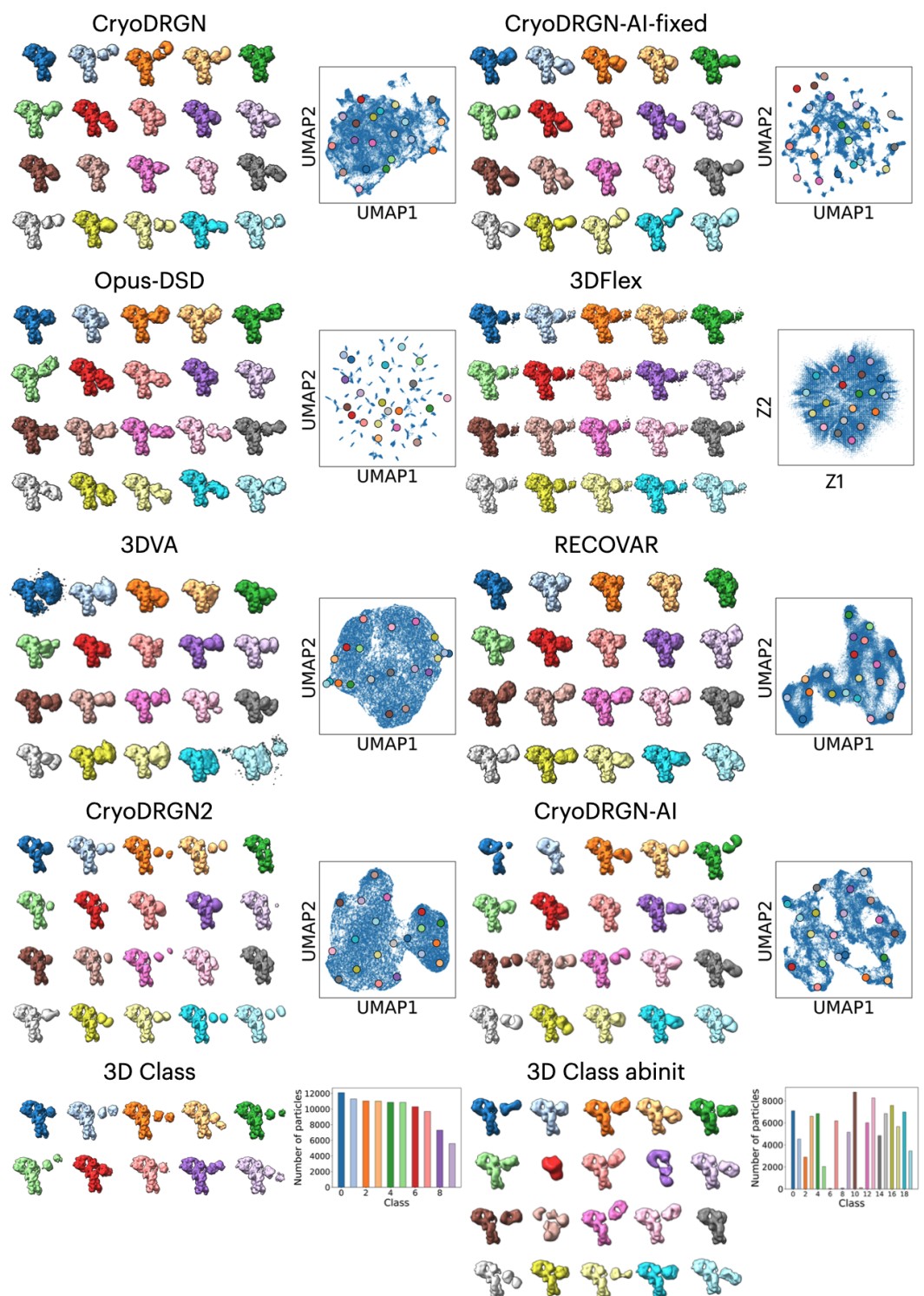

Figure S16: **Qualitative Results (IgG-RL).** Representative volumes and UMAP visualization of the latent embeddings for each method. Volumes correspond to $k$-means cluster centers with $k$=20 and are colored according to the associated point. Class volumes and particle counts are shown for 3D Classification.

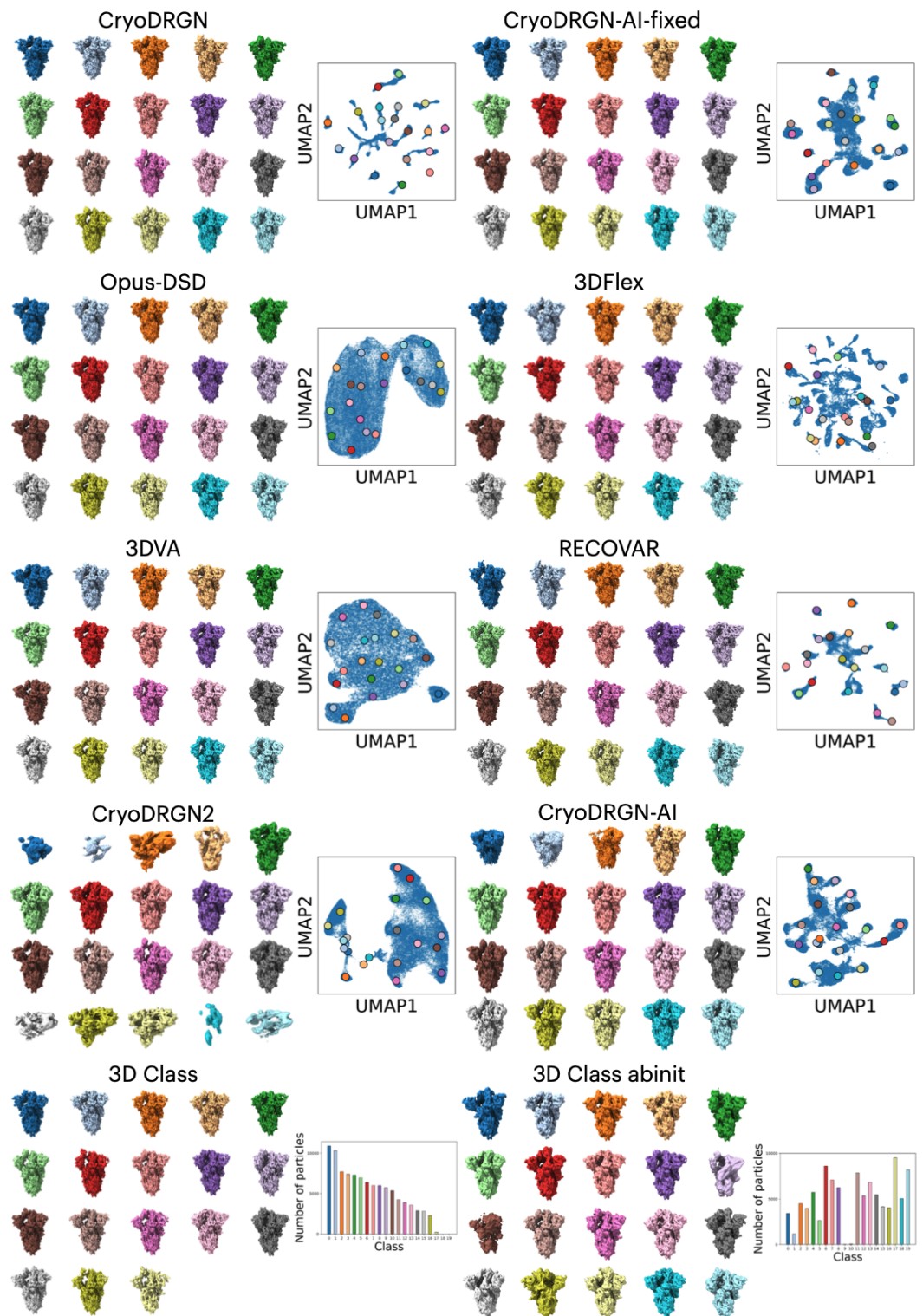

Figure S17: **Qualitative Results (Spike-MD).** Representative volumes and UMAP visualization of the latent embeddings for each method. Volumes correspond to $k$-means cluster centers with $k$=20 and are colored according to the associated point. Class volumes and particle counts are shown for 3D Classification.

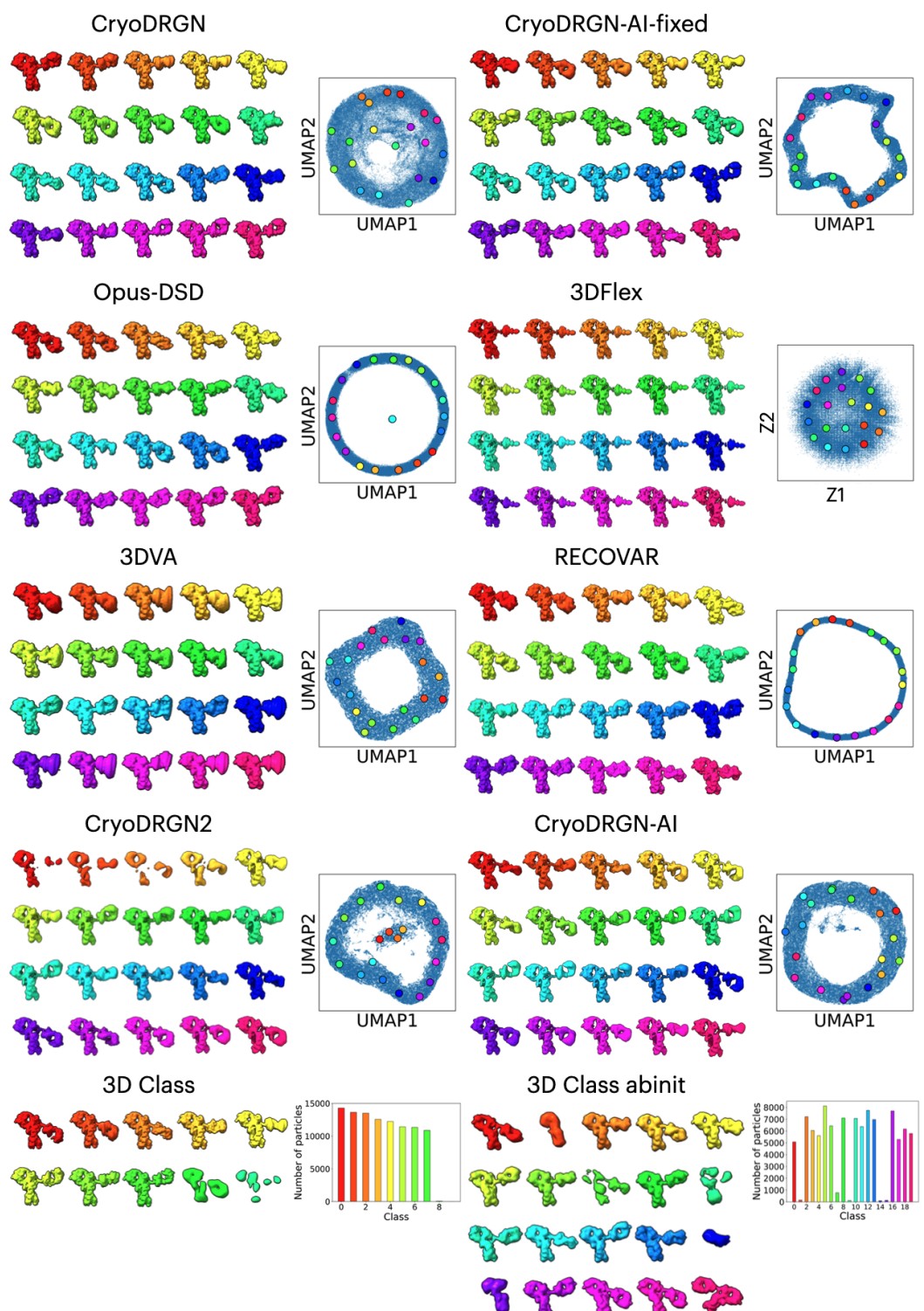

Figure S18: **Qualitative Results (IgG-1D noisier).** Representative volumes and UMAP visualization of the latent embeddings for each method. Volumes correspond to $k$-means cluster centers with $k$=20 and are colored according to the associated point. Class volumes and particle counts are shown for 3D Classification.

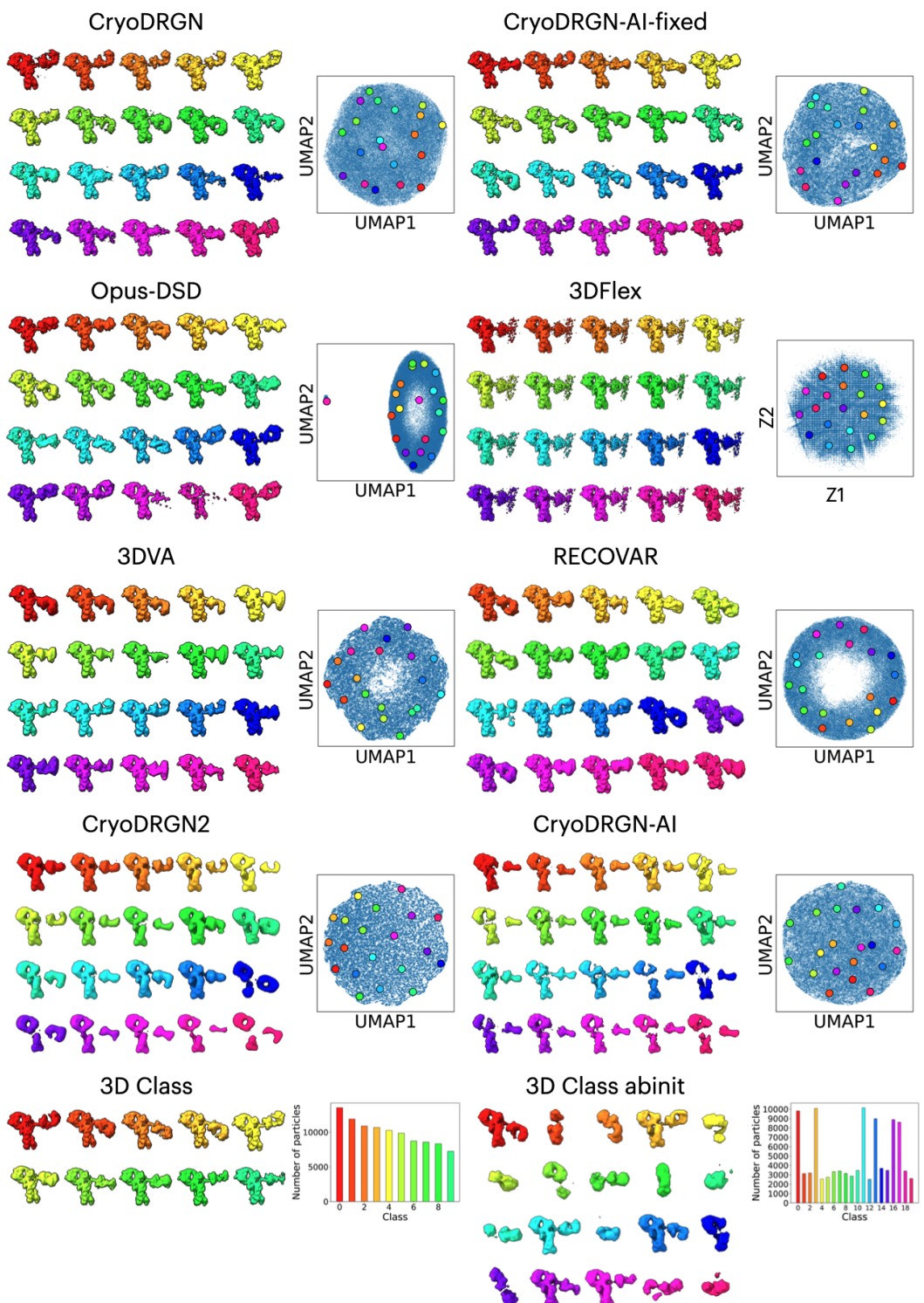

Figure S19: **Qualitative Results (IgG-1D noisiest).** Representative volumes and UMAP visualization of the latent embeddings for each method. Volumes correspond to $k$-means cluster centers with $k$=20 and are colored according to the associated point. Class volumes and particle counts are shown for 3D Classification.

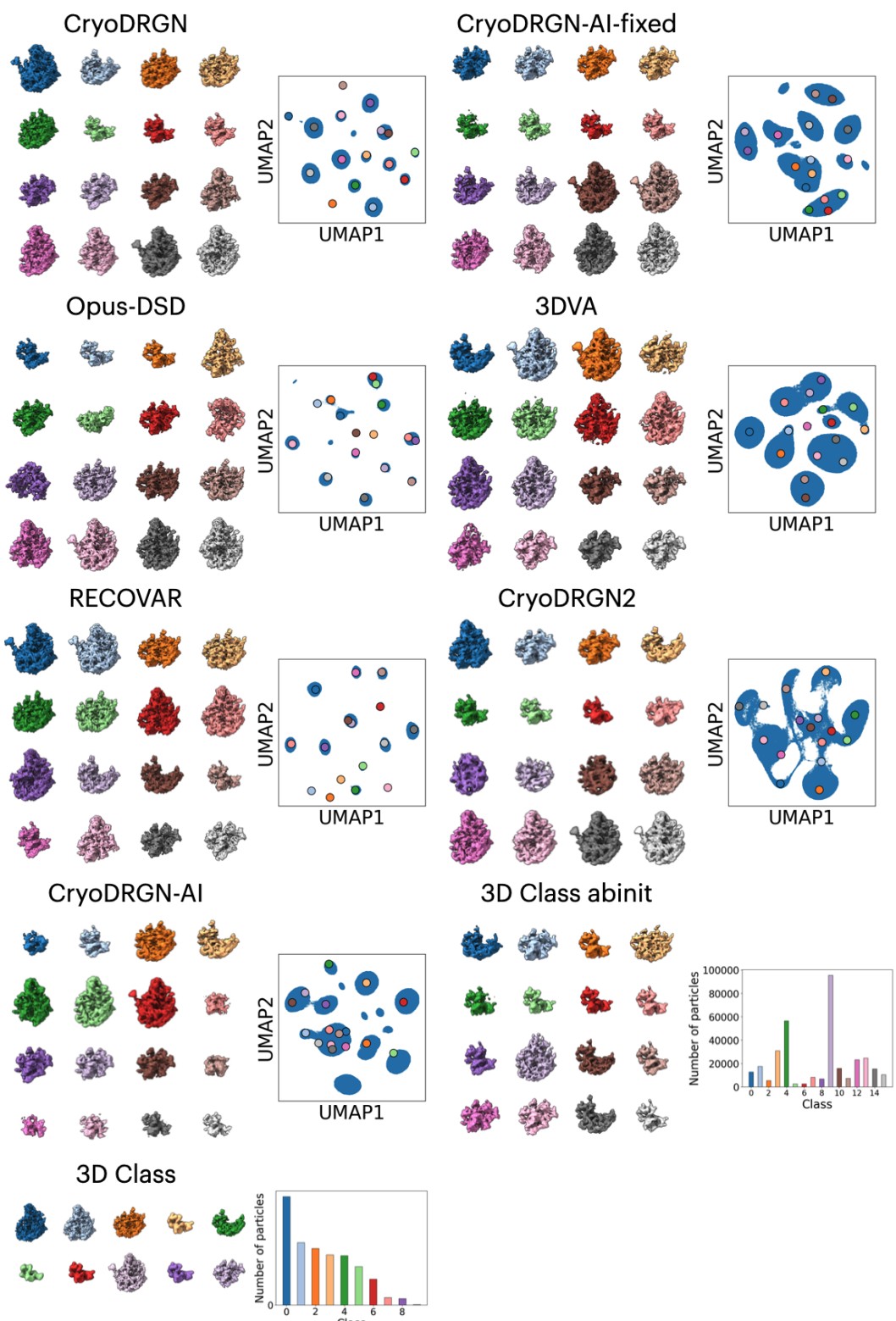

Figure S20: **Qualitative Results (Ribosembly).** Representative volumes and UMAP visualization of the latent embeddings for each method. Volumes correspond to $k$-means cluster centers with $k$=16 and are colored according to the associated point. Class volumes and particle counts are shown for 3D Classification.

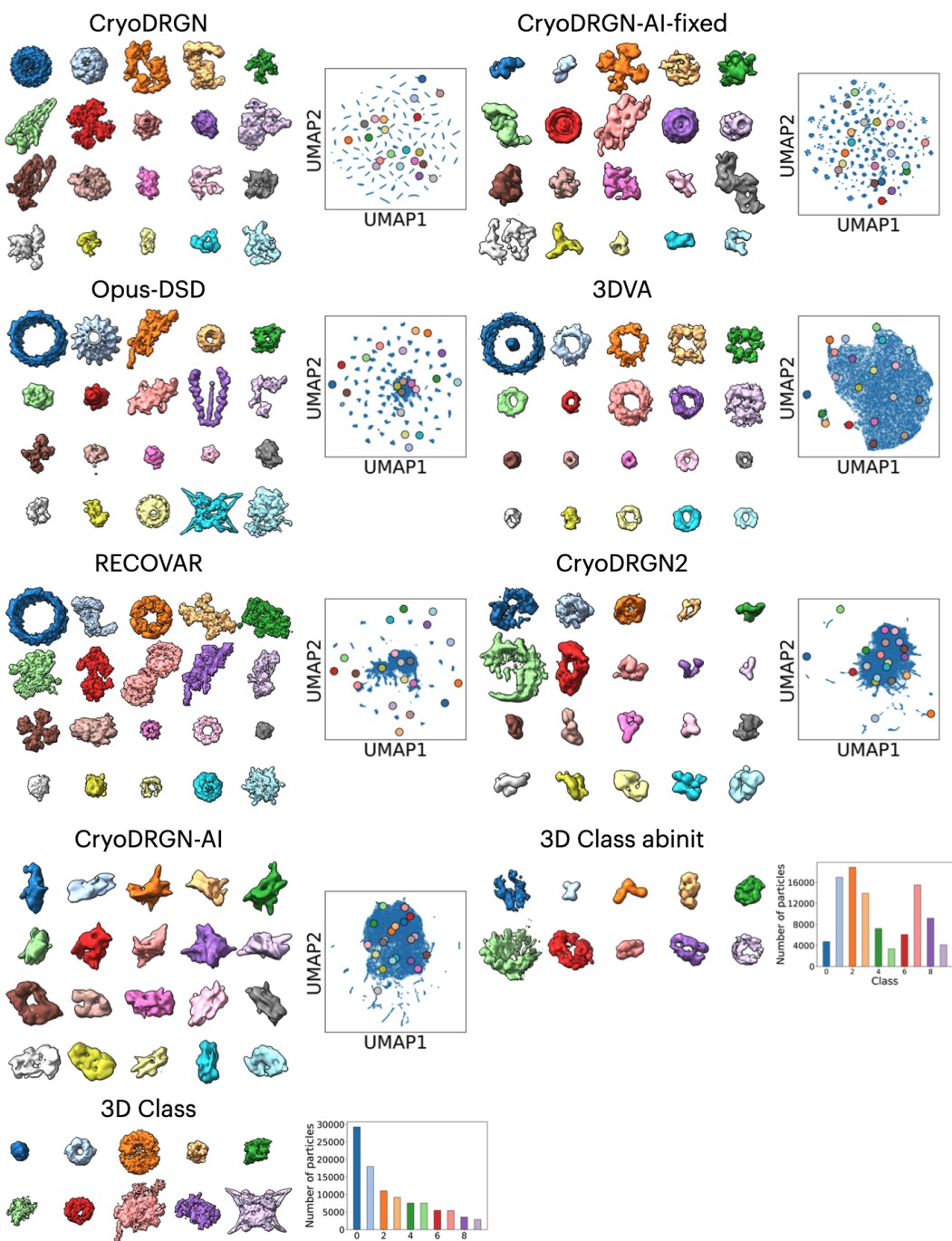

Figure S21: **Qualitative Results (Tomotwin-100).** Representative volumes and UMAP visualization of the latent embeddings for each method. Volumes correspond to $k$-means cluster centers with $k$=20 and are colored according to the associated point. Class volumes and particle counts are shown for 3D Classification.

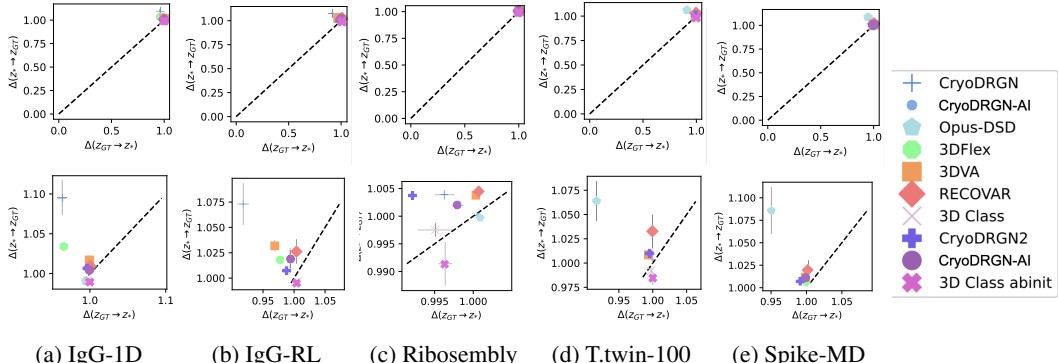

Figure S22: **Pose Information Imbalance**. In full view ($[0, 1]^2$; top row) and zoomed in (bottom row).

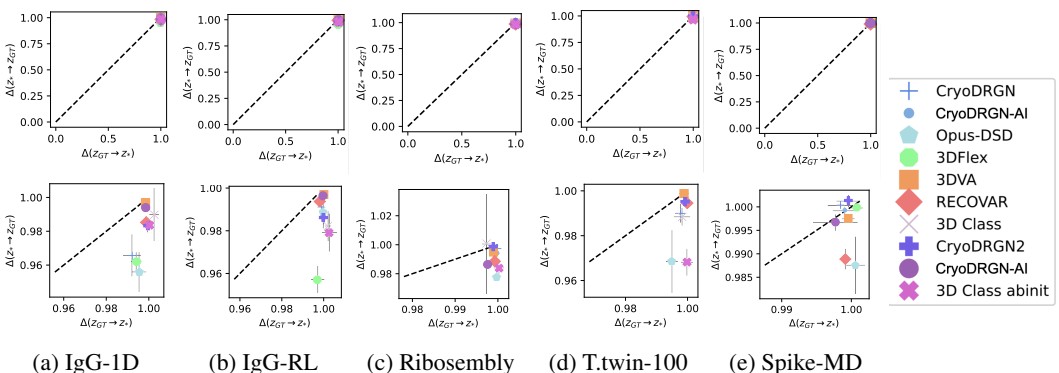

Figure S23: **CTF Information Imbalance**. In full view ($[0, 1]^2$; top row) and zoomed in (bottom row).

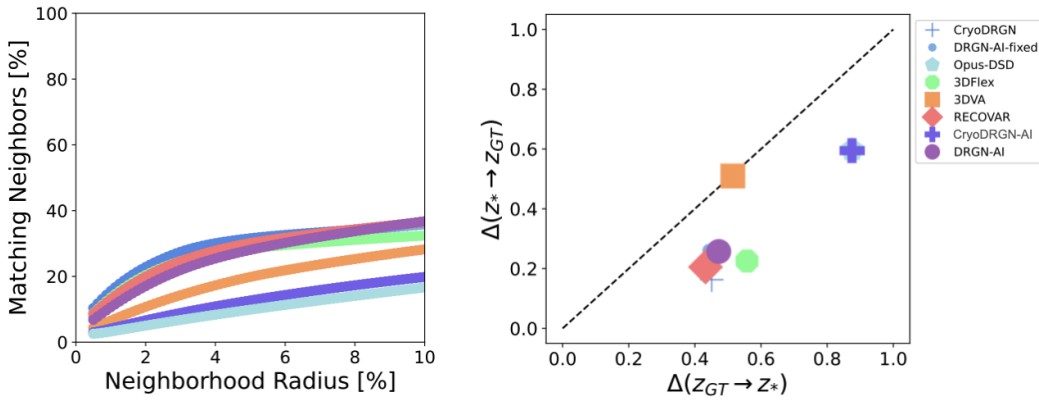

Figure S24: **Embedding metric results for the Spike-MD dataset** (left) Neighborhood similarity as a function of the neighborhood radius [%]. (right) Information Imbalance. Opus-DSD (not visible) is underneath cryoDRGN2. DRGN-AI fixed is underneath RECOVAR and DRGN-AI.

| Method | IgG-1D | | IgG-1D noisier | | IgG-1D noisiest | | IgG-RL | | Ribosembly | | Tomotwin-100 | | Spike-MD | |
|---|---|---|---|---|---|---|---|---|---|---|---|---|---|---|
| | Rot | Trans | Rot | Trans | Rot | Trans | Rot | Trans | Rot | Trans | Rot | Trans | Rot | Trans |
| CryoDRGN2 | 6.601 | 1.056 | 6.282 | 1.035 | **55.876** | **2.707** | 6.546 | 2.527 | **1.847** | **0.728** | **109.581** | **2.156** | 3.457 | 2.022 |
| CryoDRGN-AI | 5.216 | **0.412** | 5.884 | **0.435** | 82.657 | 4.264 | **2.712** | **0.215** | 2.032 | 0.872 | 110.987 | 5.347 | **1.562** | **0.265** |
| 3D Class abinit | **3.403** | 0.933 | **4.312** | 1.009 | 82.621 | 3.583 | 4.658 | 0.742 | 2.834 | 1.492 | 113.629 | 8.980 | 1.783 | 0.489 |

Table S7: **Pose Error.** Errors are quantified by the median rotation and translation errors compared to the ground truth image poses after global reference frame alignment. The rotation errors are defined as the median of the Geodesic error in units of degrees and the translation errors are defined as the median of the L2 distance in units of pixels after alignment. For the *Tomotwin-100* dataset, each structure was aligned separately for all 100 different structures.

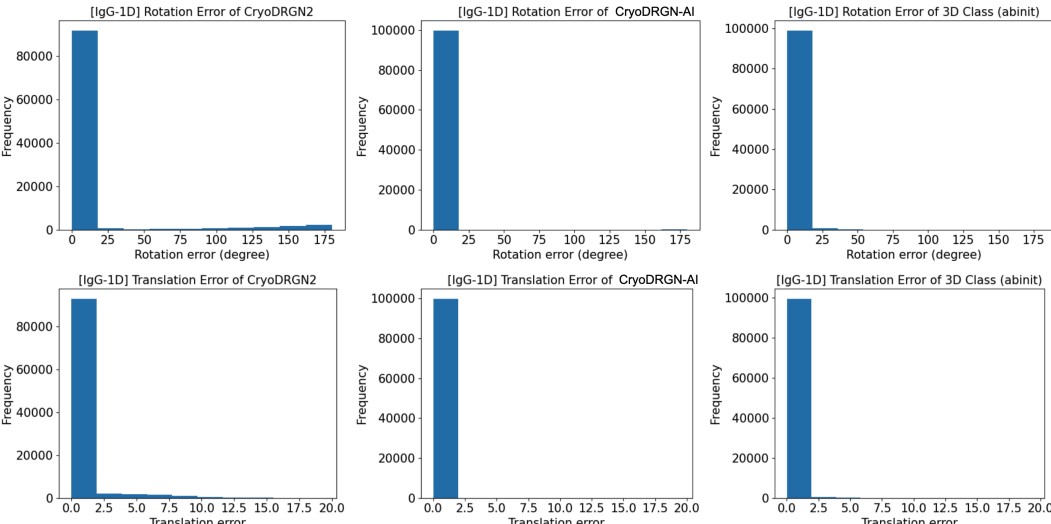

Figure S25: **Pose error for IgG-1D.** Histogram of rotation and translation errors. The first row shows rotation errors, and the second row shows translation errors.

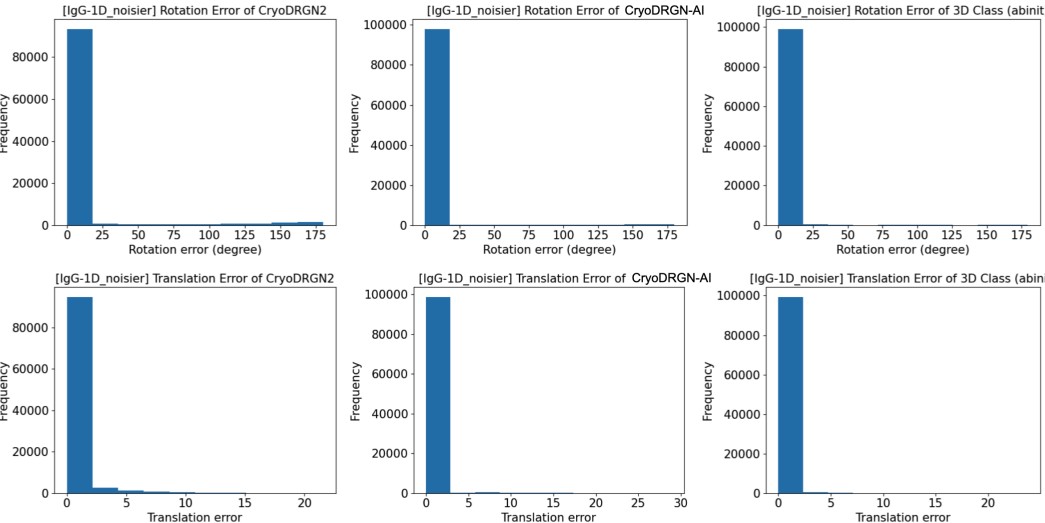

Figure S26: **Pose error for IgG-1D noisier.** Histogram of rotation and translation errors. The first row shows rotation errors, and the second row shows translation errors.

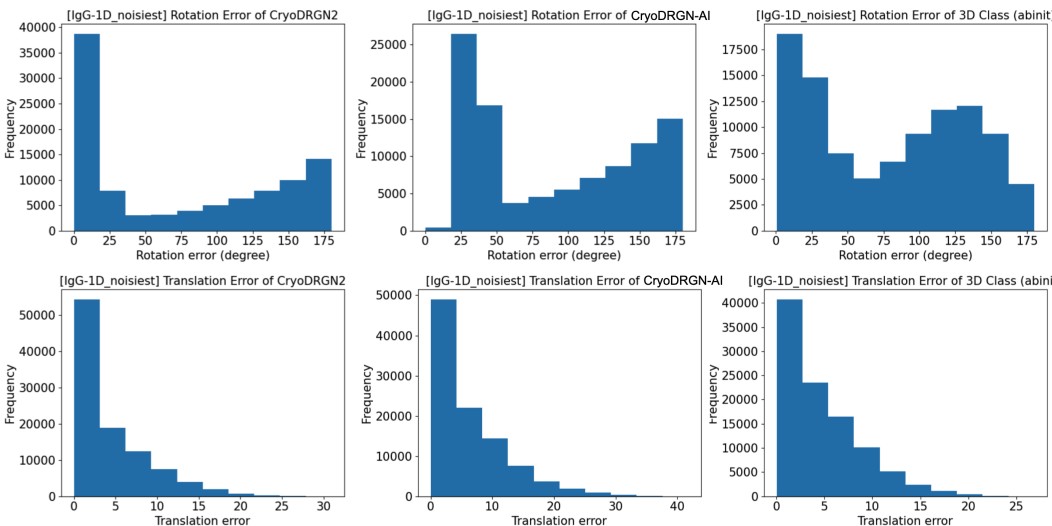

Figure S27: **Pose error for IgG-1D noisiest.** Histogram of rotation and translation errors. The first row shows rotation errors, and the second row shows translation errors.

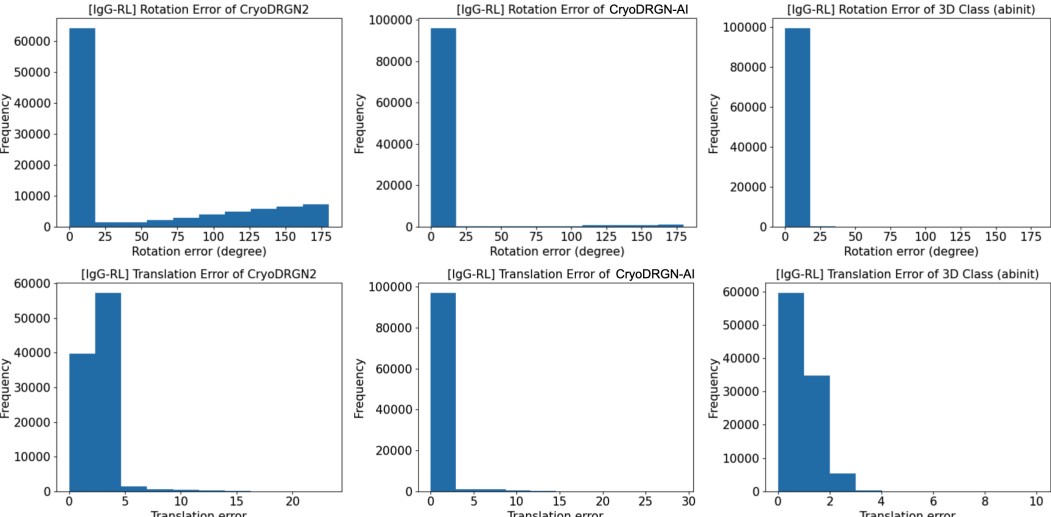

Figure S28: **Pose error for IgG-RL.** Histogram of rotation and translation errors. The first row shows rotation errors, and the second row shows translation errors.

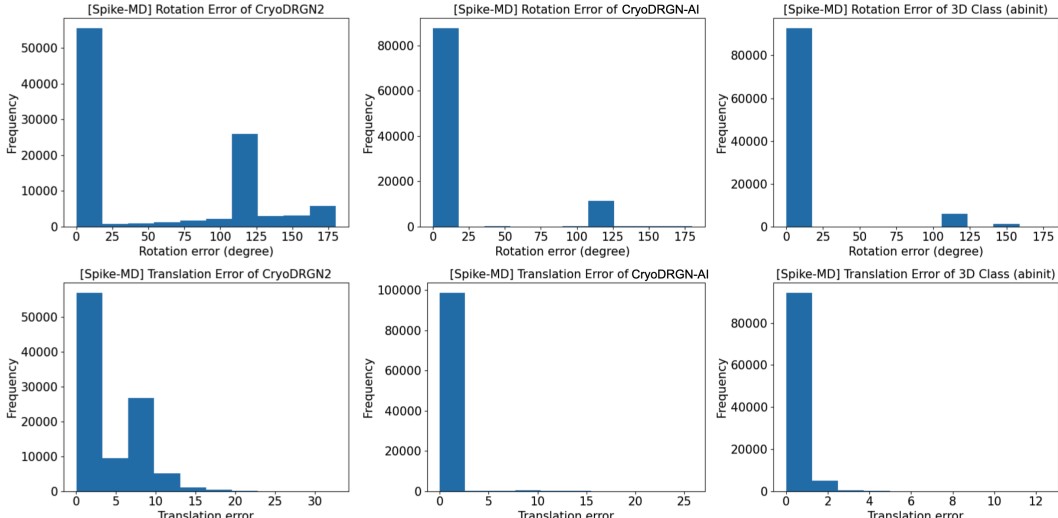

Figure S29: **Pose error for Spike-MD.** Histogram of rotation and translation errors. The first row shows rotation errors, and the second row shows translation errors.

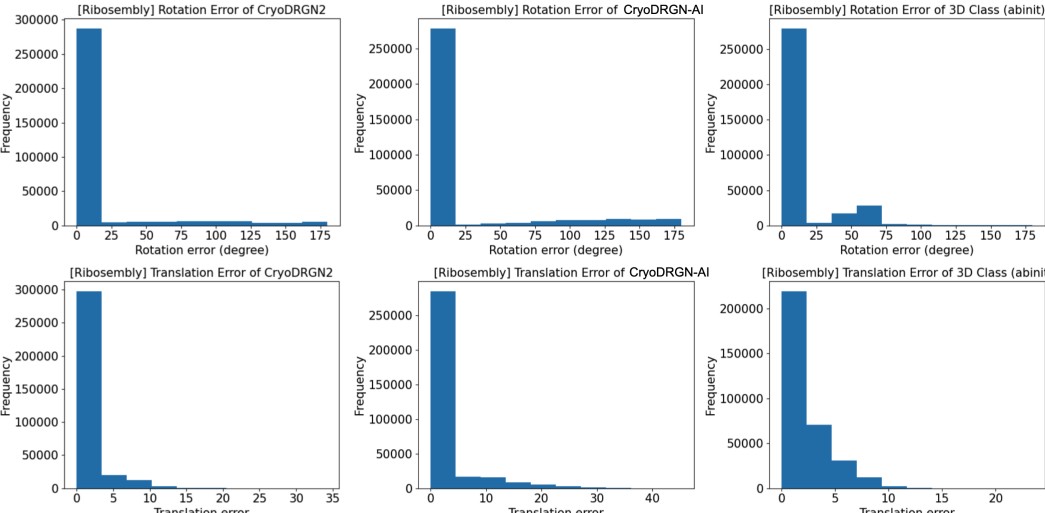

Figure S30: **Pose error for Ribosembly.** Histogram of rotation and translation errors. The first row shows rotation errors, and the second row shows translation errors.

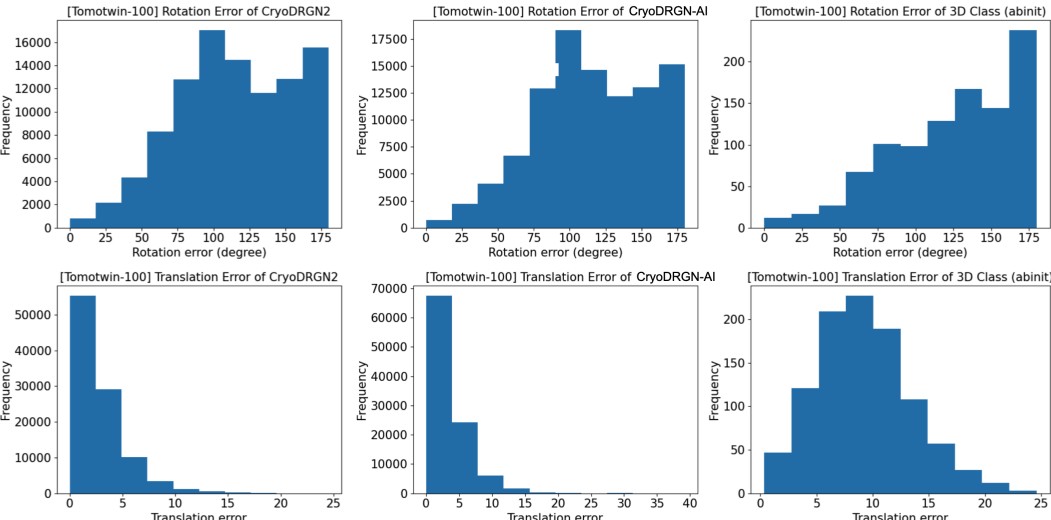

Figure S31: **Pose error for Tomotwin-100.** Histogram of rotation and translation errors. The first row shows rotation errors, and the second row shows translation errors.

# D   Glossary of Terms from Single-Particle Electron Cryo-Microscopy

## D.1   Sample

- **Biomolecular**: Pertaining to molecules involved in the biological processes of living organisms, such as proteins and nucleic acids.

- **Protein**: Large, complex molecules made up of amino acids, essential for various biological functions like catalyzing metabolic reactions and DNA replication.

- **Nucleic Acid**: A type of biomolecule, including (deoxy)ribonucleic acid (DNA, RNA, respectively). This term can refer to a single unit that can polymerize (form a long chain).

- **Complex**: In the context of biomolecular complexes, the term 'complex' refers to a stable association of two or more biomolecules that interact with each other, typically to perform a specific biological function. The interactions that hold these molecules together can be non-covalent, such as hydrogen bonds, ionic interactions, van der Waals forces, and hydrophobic effects, or covalent, such as disulfide bonds.

- **Subunit**: a part of a larger whole. The part (domain, polypeptide) is contextual to the whole (domain, protein complex).

## D.2   Heterogeneity

- **Heterogeneity**: The presence of variations in shape or the presence or absence of mass within a sample. Coming in two main sub-classes
  - **Compositional**: Related to the total amount of mass and their proportions within a sample or structure. Often used in the context of discrete differences in total mass.
  - **Conformational**: Pertaining to the various shapes or structures that a molecule can adopt. Often used in the context of continuous movement in 3D space.

- **Conformation**: The specific three-dimensional arrangement of atoms in a molecule. Often employed in the plural to refer to the different shapes a particular biomolecule can adopt.

- **(Biomolecular) Ensemble**: The collection of multiple conformations of a biomolecule. Biomolecules are typically dynamic and change conformations due to thermal fluctuations and interactions to carry out function [60, 61]. This concept is related to microstates and macrostates in statistical thermodynamics. A microstate represents a specific arrangement of particles at a given moment, while a macrostate describes the overall properties of a system, encompassing many possible microstates. Similarly, a biomolecular ensemble in cryo-EM represents the collection of microstates (individual conformations) that contribute to the observed macrostate (overall structural and functional behavior) of the biomolecule.

- **Collective Variable (CV)**: A parameter used to describe the state of a system, typically in terms of a few degrees of freedom. Further distinguished into geometric (centre of mass, angle, distance) and abstract [62]. The term CV is related to 'order parameter', and 'reaction coordinate', which is often used in the context of reactants and products in chemical catalysis [63]. However, as employed in the biomolecular simulation community, CVs typically relate to distinguishing metastable states [64].

## D.3   Model and Representation

- **Angstrom (Å)**: A unit of length equal to $0.1$ nm, or $10^{-10}$ m. Often used in chemistry because the diameter of an atom and the distance between atoms is close to $1$ Å.

- **Voxel**: A volume element representing an intensity value on a regular grid in three-dimensional space, similar to a pixel in 2D images but for a 3D array. A typical voxel volume ranges $0.5^3 - 2^3$ Å$^3$.

- **3D Map, Volume, Density**: A representation of structural data, in cryo-EM this typically refers to the 3D Coulombic (electric, electrostatic) potential instead of the electron density in other structural biology techniques based on X-ray diffraction. [65, 66]

- **Latent**: Hidden variables inferred from observed data, representing underlying structures or features in the model not directly observed.

- **Embedding**: A representation of data, for example a continuous n-dimensional vector space. Used to concretely parametrize or otherwise numerically represent a latent variable.
- **White Gaussian Noise**: noise with a flat power spectral density, meaning that its power is uniformly distributed across all frequencies. This implies that the noise has equal intensity at different frequencies, making it 'white' by analogy to white light, which contains all visible wavelengths.

## D.4  Microscopy

- **Point Spread Function (PSF)**: A function describing the response of an imaging system to a point source, indicating, for example, the system's resolution and blur characteristics.
- **Contrast Transfer Function (CTF)**: The Fourier transform of the point spread function. A mathematical description of how an electron microscope transfers contrast from the specimen to the image, influenced by various microscope parameters. We employ a common parametric form which depends on beam energy (electron wave length via the de Broglie relation), defocus and its astigmatism, spherical aberration, and amplitude contrast (ratio) [67].
- **Microscope Effects**: Artifacts and distortions introduced by the electron microscope during image acquisition. At times used in a phenomenological sense to describe effects not modelled well by the PSF/CTF.
- **Camera Effects**: Distortions or noise introduced by the optical system used to capture images. Can be used in a wide sense beyond detector effects for the entire optical system.

## D.5  Image Acquisition and Analysis

- **Micrograph**: A two dimensional image obtained using an electron microscope, typically showing a field of view that includes multiple particles. Often the image contains temporal frames in a 'movie' format, which is corrected for motion. A typical micrograph is approximately $4000^2$ pix$^2$, at $0.5 - 2$ Å per pixel.
- **Particle**: Individual biomolecular structures captured within a patch of micrograph, which is typically boxed out of the wide frame image. Can refer to the physical entity in the image, or the recorded measurement. A typical particle is approximitely $64^2 - 512^2$ pix$^2$, at $0.5 - 2$ Å per pixel.
- **Reconstruction**: a 3D volume, typically in a real spaced voxelized array form, generated by processing data from a series of two-dimensional 2D images. Distinguished further to homogeneous (one 3D volume) and heterogeneous (multiple 3D volumes).

