## A  Data and Software Availability

### A.1  Data Availability

CryoBench datasets are deposited on Zenodo at DOI: 10.5281/zenodo.11629428. We include the downsampled images ($D = 128$) analyzed in this study in `.mrcs`, `.txt`, and `.star` file formats, along with CTFs and pose data in pickle files. We also include the consensus volume and mask used for FSC computation. Full resolution images ($D = 256, 384$) and ground truth PDB files and volumes will be deposited to EMPIAR [42]. We provide the datasets under the Creative Commons Attribution 4.0 International license.

### A.2  Software Availability

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

## B.6 Signal to Noise Ratio (SNR)

We define SNR as the ratio between the variance of the signal and the variance of the noise following [47]. We calculated the standard deviation of the signal ($\sigma_{\text{signal}}$) over all CTF-applied projection images. We then computed $\sigma_{\text{noise}} = \sigma_{\text{signal}}/\sqrt{\text{SNR}}$. Finally, we added noise to each particle, drawn from a Gaussian distribution with a mean of 0 and a standard deviation of $\sigma_{\text{noise}}$.

Additionally, we illustrate cryo-EM images for all datasets in Figure 9.

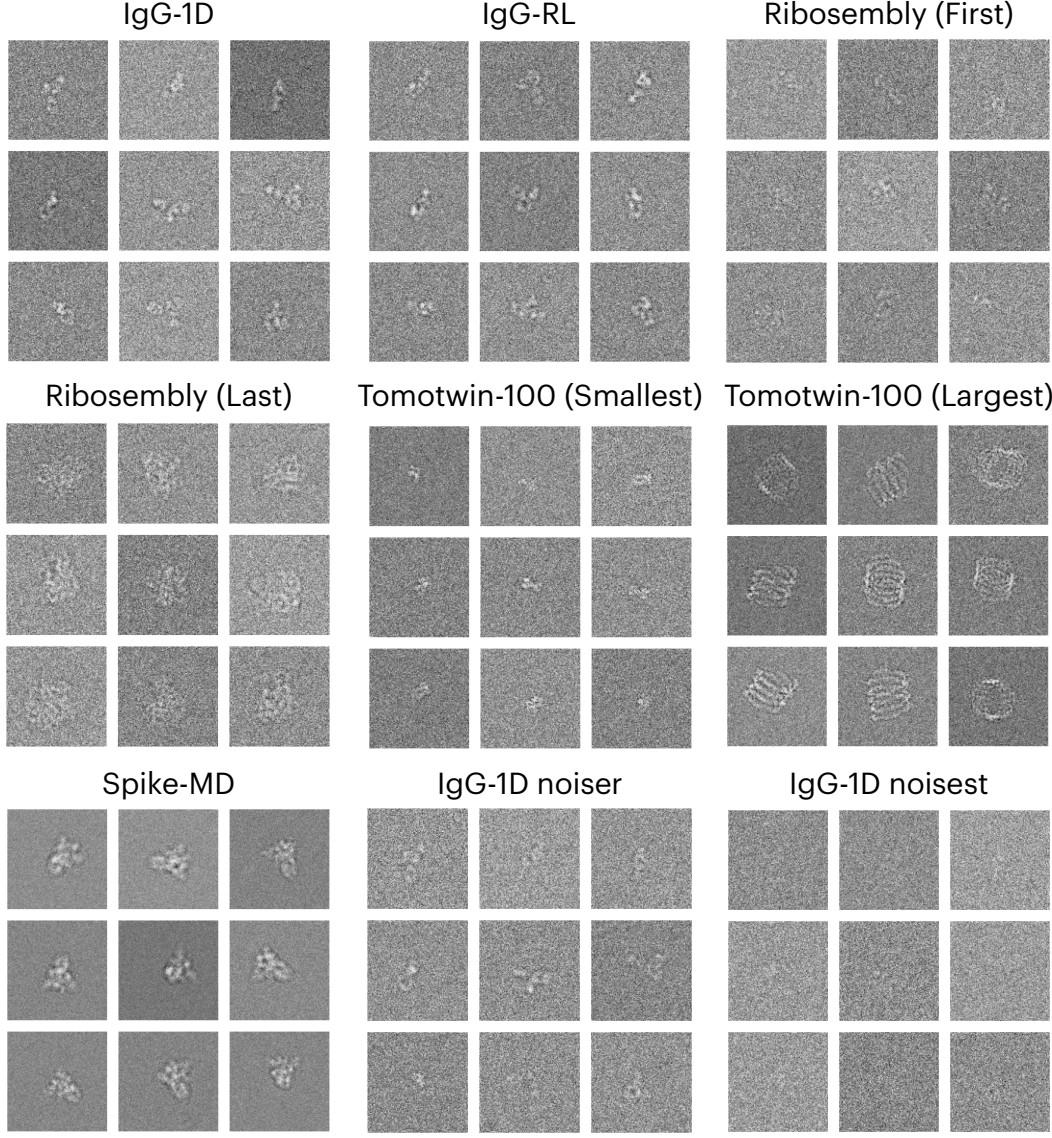

Figure 9: **Cryo-EM images for all datasets.** The first structures are shown for `IgG-1D` and `IgG-RL`, the first and last structures are shown for `Ribosembly` and `Tomotwin-100`, and a mix of structures is shown for `Spike-MD`.

## C Experimental Settings

### C.1 CryoDRGN, CryoDRGN2

CryoDRGN [5] is a deep generative network-based method where the input images are encoded in the (conformational) latent space and the latent coordinates are decoded into 3D volumes in Fourier domain via an implicit neural representation [4]. In its second version CryoDRGN2 [41], better *ab initio* capabilities were improved with changes to the hierarchical pose search (HPS) algorithm for image pose inference. In our benchmark, we use CryoDRGN for *fixed*, and CryoDRGN2 for *ab initio* purposes.

We trained CryoDRGN and CryoDRGN2 using the official PyTorch implementation[1], version 3.0.0b. We used the default settings with the z-dimension set to 8. For the total number of training epochs, 20 and 30 were used, respectively. We used one V100 GPU for training.

### C.2 DRGN-AI, DRGN-AI-fixed

DRGN-AI [40] is a deep generative network-based method, inspired by CryoDRGN. DRGN-AI uses both HPS and stochastic gradient descent in pose estimation, while utilizing a differential lookup table instead of an encoder network to encode the pose and conformational latent variable information. We denote the *fixed pose* mode of operation with "DRGN-AI-fixed" and *ab initio* with "DRGN-AI."

We trained DRGN-AI and DRGN-AI-fixed using the official PyTorch implementation[2], version 0.2.2b0. We used the default settings with the z-dimension set to 4 and the total number of training epochs set to 100. We used one A100 GPU for training.

### C.3 Opus-DSD

Opus-DSD [9] is also a deep generative network-based method, built upon CryoDRGN. The network architecture is similar to CryoDRGN except that it uses a 3D Convolutional Neural Network (CNN) and priors for the latent conformational variable.

We trained Opus-DSD using the official PyTorch implementation[3]. We used the default settings with the z-dimension set to 12, `valfrac` of 0.25, `downfrac` of 0.75, and `lamb` of 1.0, `bfactor` of 4.0, and `templateres` of 192 as recommended on the official GitHub. For the `Spike-MD` dataset, we use a `downfrac` of 1.00 and `templateres` of 256. The total number of training epochs was set to 20. The volume reconstructed by Opus-DSD is smaller than the original image dimensions. Consequently, to compute the volume metric (Per-Conformation FSC), we added zero paddings to match the dimensions of the original image. We used four A100 GPUs for training.

### C.4 RECOVAR

RECOVAR [10] is a white-box approach that utilizes principal component analysis (PCA), which is computed through regularized covariance estimation.

We trained RECOVAR using the official PyTorch implementation[4]. We used the default settings with the z-dimension set to 10 and applied the mask as an input. We used one V100 GPU for training.

### C.5 CryoSPARC

We used the official CryoSPARC[5] version 4.4.0 to train 3DFlex, 3DVA, 3D Classificaion (fixed, *ab initio*). Some methods in CryoSPARC require a consensus volume. We created this volume for

---

[1] `https://github.com/ml-struct-bio/cryodrgn`
[2] `https://github.com/ml-struct-bio/drgnai`
[3] `https://github.com/alncat/opusDSD`
[4] `https://github.com/ma-gilles/recovar`
[5] `https://cryosparc.com`

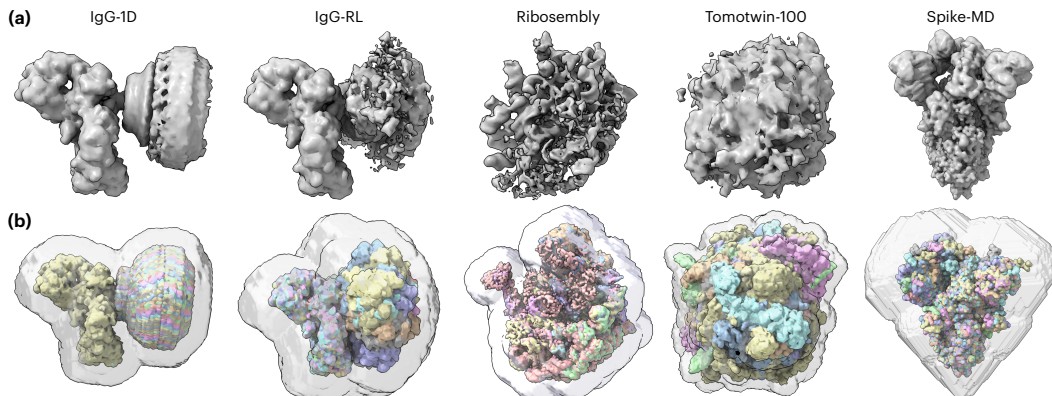

Figure 10: **Consensus volumes and Masks.** (a) Consensus volumes (Backprojection) for each dataset (b) Mask for each dataset. 10 G.T. volumes are shown within the mask for `Spike-MD`, and all G.T. volumes are shown for other datasets.

each dataset by using the backprojection [5] of all corresponding cryo-EM images. We provide the backprojected volume (consensus volume) and masks in Figure 10.

**3DFlex.** 3DFlex [39] is a heterogeneous reconstruction method provided in the CryoSPARC software suite. 3DFlex is a deep learning-based method in which a deep neural network is trained to construct deformation flow fields as a function of the conformational latent space coordinates to construct the heterogeneous reconstruction as a "deformation" of the single canonical 3D volume.

In the mesh preparation phase (*Flex Mesh Prep*), we provided the consensus volume and mask as inputs. We adjusted the settings as follows: `Mask threshold` was set to 2, `Mask dilation` to 5, `Mask soft padding` to 10, `Min.rigidity weight` to 1. For the *3D Flex Training*, we set the `Rigidity` parameter to 10 and left all other training parameters to their default settings. The z-dimension is 2.

Due to the its high levels of heterogeneity, Spike-MD required special treatment. First, the particle stack was normalised such that the mean of each image was 0 and the variance was 1. A 3DFlex model was trained with consensus poses and volume from ab initio reconstruction, and the following hyperparameters. The number of latent dimensions was 3, the MLP neural network which dictates the deformations of the 3DFlex model had 256 hidden layers, we trained the model for 32 epochs beyond the standard training time. All other parameters were left to their default values.

**3DVA.** 3DVA [7] is a heterogeneous reconstruction algorithm, which is formulated as a Probabilistic PCA approach and utilizes E-M to obtain the heterogeneous reconstructions.

We provided the particles and mask as inputs and set the latent dimension to 3 (default). Moreover, the `Filter resolution` was set to 5 for `Spike-MD`, 10 for `IgG-1D`, `IgG-RL`, and `Ribosembly`, and 15 for `Tomotwin-100`.

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

different neighbourhood sizes, and here we used $k = 1, 3, 10, 30$ $(0.05, 0.015, 0.5, 1.5\%)$ of neighbourhood size. Significantly larger neighbourhood sizes approached the orthogonal (1,1) region. Error bars are visible in Tomotwin-100 (Fig. 6d), but smaller than marker size for other datasets.

Small amounts of smearing were applied to average over the 1000-fold duplication of the ground truth heterogeneity latent in the image. Additive noise from a uniform distribution, $u \sim U[-\epsilon, \epsilon]$ was added according to Table 4.

The ground truth pose embedding is a 9 dimensional flattened vector of the rotation matrix (translation neglected). The ground truth CTF embedding is a 4 dimensional vector of the two defoci, and the sine and cosine of the angle of astigmatism, normalized by subtracting off the mean and dividing by the standard deviation.

| Dataset | Collective Variable | $\epsilon$ |
|---|---|---|
| IgG-1D | angle in degrees (before sine / cosine transform) | 0.05 |
| IgG-RL | center of mass (Å), angle in degrees (before sine / cosine transform) | 0.1 |
| Ribosembly | voxel intensity | 0.1 |
| Tomotwin-100 | rank size | 0.1 |
| MD | CV1 and CV2 | 0.1 |

Table 4: Smearing ground truth heterogeneity latent embeddings.

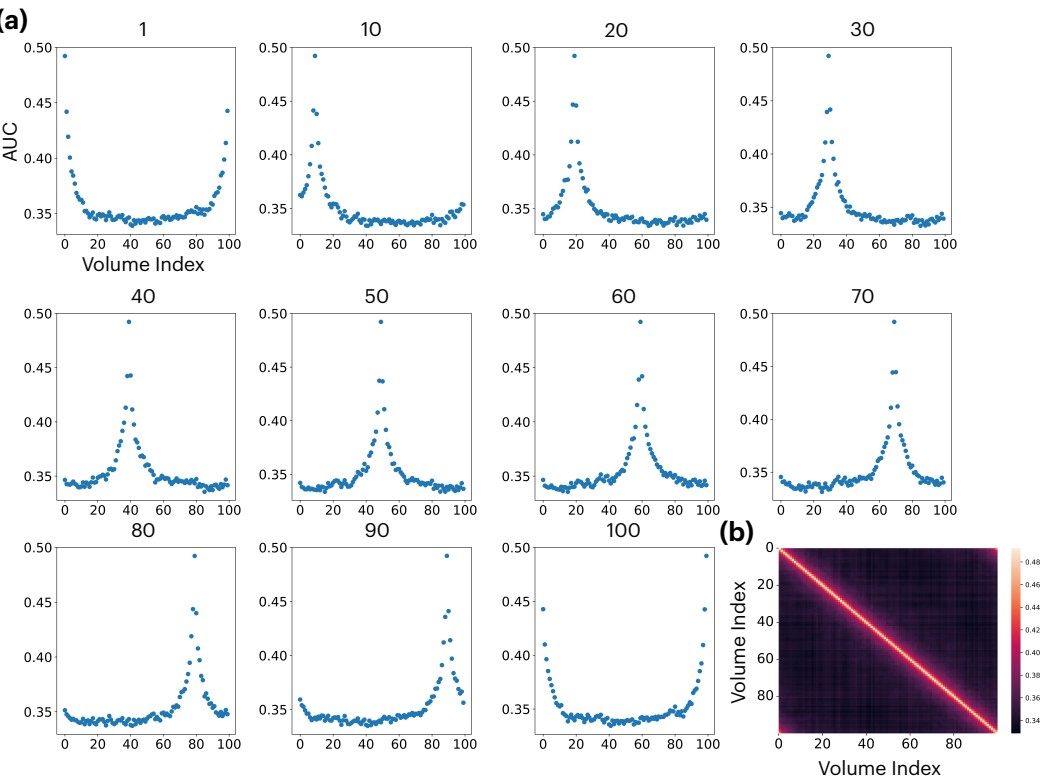

Figure 11: **Metric verification. (a)** AUC-FSC between one G.T and all 100 G.T.s of the `IgG-1D` dataset. Each plot corresponds to the reference G.T volume, indicated by the number above the plot. **(b)** Heatmap comparing all 100 G.Ts against all 100 G.Ts.

# D    Supplementary Results

## D.1    Metric verification

**UMAP visualization.** In Section 5, we provide UMAP plots computing using the official framework[6], applying the default parameters.

**AUC-FSC.** Figure 11(a) illustrates the AUC-FSC for the ground truth volumes of `IgG-1D` dataset. The AUC reaches its highest point at one specific index, indicating the value is sensitive to structural differences. Given that the IgG-1D dataset includes 1D circular motion, the volume indices 1 and 100 show two peak points. Figure 11(b) demonstrates that the heatmap displays the highest values when AUC values are compared between identical volumes.

## D.2    Mask vs No Mask

We utilize a mask when computing the FSC metrics reported elsewhere in the text. Here, we provide an analysis comparing the use of a mask versus no mask with Per-Conformation FSC (Fig. 12). For mask generation, we first aggregated all ground truth volumes using the `volume add` in ChimeraX. Subsequently, we then applied the `Volume Tools` in CryoSPARC. Specifically, for `IgG-1D`, `IgG-RL`, and `Ribosembly`, the `Dilation radius (pix)` and `Soft padding width (pix)` were set at 8 and 5, respectively. For `Tomotwin-100`, these parameters were adjusted to a `Dilation radius (pix)` of 5 and a `Soft padding width (pix)` of 3. For `Spike-MD`, we take the union of all binarized volumes and use the cryoDRGN `gen_mask` command with a dilation of 25 Å and soft

---

[6]`https://umap-learn.readthedocs.io/en/latest/api.html`

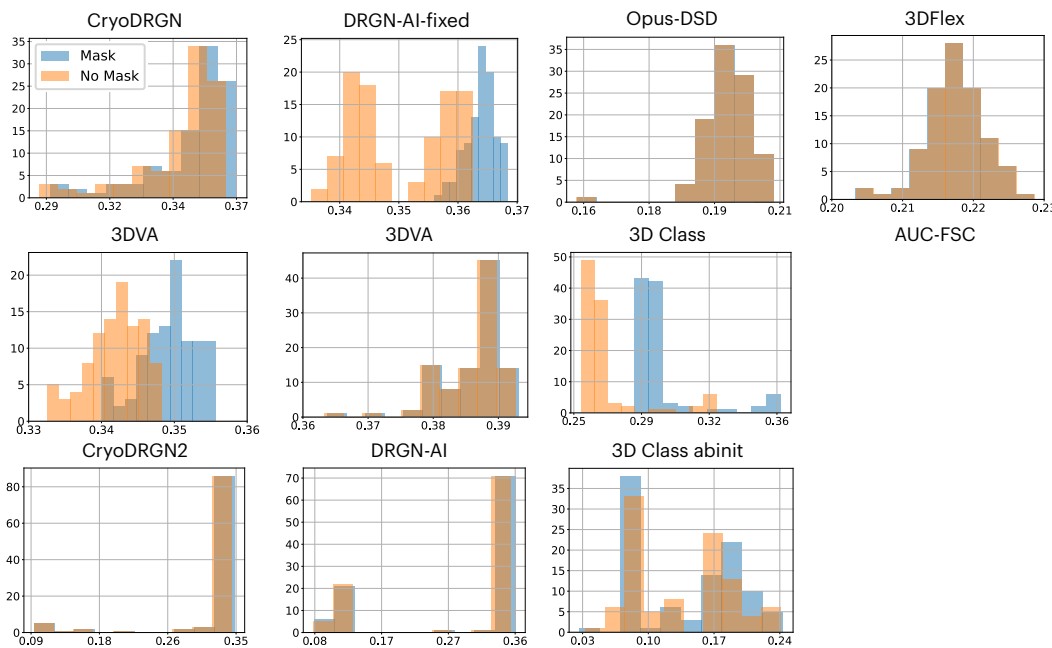

Figure 12: **Mask comparison with IgG-1D.** Histogram comparing Per-Conformation FSC for each method, with and without a mask.

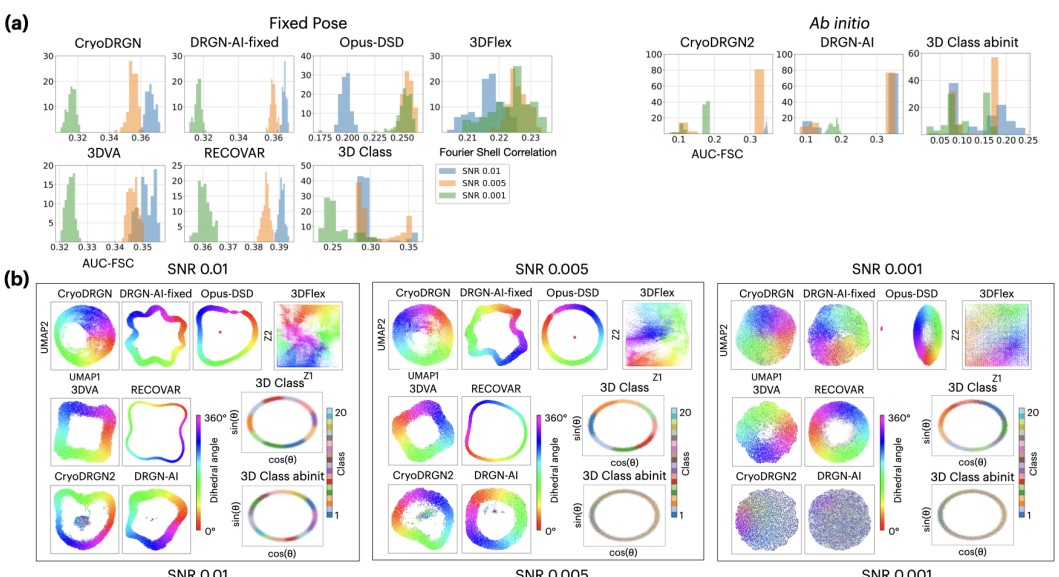

Figure 13: **IgG-1D with noise. (a)** Histogram of Per-Conformation FSC for each method at SNR levels of 0.01, 0.005, 0.001. **(b)** UMAP visualizations colored by G.T. dihedral conformations of each method.

cosine edge of 15 Å. Masking out background noise generally enhances performance when computing volume metrics.

### D.3 Noise Comparison

As shown in Figure 13, we applied higher noise settings (SNR 0.005, 0.001) to the `IgG-1D` dataset. With increasing noise levels, there is a noticeable reduction in volume metrics, and the capability to differentiate between different conformations decreases.

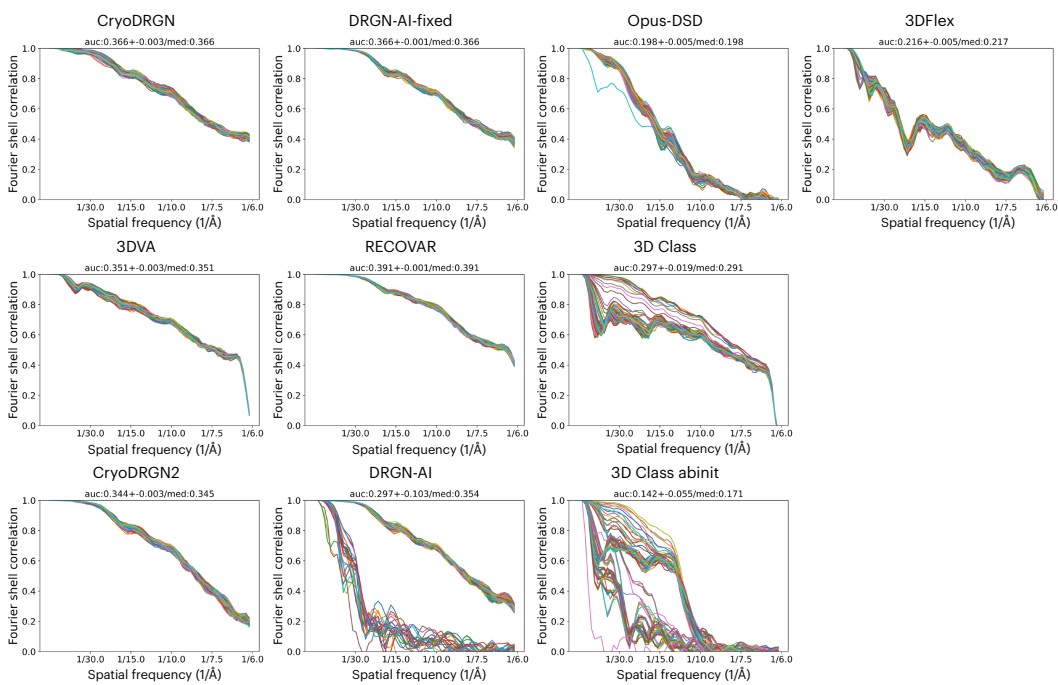

Figure 14: **Per-Conformation FSC per particle.** All 100 FSCs for the IgG-1D dataset at an SNR level of 0.01. Masks were applied to compute the FSCs.

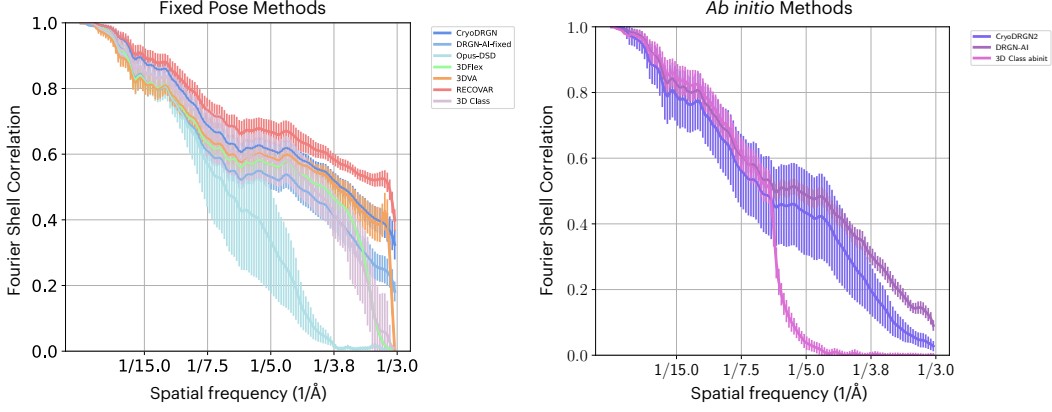

Figure 15: **Per-Conformation FSC for Spike-MD.**

## D.4 Per-Conformation FSC

We presented the average values and error bars for Per-Conformation FSC across all datasets for each method in the Figure 2, 3, 4, 6, 7. In this section, we illustrate all 100 FSC plots for the `IgG-1D` dataset for all methods in Figure 14. Additionally, we present FSC curves for the `Spike-MD` dataset in Figure 15.

## D.5 Volume FSC

We illustrate the *Volume FSC* plots for each method across all datasets in Figure 16. Given a reconstructed volume, the AUC of the FSC at varying resolutions is computed between the reconstructed volume and all original volumes. The maximum AUC is taken to be its *Volume FSC*. The metric can

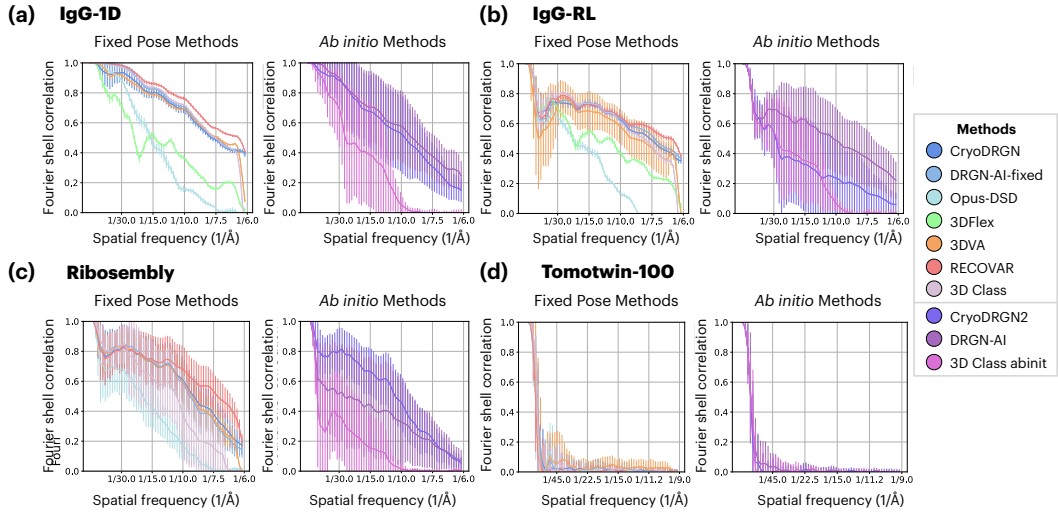

Figure 16: **Volume FSC.**

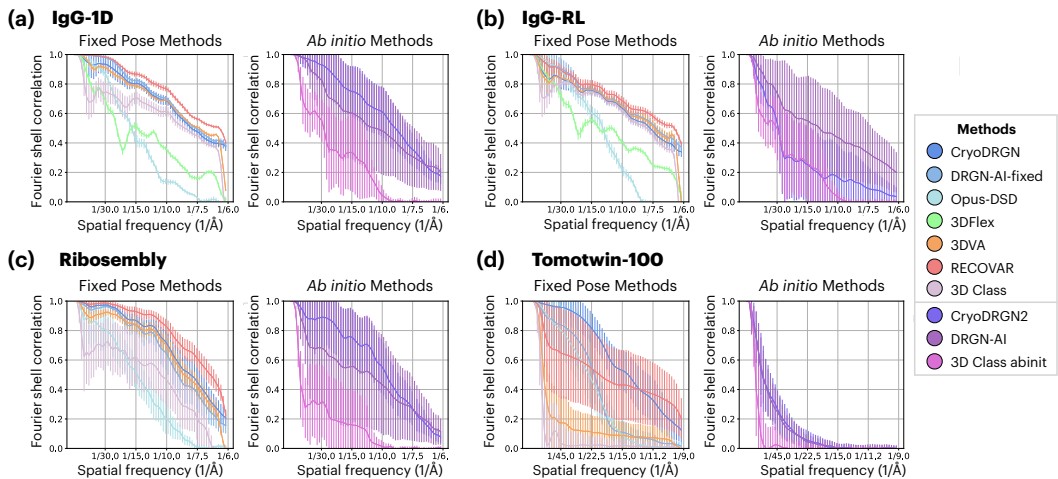

Figure 17: **Per-image FSC.**

be written as:

$$Volume\text{-}FSC(U) = \max_g \text{AUC}_t(x, FSC_t(U, V^{(g)})) \qquad (4)$$

$$FSC_t(U, V^{(g)}) = \left( \frac{\Sigma_{s \in S_t} U_s V_s^{(g)}}{\sqrt{\Sigma_{s \in S_t} |U_s|^2 \Sigma_{s \in S_t} |V_s^{(g)}|^2}} \right) \qquad (5)$$

where $U$ is the Fourier transform of the reconstructed volume, $V^{(g)}$ is the Fourier transform of the $g$'th ground truth volume, $S_t$ represents the set of Fourier voxels in a spherical shell at a distance $t$ from the origin, and $x$ denotes the resolution. In practice, we choose cluster centroid volumes of each method as representative reconstructions for evaluation.

## D.6 Per-image FSC

We propose *Per-image FSC* as a metric for jointly assessing reconstruction quality and image conformation estimation. Here, for each of 100 images uniformly chosen from the datasets, we reconstruct an associated volume and assess its FSC AUC to the image's ground truth volume. Thus,

unlike with *Volume FSC*, methods must produce a high quality reconstruction that is also consistent with the conformation in a given image. For 3DVA, we aggregate the consensus density map with all three eigen-volumes according to the latent coordinates of each image. For 3D Class, the class volume assigned to a given image is used as its reconstruction. Figure 17 provide Per-image FSC plots for each method across all datasets.

## D.7 Qualitative Evaluation

For the qualitative evaluation, we provide additional visualization results for the reconstructed volumes and UMAPs. Figure 18, 19, 20, 21, 22, 23, and 24 display K-means centers and UMAP, with dots corresponding to each center.

## D.8 Information Imbalance

**CTF and Pose:** Information imbalance with respect to the ground truth latent pose (rotation only, not translation) and CTF parameters is generally in the orthogonal region (1,1) for all methods (Figs. 25,26). However, zooming in, for pose, CryoDRGN and Opus-DSD are off the shared information x=y line, indicating their minor entanglement is more pronounced that other methods. For CTF the trends are less clear, but Opus-DSD and 3D Class abinit are generally the furthest away from the orthogonal region.

## D.9 Spike-MD embedding metrics

The percentage of matching neighbors was calculated as a function of the neighborhood radius for the Spike-MD dataset (Figure 27-left). Consistent with UMAP visualizations, we observe a relatively low similarity in neighborhoods between the embeddings and the ground truth molecular dynamics collective variables for small neighbhoorhood radii.

Information imbalance of the Spike-MD dataset (Figure 27-right) shows 3DVA on the shared information line at (0.5,0.5) - a very similar result as in IgG-1D. Opus-DSD and CryoDRGN2 are near (0.9,0.6), the closest to the orthogonal region for the Spike-MD dataset compared with other methods. For Opus-DSD, this is the closest to the orthogonal region compared with its information imbalance on the other datasets. For CryoDRGN2, this is a similar value as the challenging datasets (IgG-RL and Tomotwin-100). The other methods employed in these experiments (CryoDRGN, DRGN-AI-fixed, 3DFlex, RECOVAR, DRGN-AI) are closer to the equivalent zone and cluster together near (0.5,0.2).

## D.10 K-Means Clustering Accuracy

To additionally assess the ability of methods to classify particles arising from discrete structures, for `Ribosembly` and `Tomotwin-100`, we $k$-means cluster the latents for each method, with $k$ set to the number of ground truth structures in the dataset, and compare the cluster assignments to the true structural labels. We employ two common metrics for clustering consistency, the Adjusted Rand Index (ARI) and Adjust Mutual Information (AMI). As shown in Table 5, results are generally consistent with the clustering accuracy shown in Table 2, with RECOVAR and CryoDRGN performing the best on `Ribosembly` and `Tomotwin-100`, respectively.

| Method | Ribosembly | | Tomotwin-100 | |
|---|---|---|---|---|
| | ARI | AMI | ARI | AMI |
| CryoDRGN | 0.789 | 0.886 | **0.956** | **0.983** |
| DrgnAI-fixed | 0.718 | 0.854 | 0.791 | 0.906 |
| Opus-DSD | 0.707 | 0.812 | 0.500 | 0.781 |
| 3DVA | 0.726 | 0.860 | 0.058 | 0.335 |
| RECOVAR | **0.807** | **0.908** | 0.315 | 0.649 |
| CryoDRGN2 | 0.549 | 0.698 | **0.116** | **0.374** |
| DrgnAI-abinit | **0.630** | **0.800** | 0.086 | 0.275 |

Table 5: **K-Means Clustering Accuracy.** Adjusted Rand Index (ARI) and Adjusted Mutual Information (AMI) between true structural labels and predicted labels for each particle. Predicted labels are obtained by running $k$-means clustering on the particle latents, with $k$ set to the number of ground truth structures. These findings align with those previously reported for neighborhood similarity, as shown in Table 2.

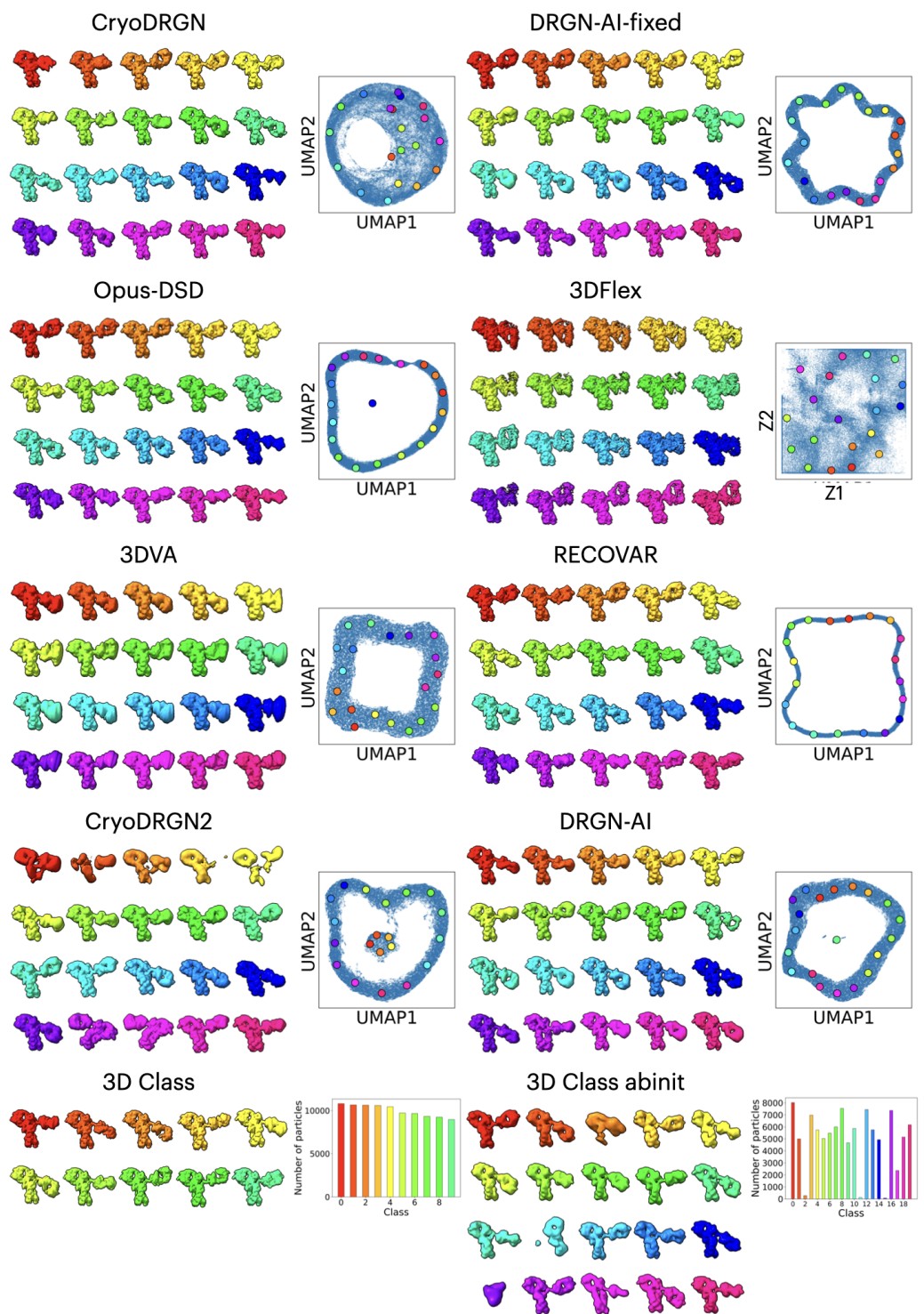

Figure 18: **Qualitative Results (IgG-1D).** For each method, representative volumes and a UMAP plot of the latent space are shown. Volumes correspond to K-Means cluster centers with K=20. Cluster centers are marked on the UMAP plot with a dot of the corresponding color. Class volumes and particle counts are shown for 3D Classification.

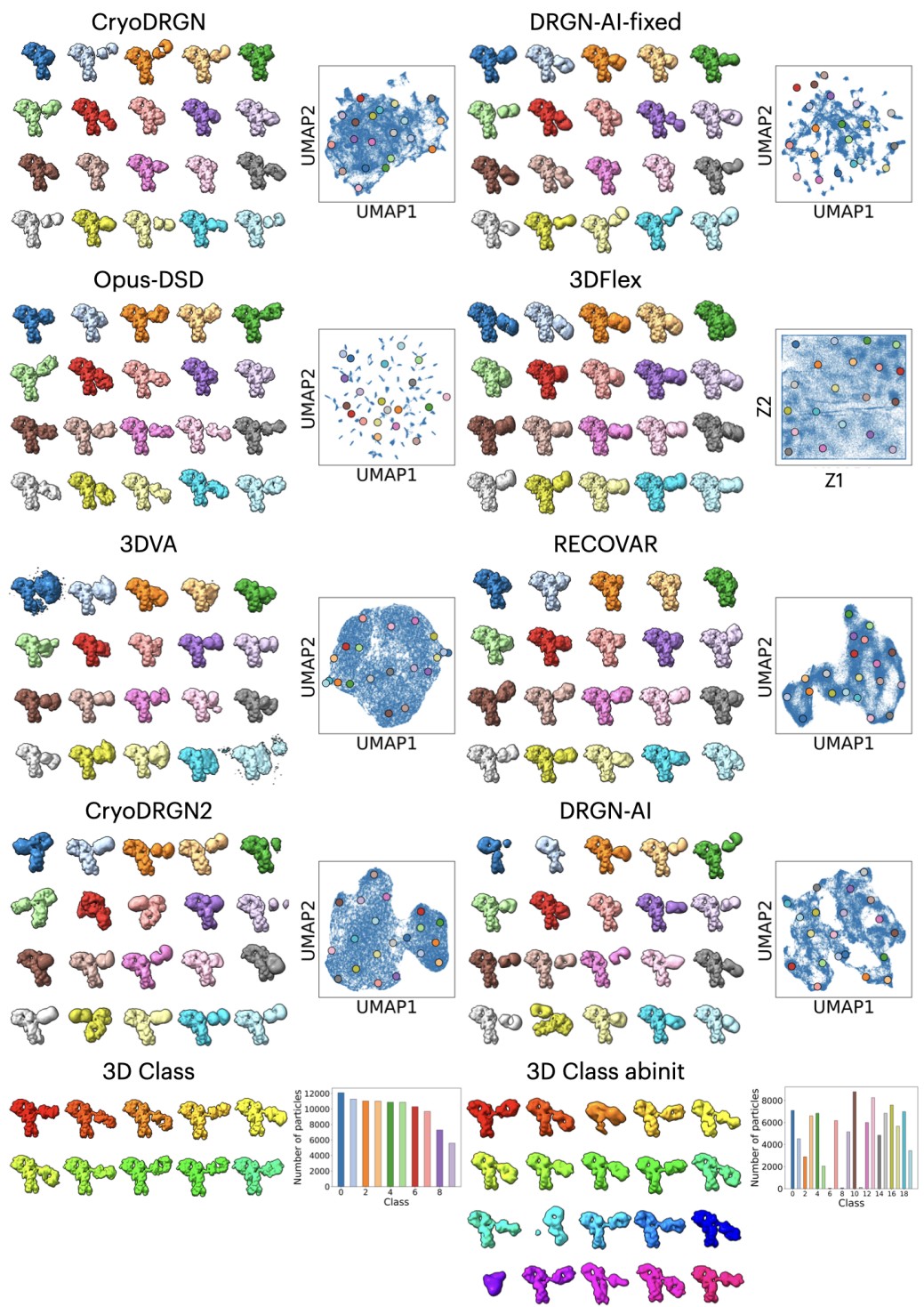

Figure 19: **Qualitative Results (IgG-RL).** For each method, representative volumes and a UMAP plot of the latent space are shown. Volumes correspond to K-Means cluster centers with K=20. Cluster centers are marked on the UMAP plot with a dot of the corresponding color. Class volumes and particle counts are shown for 3D Classification.

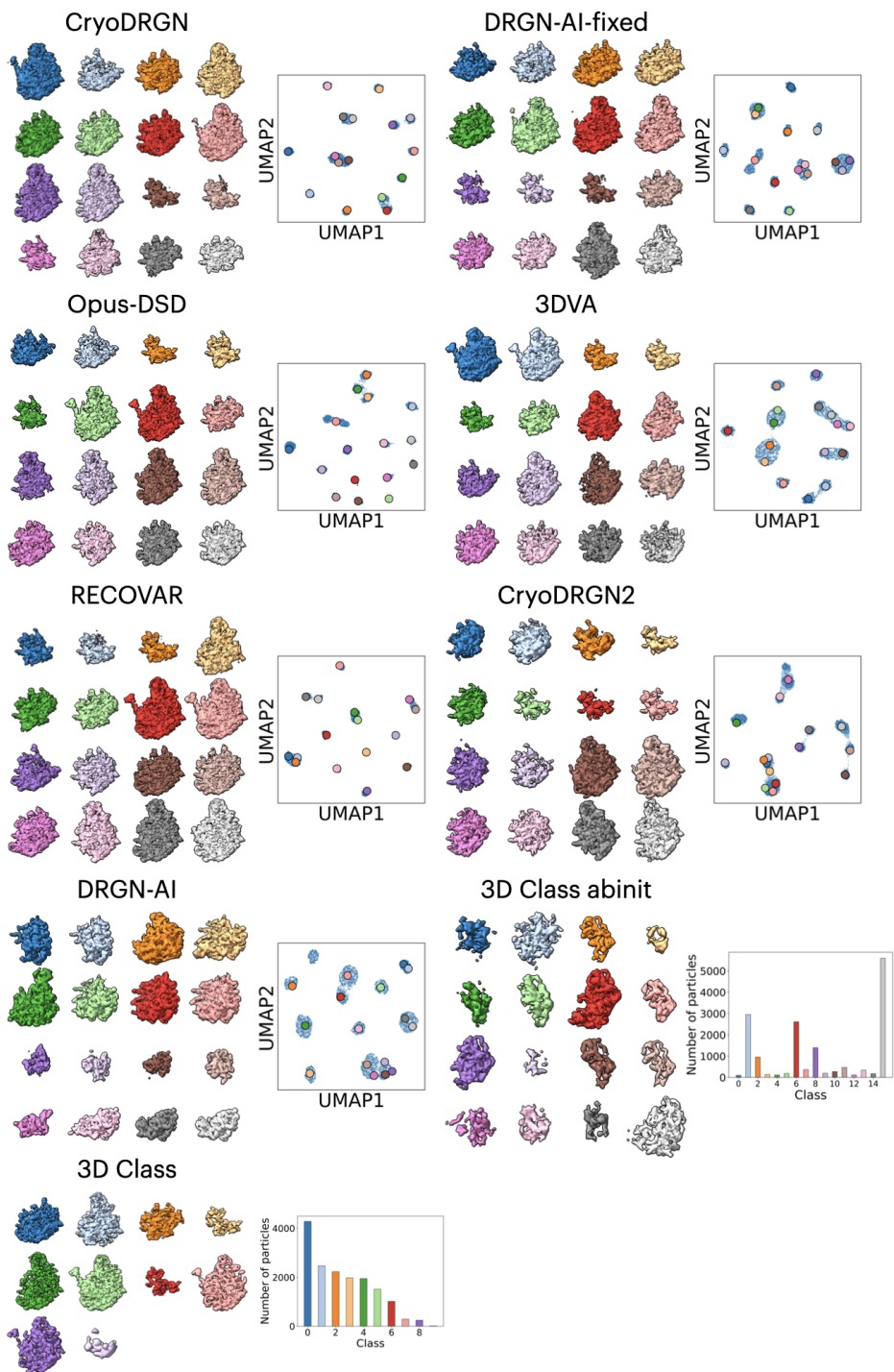

Figure 20: **Qualitative Results (Ribosembly).** For each method, representative volumes and a UMAP plot of the latent space are shown. Volumes correspond to K-Means cluster centers with K=20. Cluster centers are marked on the UMAP plot with a dot of the corresponding color. Class volumes and particle counts are shown for 3D Classification.

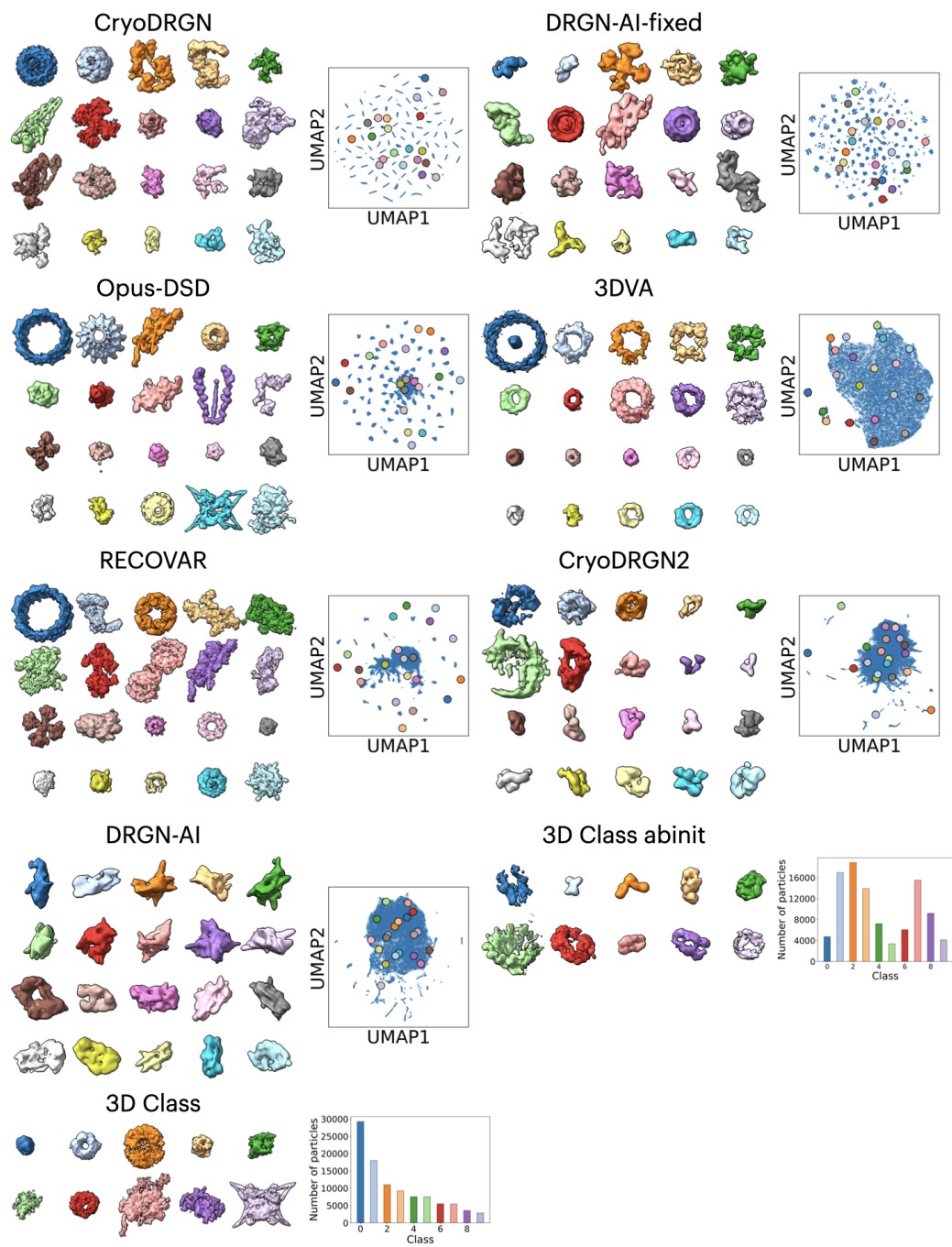

Figure 21: **Qualitative Results (Tomotwin-100).** For each method, representative volumes and a UMAP plot of the latent space are shown. Volumes correspond to K-Means cluster centers with K=20. Cluster centers are marked on the UMAP plot with a dot of the corresponding color. Class volumes and particle counts are shown for 3D Classification.

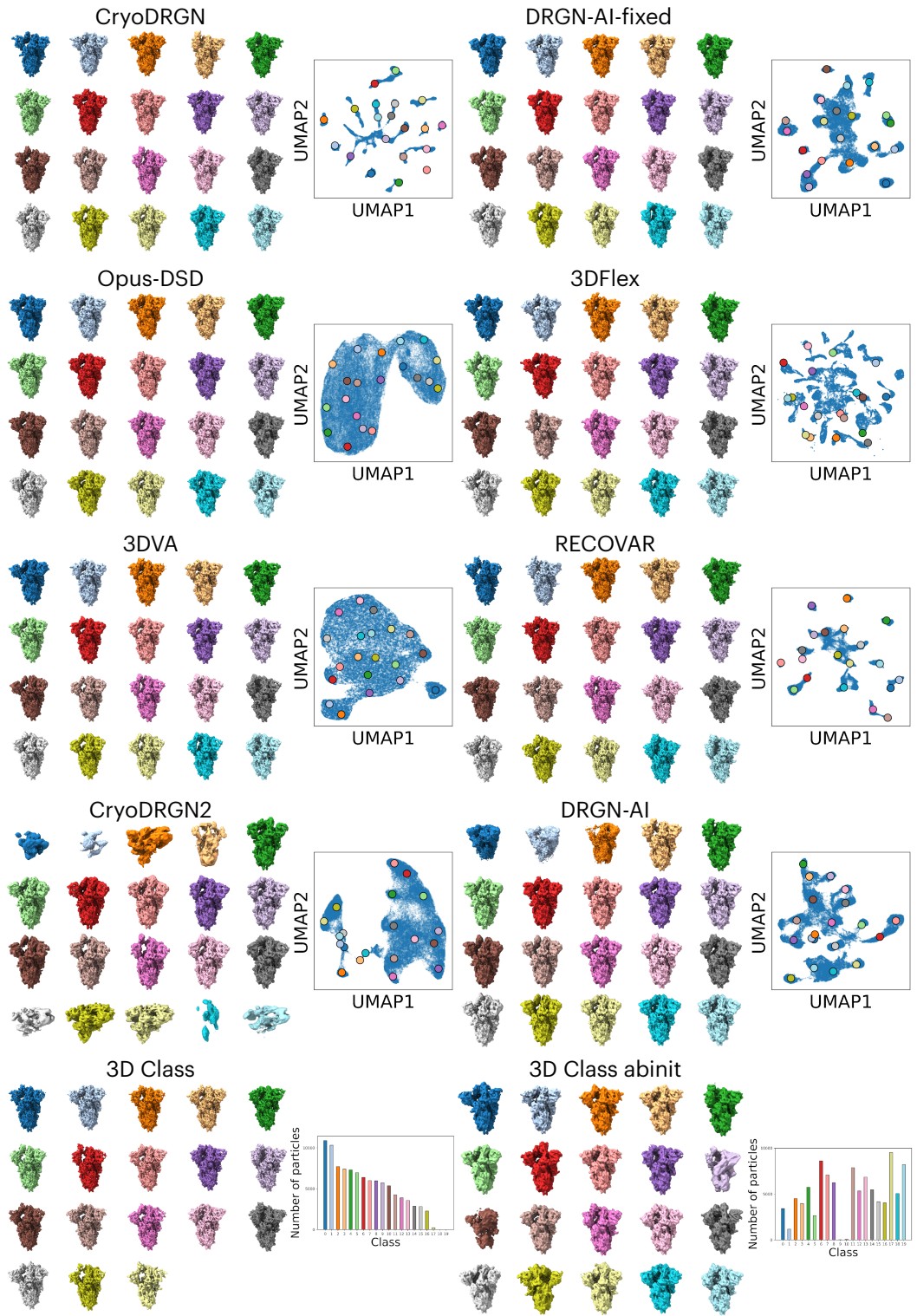

Figure 22: **Qualitative Results (Spike-MD).** For each method, representative volumes and a UMAP plot of the latent space are shown. Volumes correspond to K-Means cluster centers with K=20. Cluster centers are marked on the UMAP plot with a dot of the corresponding color. Class volumes and particle counts are shown for 3D Classification.

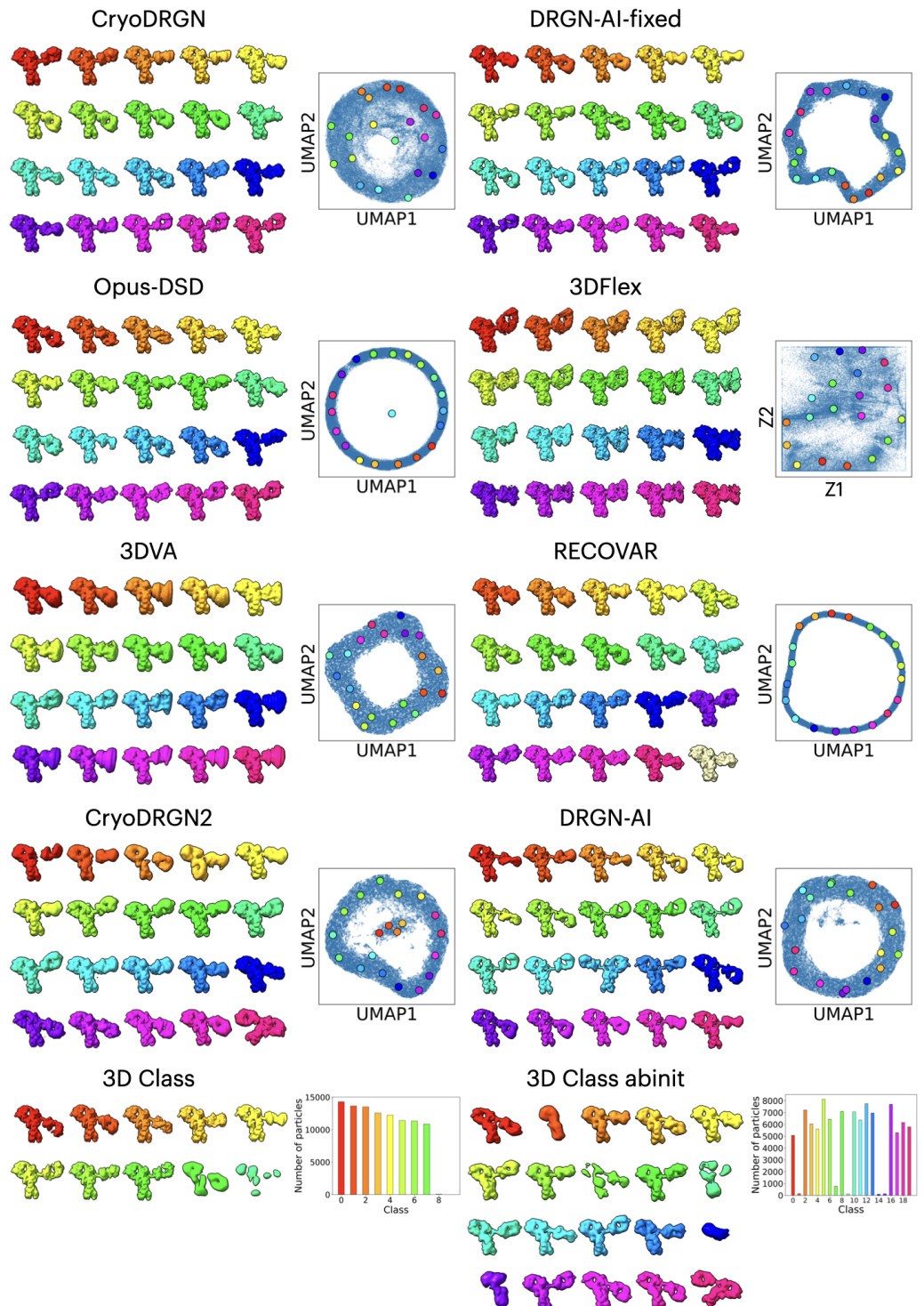

Figure 23: **Qualitative Results (IgG-1D noisier).** For each method, representative volumes and a UMAP plot of the latent space are shown. Volumes correspond to K-Means cluster centers with K=20. Cluster centers are marked on the UMAP plot with a dot of the corresponding color. Class volumes and particle counts are shown for 3D Classification.

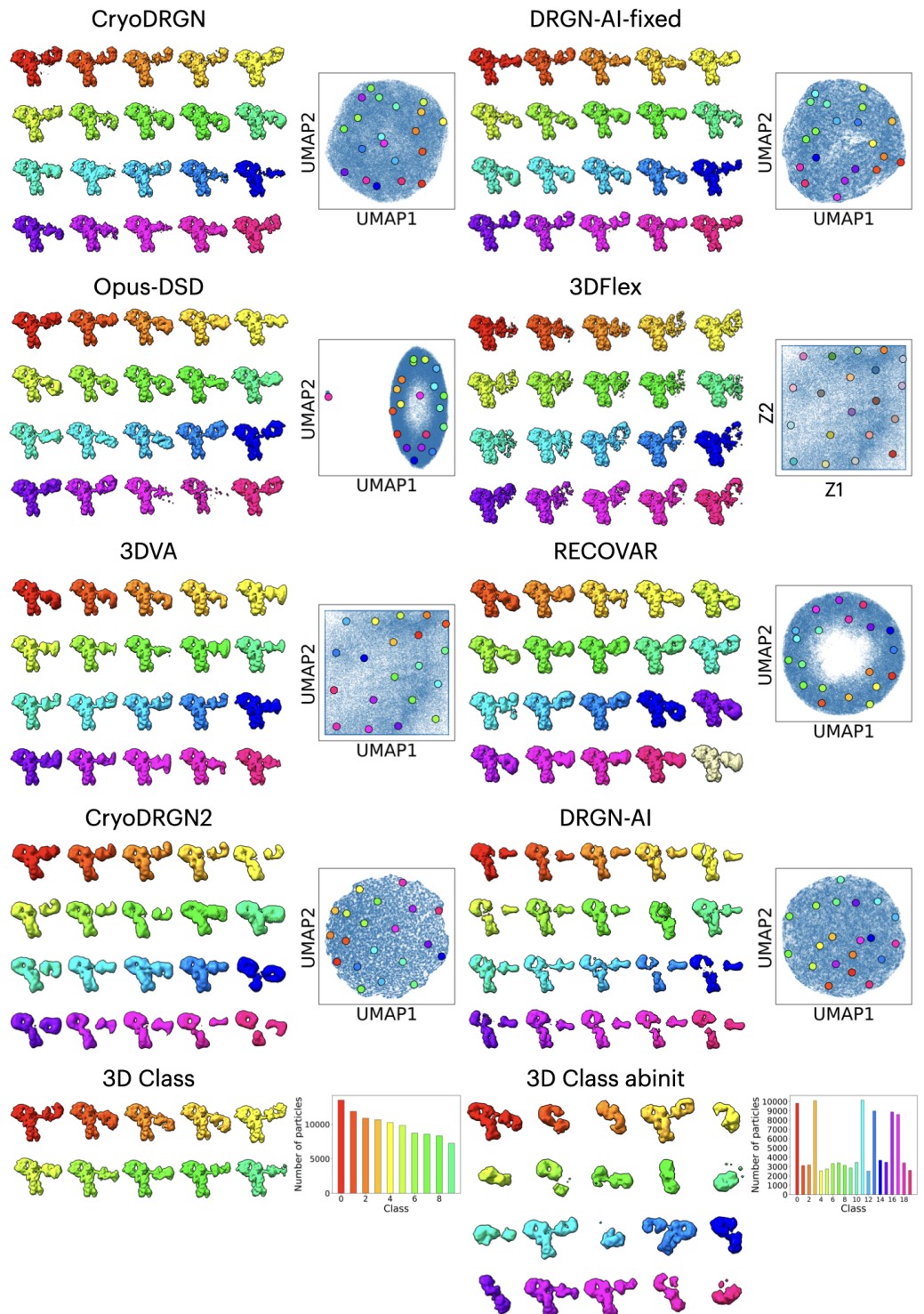

Figure 24: **Qualitative Results (IgG-1D noisiest).** For each method, representative volumes and a UMAP plot of the latent space are shown. Volumes correspond to K-Means cluster centers with K=20. Cluster centers are marked on the UMAP plot with a dot of the corresponding color. Class volumes and particle counts are shown for 3D Classification.

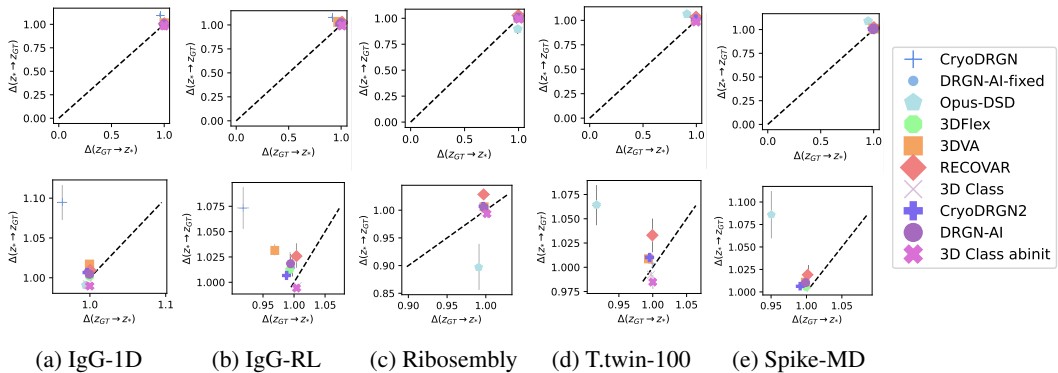

(a) IgG-1D    (b) IgG-RL    (c) Ribosembly    (d) T.twin-100    (e) Spike-MD

Figure 25: **Pose Information Imbalance**. In full view ($[0, 1]^2$; top row) and zoomed in (bottom row).

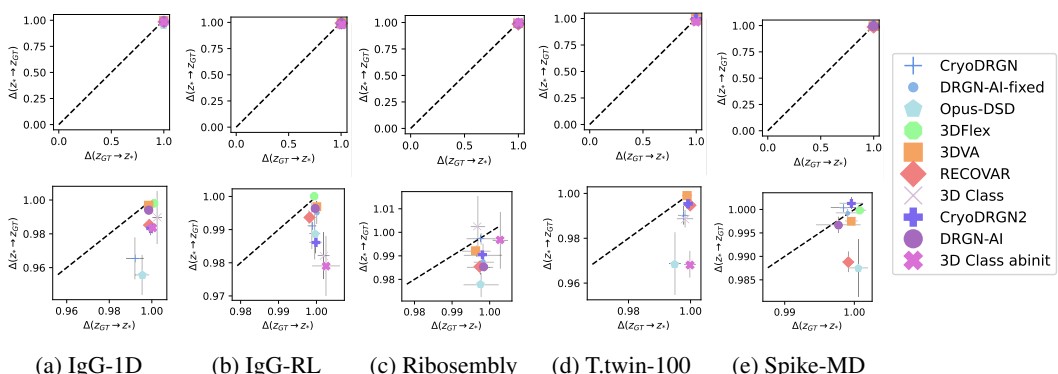

(a) IgG-1D    (b) IgG-RL    (c) Ribosembly    (d) T.twin-100    (e) Spike-MD

Figure 26: **CTF Information Imbalance**. In full view ($[0, 1]^2$; top row) and zoomed in (bottom row).

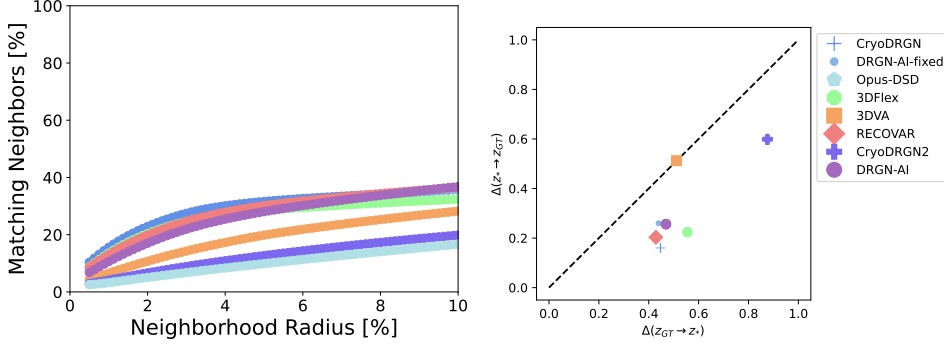

Figure 27: **Embedding metric results for the Spike-MD dataset** (left) Neighborhood similarity as a function of the neighborhood radius [%]. (right) Information Imbalance. CryoDRGN2 (not visible) is underneath Opus-DSD.

# E  Glossary of Terms from Single-Particle Electron Cryo-Microscopy

## E.1  Sample

- **Biomolecular**: Pertaining to molecules involved in the biological processes of living organisms, such as proteins and nucleic acids.

- **Protein**: Large, complex molecules made up of amino acids, essential for various biological functions like catalyzing metabolic reactions and DNA replication.

- **Nucleic Acid**: A type of biomolecule, including (deoxy)ribonucleic acid (DNA, RNA, respectively). This term can refer to a single unit that can polymerize (form a long chain).

- **Specimen**: The biological sample that is the object of investigation.

- **Complex**: In the context of biomolecular complexes, the term 'complex' refers to a stable association of two or more biomolecules that interact with each other, typically to perform a specific biological function. The interactions that hold these molecules together can be non-covalent, such as hydrogen bonds, ionic interactions, van der Waals forces, and hydrophobic effects, or covalent, such as disulfide bonds.

- **Subunit**: a part of a larger whole. The part (domain, polypeptide) is contextual to the whole (domain, protein complex).

## E.2  Data Source

- **Real, Experimental, Empirical**: Data based on observed and measured phenomena, derived from real-world evidence rather than theory or pure logic.

- **Synthetic, Simulated**: Data generated by algorithms or models, mimicking real-world data for testing and training purposes.

- **Protein Data Bank (PDB)**: A publicly accessible database for the three-dimensional structural data of large biological molecules such as proteins and nucleic acids. Atomic models are indexed by alphanumeric codes, and in this work we list them in the SI.

## E.3  Heterogeneity

- **Heterogeneity**: The presence of variations in shape or the presence or absence of mass within a sample. Coming in two main sub-classes
  - **Compositional**: Related to the total amount of mass and their proportions within a sample or structure. Often used in the context of discrete differences in total mass.
  - **Conformational**: Pertaining to the various shapes or structures that a molecule can adopt. Often used in the context of continuous movement in 3D space.

- **3D Structure**: The spatial form or shape of an object, which in the context of cryo-EM refers to the 3D structure of biomolecules. Often contrasted with the sequence of a biomolecule, or schematic (e.g. 2D) representations communicating atom type of bond connectivity.

- **Conformation**: The specific three-dimensional arrangement of atoms in a molecule. Often employed in the plural to refer to the different shapes a particular biomolecule can adopt.

- **Collective Variable (CV)**: A parameter used to describe the state of a system, typically in terms of a few degrees of freedom. Further distinguished into geometric (centre of mass, angle, distance) and abstract [54]. The term CV is related to 'order parameter', and 'reaction coordinate', which is often used in the context of reactants and products in chemical catalysis [55]. However, as employed in the biomolecular simulation community, CVs typically relate to distinguishing metastable states [56].

## E.4  Model and Representation

- **Angstrom (Å)**: A unit of length equal to $0.1$ nm, or $10^{-10}$ m. Often used in chemistry because the distance of and between atoms is close to $1$ Å.

- **Voxel**: A volume element representing an intensity value on a regular grid in three-dimensional space, similar to a pixel in 2D images but for a 3D array. A typical voxel volume ranges $0.5^3 - 2^3$ Å$^3$.

- **3D Map, Volume, Density, Model**: A representation of spatial data, in cryo-EM this typically refers to the 3D Coulombic (electric, electrostatic) potential instead of the electron density in other structural biology techniques based on X-ray diffraction. [57, 58]

- **Latent**: Hidden variables inferred from observed data, representing underlying structures or features in the model not directly observed.

- **Embedding**: A representation of data, for example a continuous n-dimensional vector space. Used to concretely parametrize or otherwise numerically represent a latent variable.

- **White Gaussian Noise**: noise with a flat power spectral density, meaning that its power is uniformly distributed across all frequencies. This implies that the noise has equal intensity at different frequencies, making it 'white' by analogy to white light, which contains all visible wavelengths.

## E.5 Microscopy

- **Point Spread Function (PSF)**: A function describing the response of an imaging system to a point source, indicating, for example, the system's resolution and blur characteristics.

- **Contrast Transfer Function (CTF)**: The Fourier transform of the point spread function. A mathematical description of how an electron microscope transfers contrast from the specimen to the image, influenced by various microscope parameters. We employ a common parametric form which depends on beam energy (electron wave length via the de Broglie relation), defocus and its astigmatism, spherical aberration, and amplitude contrast (ratio) **??**.

- **Microscope Effects**: Artifacts and distortions introduced by the electron microscope during image acquisition. At times used in a phenomenological sense to describe effects not modelled well by the PSF/CTF.

- **Camera Effects**: Distortions or noise introduced by the optical system used to capture images. Can be used in a wide sense beyond detector effects for the entire optical system.

## E.6 Image Acquisition and Analysis

- **Micrograph**: A two dimensional image obtained using an electron microscope, typically showing a field of view that includes multiple particles. Often the image contains temporal frames in a 'movie' format, which is corrected for motion. A typical micrograph is approximately $4000^2$ pix$^2$, at $0.5 - 2$ Å per pixel.

- **Particle**: Individual biomolecular structures captured within a patch of micrograph, which is typically boxed out of the wide frame image. Can refer to the physical entity in the image, or the recorded measurement. A typical particle is approximitely $64^2 - 512^2$ pix$^2$, at $0.5 - 2$ Å per pixel.

- **Reconstruction**: a 3D volume, typically in a real spaced voxelized array form, generated by processing data from a series of two-dimensional 2D images. Distinguished further to homogeneous (one 3D volume) and heterogeneous (multiple 3D volume).

## F Broader Impact

While the advancements in protein structure prediction offer tremendous potential benefits in biological discovery, there are also ethical considerations regarding data privacy, responsible technology use, and equitable access to healthcare innovations. Although our work focuses on synthetic benchmarks for Cryo-EM reconstruction tasks, it's important to note that our datasets are based on real data. Therefore, addressing these concerns is essential to ensure that deep learning technologies are deployed responsibly and ethically to maximize their positive societal impact.