# OpenReview forum: "CryoBench: Diverse and challenging datasets for the heterogeneity problem in cryo-EM"
_NeurIPS.cc/2024/Datasets_and_Benchmarks_Track — NeurIPS 2024 Track Datasets and Benchmarks Spotlight_

### Official Review · Reviewer_eH5Y · 2024-06-18
**Excellent and important study**

**Rating:** 8
**Confidence:** 4
**Clarity:** Yes.

**Review:**

The motivation, approach, data assessment, and initial benchmarking all is well-done. The data-sharing and code-sharing plan is unclear. I had some confusion over why the embedding metrics were chosen over others since these seem to be relatively recent metrics that were not previously used in the structural bio domain that the paper is in.

**Strengths:**

The paper is tackling a very important topic, limitations are well-written with a future roadmap of how to correct them, the data seems high quality and thoroughly assessed. I think this study will be very important and useful for the NeurIPS community.

**Additional Feedback:**

na.

**Correctness:**

I did not find any technical errors. The embedding comparison metrics are chosen from relatively new and somewhat low- (or under-)cited literature though not in the structural biology domain. I do wonder if there are more suitable metrics that may be more standard to structural bio or chem fields to add in addition to the currently used information metrics.

**Documentation:**

The data sharing and reproducibility / user-friendliness discussion needs to be improved.

**Ethics:**

No.

**Limitations:**

I didn't understand the data survey point the authors filled out: "Did you include the license to the code and datasets? [No] The code and the data are proprietary." What does this mean? If the data and code will not be shared this would greatly limit the study's use for the NeurIPS audience.

**Opportunities For Improvement:**

Minor point: capitalization is erratic in the references in terms of acronyms for techniques, "Cryo-em", etc.

**Relation To Prior Work:**

Yes.

**Summary And Contributions:**

Cryo-EM is becoming just as important as X-ray crystallography for protein structure and conformation studies. This study sets up a foundational benchmark for ML to make progress in this area. Metrics are well-motivated and demoed, the data entries are diverse in terms of features and difficulty, and the study is very well-written and convincing. I don't have any major suggestions other than some confusion over the data and code sharing plan. The authors should also make it more clear future plans for growing the repository of data, some code demos walking readers through how to use the data and tools, and explain whether future contributions to the dataset are invited.

---

> ### Author Rebuttal · Authors · 2024-08-17
>
> That was a mistake in the checklist. Thank you for catching this, and sorry for the confusion!
>
> We will upload all the code, including dataset pipeline, metrics, image formation, etc., to the github repository and make it publicly available. We provide the datasets under the Creative Commons Attribution 4.0 International license. Moreover, we have uploaded all datasets to Zenodo.
> 1. Conf-het: https://zenodo.org/records/11629428
> 2. Comp-het: https://zenodo.org/records/12528292
> 3. Spike-MD: https://zenodo.org/records/12528784
>
> A detailed guide to downloading the datasets is on the website (https://cryobench.cs.princeton.edu/).
> Thank you  for bringing the erratic capitalization and typos to our attention; We will correct them.
>
> We also appreciate your questions regarding the embedding comparison metrics. We are not aware of any standardized metrics for comparing cryo-em conformational heterogeneity embeddings, however our use of some metrics are inspired by a recent cryo-EM study [1] used several disentanglement metrics from ref. [2]. We believe that developing a “gold standard” embedding metric for cryo-EM is still an ongoing process, and we will incorporate additional discussion in the final version of our paper. Furthermore, we have added a quantification by standard clustering accuracy metrics for the compositional heterogeneity datasets (SI Table 5).
>
> [1] Klindt, D. A., Hyvärinen, A., Levy, A., Miolane, N., & Poitevin, F. (2024). Towards Interpretable Cryo-EM : Disentangling Latent Spaces of Molecular Conformations. ArXiv, 1–22.
>
> [2] Locatello, F., Bauer, S., Lucie, M., Rätsch, G., Gelly, S., Schölkopf, B., & Bachem, O. (2019). Challenging common assumptions in the unsupervised learning of disentangled representations. 36th International Conference on Machine Learning, ICML 2019, 2019-June, 7247–7283.

---

> > ### Comment · Reviewer_eH5Y · 2024-08-17
> > **Reviewer has no further concerns.**
> >
> > Thanks! This answers all my concerns satisfactorily and was clear.

---

### Official Review · Reviewer_iw7S · 2024-07-10

**Rating:** 7
**Confidence:** 4
**Correctness:** yes
**Clarity:** yes

**Review:**

1. I think the benchmark on Cryo-EM and ML-aided tools are very important for the research community. The data construction, metrics are carefully designed in this benchmark.

2. The paper is clearly presented.

3. It might be more comprehensive to include clarification on more related literature, such as [1], [2] and [3]

4. The benchmark can be more useful and meaningful if the data and code are released.


[1] SHREC 2020: Classification in cryo-electron tomograms

[2] De novo structural pattern mining in cellular electron cryotomograms

[3] AITom: Open-source AI platform for cryo-electron Tomography data analysis

**Strengths:**

see above

**Additional Feedback:**

no

**Documentation:**

yes

**Ethics:**

yes

**Opportunities For Improvement:**

see above

**Relation To Prior Work:**

yes

**Summary And Contributions:**

The paper proposes CryoBench, a suite of datasets, metrics, and performance benchmarks for heterogeneous reconstruction in cryo-EM.

---

> ### Author Rebuttal · Authors · 2024-08-17
>
> Thanks for the suggestion about related papers! We will include those three papers in the final version.
>
> We have uploaded all datasets to Zenodo:
>
> 1. Conf-het: https://zenodo.org/records/11629428
> 2. Comp-het: https://zenodo.org/records/12528292
> 3. Spike-MD: https://zenodo.org/records/12528784
>
> Regarding our github, we are working on uploading all the code, including dataset pipeline, metrics, image formation, etc., to the github repository and making it publicly available. And we have recently created a website which has a detailed guide to downloading the datasets (https://cryobench.cs.princeton.edu/).

---

### Official Review · Reviewer_D6Nq · 2024-07-25
**Authors present "CryoBench: Datasets and Benchmarks for Heterogeneous Cryo-EM Reconstruction", an approach for standarized benchmarking for Heterogeneous Cryo-EM reconstruction.**

**Rating:** 8
**Confidence:** 5
**Correctness:** No comments
**Clarity:** Yes

**Review:**

1. Quality

The submission is of high quality, presenting thorough evaluations and detailed comparisons of state-of-the-art cryo-EM reconstruction methods.

2. Clarity

The work is well-organized, with clear descriptions of the datasets, methods, and metrics used for evaluation. The figures and tables are informative and support the text effectively. However, some sections, particularly those detailing the new metrics, could benefit from additional clarification to ensure they are accessible to a broader audience.

3. Originality

The introduction of new quantitative metrics for evaluating cryo-EM reconstruction methods is a significant contribution. The creation of five synthetic datasets designed to challenge current methods further demonstrates the originality of this work.

4. Significance

This work is highly significant for the field of cryo-EM and structural biology. By providing a standardized benchmark, CryoBench facilitates the comparison and improvement of heterogeneous reconstruction methods.


Pros

    1. Comprehensive Evaluation: Thorough assessment of ten state-of-the-art cryo-EM reconstruction methods using both qualitative and   quantitative metrics.
    2. Novel Metrics: Introduction of new metrics such as Neighborhood Similarity, Information Imbalance, and Per-Conformation FSC.
    3. Synthetic Datasets: Creation of challenging synthetic datasets that simulate realistic heterogeneity in cryo-EM data.
    4. Clear Presentation: Well-organized with informative figures and tables that enhance the clarity of the results.
    5. Future Directions: Thoughtful discussion of future work, highlighting potential improvements and new areas of research.

Cons

    1. Complexity of Metrics: Some of the new metrics introduced may be complex and could benefit from additional explanation or simplification.
    2. Ab Initio Methods: The performance of ab initio methods is relatively weak, indicating that further work is needed to improve these approaches.
    3. Real-World Data: The benchmark focuses on synthetic datasets, and while these are useful for controlled comparisons, the inclusion of real-world data would enhance the practical relevance of the evaluations.

**Strengths:**

Significance of the Contribution

    Innovative Benchmarking Framework: CryoBench introduces a novel framework for evaluating heterogeneous cryo-EM reconstruction methods, offering standardized metrics and datasets.
    New Quantitative Metrics: Introduces valuable metrics like Neighborhood Similarity, Information Imbalance, and Per-Conformation FSC for precise evaluations.

Relevance to the Broader Research Community

    Wide Applicability: Standardized datasets and metrics are useful across structural biology and cryo-EM, fostering a cohesive research community.
    Facilitates Method Development: Identifies strengths and weaknesses in current methods, guiding future advancements in cryo-EM technology.

Quality of the Research

    Comprehensive Evaluation: Thoroughly evaluates ten state-of-the-art methods using both qualitative and quantitative assessments for robust comparisons.
    Detailed Methodology: Provides extensive details on datasets, training procedures, and evaluation metrics, enhancing reproducibility.
    Challenging Datasets: Creates realistic synthetic datasets to push the boundaries of current methods.

Ethical and Social Implications

    Advancement in Structural Biology: Improved methods can accelerate discoveries in drug development and disease research.

**Additional Feedback:**

No

**Documentation:**

Yes

**Limitations:**

The authors have made a commendable effort in addressing the limitations and potential societal impacts of their work. However, there are areas where further discussion could enhance the manuscript:

Real-World Applicability:

The authors focus on synthetic datasets with known ground truths. I encourage authors to perform validation and benchmarking on real cryo-EM datasets to bridge the gap between synthetic data performance and real-world applicability. A small real-world data required for such evaluations may be available on https://www.nature.com/articles/s41597-023-02280-2

Complex Noise Models:

The noise statistics used might not fully capture the complexities of real-world data. I suggest to incorporate more sophisticated and realistic noise models in future datasets to better simulate actual experimental conditions.

Detailed Failure Analysis:

The manuscript provides a comprehensive evaluation but lacks detailed analysis of failure cases. A better idea is to include a thorough examination of why certain methods fail under specific conditions to provide deeper insights and guide future improvements.

**Opportunities For Improvement:**

Opportunities for Improvement


Scope of Datasets: While the synthetic datasets are challenging, incorporating more diverse and realistic datasets, including non-structural heterogeneity like junk particles and non-uniform pose distributions, would enhance relevance.


Ground Truth Dependence: The reliance on synthetic data with known ground truths limits the applicability to real-world scenarios where ground truth is unknown.


Generalization: The methods' performance on synthetic data may not directly translate to real cryo-EM data, necessitating validation on actual datasets.


Noise Statistics: The noise models used in synthetic datasets could be more complex to better reflect real experimental conditions.


Detailed Results: While the paper provides extensive evaluations, more detailed analysis of failure cases and insights into why certain methods perform better would be valuable.

**Relation To Prior Work:**

Yes

**Summary And Contributions:**

The submission introduces CryoBench, a benchmark designed for evaluating cryo-EM heterogeneous reconstruction methods. Key contributions include:

    1. Five synthetic datasets that showcase conformational and compositional heterogeneity.
    2. Comprehensive evaluation of ten state-of-the-art methods using innovative quantitative metrics and qualitative visualizations.
    3. Introduction of new metrics: Neighborhood Similarity, Information Imbalance, and Per-Conformation Fourier Shell Correlation (FSC).
    4. Key findings: RECOVAR and DRGN-AI demonstrate strong performance across datasets, while ab initio methods face challenges with complex mixtures.
    5. Future directions: Proposing the use of more complex noise models, additional metrics, and the incorporation of non-structural heterogeneity to better apply methods to real cryo-EM data.

CryoBench aims to advance cryo-EM and structural biology by challenging current methods and encouraging the development of new approaches.

---

> ### Author Rebuttal · Authors · 2024-08-17
>
> We appreciate the reviewer’s exceptionally positive review! To enhance the realism of our dataset, we’ve made two key improvements: introducing a realistic TEM noise simulator and implementing a non-uniform particle distribution in our Ribosembly dataset, as detailed in our response to R1.
>
> We will clarify the metrics (thank you for the feedback), and thanks for bringing up the cryoPPP dataset. While this is geared towards particle picking of homogeneous structures, we can add this as a citation in our related works section. We will also add more detailed analysis of failure cases and insights into why certain methods perform better in the final version.
>
> In the future, we’d like to consider other forms of realistic heterogeneity in our benchmark, including junk particles and non-uniform pose distributions, and potentially real datasets as well with our new metrics. Thank you for the suggestions.

---

> > ### Comment · Reviewer_D6Nq · 2024-08-24
> > **No further concern**
> >
> > The authors have made two key improvements to enhance the realism of dataset and agreed to clarify the metrics.
> >
> > This answers my concerns.

---

### Official Review · Reviewer_1HSM · 2024-07-28
**Review of CryoEM benchmarks**

**Rating:** 9
**Confidence:** 3
**Correctness:** Yes
**Clarity:** Yes

**Review:**

This paper is makes original contributions in benchmarking for CryoEM.  Publicly available datasets for 5 different tasks and measurable metrics to compare against.  The authors then provide comparison of several methods.  This is an important first result for the field on which other studies can then build.

**Strengths:**

The study is comprehensive and well-structured.  Each of the benchmark datasets are chosen to have complementarity.

**Additional Feedback:**

n/a

**Documentation:**

Yes

**Limitations:**

The authors acknowledge that the synthetic datasets could have more realism injected but that is left for future work.

**Opportunities For Improvement:**

It would be nice to have comparisons of the synthetic datasets with some real data to validate their utility.

**Relation To Prior Work:**

Yes, this is discussed in the paper and where this study extends previous work.

**Summary And Contributions:**

This paper presents 5 synthetic datasets for CryoEM with ground truth information; compares 10 SOTA methods against those datasets; and presents metrics (3 qualitative and 2 quantitative).

---

> ### Author Rebuttal · Authors · 2024-08-17
>
> We thank the reviewer for their extremely positive review! We’ve incorporated two changes to improve the realism of our datasets – a realistic TEM noise simulator and a non-uniform distribution of particles for our Ribosembly dataset (described in our response to R1).
>
> In the future, we’d like to consider other forms of realistic heterogeneity in our benchmark, including junk particles and non-uniform pose distributions, and potentially real datasets as well. Thank you for the suggestions.

---

### Official Review · Reviewer_8cUz · 2024-08-02
**Extensive benchmark of diverse tasks in cryo-EM**

**Rating:** 6
**Confidence:** 4
**Correctness:** Yes
**Clarity:** Yes

**Review:**

The datasets and metrics supplied are rigorous challenges using simulated data and appropriate measures for judging success. The task under consideration -- cryo-EM model building w/ structural heterogeneity -- generally lacks high quality benchmarks, so this work establishes an important milestone in the field.

**Strengths:**

This is a well-motivated benchmark that supplies appropriate tests in a challenging domain. Clear relevance to the broader structural biology research community.

**Additional Feedback:**

n/a

**Documentation:**

The github repository https://github.com/ml-struct-bio/CryoBench was not publicly available at the time of review. The Zenodo dataset was available.

**Ethics:**

No concerns

**Limitations:**

Yes, they absolutely have.

**Opportunities For Improvement:**

The opportunities for improvement are appropriately called out by the authors themselves: synthetic data is never going to have the same confounding noise properties as experimental data, so more and different noise processes will add to the relevance of the work.

**Relation To Prior Work:**

Yes

**Summary And Contributions:**

The Zhong lab leverages its expertise in cryo-EM and deep learning methods for structural heterogeneity to assemble a benchmark with multiple datasets and metrics.

---

> ### Author Rebuttal · Authors · 2024-08-17
>
> Thank you for the supportive feedback on our work. We have made a few changes summarized below that we will incorporate into the final version of this paper.
> We agree that using synthetic data is a limitation, in particular, if synthetic datasets don’t reflect the challenges encountered in real data. Despite this, we believe using synthetic data is an important aspect of our benchmarking as 1) it provides ground truth information for all latent variables, which enables rigorous evaluation of methods; this is especially important since metrics are lacking in cryo-EM, and 2) we focus on creating challenging forms of heterogeneity that are relevant to frontier challenges of real systems.
>
> However, to explore potential improvements in making our synthetic data as realistic as possible, we tested a more involved cryoEM particle simulator using cisTEM (Computational Imaging System for Transmission Electron Microscopy) [1]. This multislice simulator accounts for artifacts from sample thickness, noise from explicitly modeled water molecules, sample motion and radiation damage to the sample as a function of electron dose. We regenerated the IgG-1D dataset using cisTEM with a dose of 30 e-/Å2, distributed across 30 frames, a sample thickness of 50 nm and a multislice propagation distance of 5 Å; example images as compared to IgG-1D, IgG-1D-noisier, and 1gG-1D-noisiest, are provided in Figure 1 in the PDF. We run CryoDRGN, 3DVA, and DRGN-AI as representatives of a neural, non-neural, and ab initio method on this dataset; preliminary results are given in Figure 2 in the PDF. We note that this simulator takes a few minutes per image to generate, and given that the images (and performance of each method) are qualitatively similar to our simpler gaussian noise model, we did not follow this procedure for the benchmark datasets. We will include this as SI.
>
> As a second example of improving the realism of our dataset, since the initial submission of this work, we generated a non-uniform number of images for each ground truth structure in the Ribosembly dataset to follow the distribution in the original publication [2]. The dataset ranges from 3.9k particles in the rarest state to 40k particles for the most common state, and produces a dataset of 335,240 images. We will replacest this dataset in the paper (recently uploaded to arxiv), and we provide a Per-Conformation FSC in Table 1 and 2 in the PDF.
> N_particles: [9076, 14378, 23547, 44366, 30647, 38500, 3915, 3980, 12740, 11975, 17988, 5001, 35367, 37448, 40540, 5772]
>
> Regarding our github, we are working on uploading all the code, including dataset pipeline, metrics, image formation, etc., to the github repository and making it publicly available. And we have recently created a website which has a detailed guide to downloading the datasets (https://cryobench.cs.princeton.edu/).
>
> [1] Himes et al., Cryo-TEM simulations of amorphous radiation-sensitive samples using multislice wave propagation
>
> [2] Qin et al., Cryo-em captures early ribosome assembly in action, Nature Communications 2023

---

### Author Rebuttal · Authors · 2024-08-17

We appreciate all of our reviewers’ exceptionally positive reviews and constructive feedback. We will be incorporating additional discussions based on this feedback in the final version of our manuscript. They include investigating more realistic datasets (detailed in our response to reviewer 8cUz), more analysis between methods, including additional user-friendly guides, and clarifying the explanation of metrics. Finally, we will make all our datasets and codes publicly available.

---

### Decision · Program_Chairs · 2024-09-26

**Decision:**

Accept (Spotlight)

**Comment:**

All the reviewers unanimously recognize that the proposed benchmark for heterogeneous Cryo-EM reconstruction is highly significant and fills a significant gap in the field. The five datasets, evaluation metrics, and the benchmarking of the 10 methods are valuable for the community to test and develop Cryo-EM reconstruction methods.

The main strengths are: (1) five synthetic datasets that showcase conformational and compositional heterogeneity; (2) comprehensive evaluation of ten state-of-the-art methods using innovative quantitative metrics and qualitative visualizations; (3) introduction of new metrics; and (4) well-organized with informative figures and tables that enhance the clarity of the results.

The two main shortcomings noticed by the reviewers in the first version, (1) datasets are not fully released and (2) the noise in the simulated datasets may not be realistic, are well addressed by the authors in the rebuttal and revision phase.

Overall, this work is an important contribution that can help machine learning community develop AI methods for Cryo-EM reconstruction that can broadly benefit the structural biology field.